EMBO
reports

# Tetherin antagonism by SARS-CoV-2 ORF3a and spike protein enhances virus release

Hazel Stewart[1] (ID), Roberta Palmulli[1] (ID), Kristoffer H Johansen[1,2] (ID), Naomi McGovern[1] (ID), Ola M Shehata[3], George W Carnell[4], Hannah K Jackson[1] (ID), Jin S Lee[1] (ID), Jonathan C Brown[5] (ID), Thomas Burgoyne[6,7], Jonathan L Heeney[4], Klaus Okkenhaug[1] (ID), Andrew E Firth[1], Andrew A Peden[3] (ID) & James R Edgar[1,*] (ID)

## Abstract

The antiviral restriction factor, tetherin, blocks the release of several different families of enveloped viruses, including the Coronaviridae. Tetherin is an interferon-induced protein that forms parallel homodimers between the host cell and viral particles, linking viruses to the surface of infected cells and inhibiting their release. We demonstrate that SARS-CoV-2 infection causes tetherin downregulation and that tetherin depletion from cells enhances SARS-CoV-2 viral titres. We investigate the potential viral proteins involved in abrogating tetherin function and find that SARS-CoV-2 ORF3a reduces tetherin localisation within biosynthetic organelles where Coronaviruses bud, and increases tetherin localisation to late endocytic organelles via reduced retrograde recycling. We also find that expression of Spike protein causes a reduction in cellular tetherin levels. Our results confirm that tetherin acts as a host restriction factor for SARS-CoV-2 and highlight the multiple distinct mechanisms by which SARS-CoV-2 subverts tetherin function.

**Keywords** ORF3a; restriction factor; SARS-CoV-2; spike; tetherin
**Subject Categories** Membranes & Trafficking; Microbiology, Virology & Host Pathogen Interaction

## Introduction

The causative agent of coronavirus disease 2019 (COVID-19) is the coronavirus, severe acute respiratory syndrome coronavirus 2 (SARS-CoV-2; Matheson & Lehner, 2020; Wu *et al*, 2020). Sarbecoviruses are positive-sense single stranded RNA viruses, and their genomes encode a large non-structural replicase complex and four main structural proteins, spike (S), membrane (M), envelope (E), and nucleocapsid (N). In addition, the SARS-CoV-2 genome encodes other accessory proteins that facilitate replication, cell entry and immune evasion.

Viruses can be broadly categorised by the presence or absence of a host-derived lipid envelope. Membrane envelopment protects the viral capsid from the external environment, reduces host immune recognition, and aids viral entry to new cells. However, this envelopment process does provide an opportunity for host cells to integrate anti-viral factors within the forming virions.

SARS-CoV-2 cellular entry is mediated by spike protein binding to the host receptor angiotensin-converting enzyme-2 (ACE2; Hoffmann *et al*, 2020b). Unlike that of 2002–2003 SARS-CoV (hereafter named SARS-CoV-1), the SARS-CoV-2 spike protein contains a polybasic furin cleavage site which facilitates the cleavage of the spike into two proteins, S1 and S2 that remain non-covalently associated (Coutard *et al*, 2020; Hoffmann *et al*, 2020a). The S2 fragment is further primed by the serine protease TMPRSS2 (Hoffmann *et al*, 2020b), whilst the S1 fragment binds Neuropilin-1 (Cantuti-Castelvetri *et al*, 2020; Daly *et al*, 2020), facilitating virus entry and infection. Coronaviruses enter TMPRSS2-positive cells by direct fusion at the plasma membrane, and are endocytosed by TMPRSS2-negative cells (Koch *et al*, 2021), following which their envelope fuses within late endosomes/lysosomes (Burkard *et al*, 2014), liberating the viral nucleocapsid to the cytosol of the cell. Upon uncoating, the RNA genome is released into the host cytosol and replication-transcription complexes assemble to drive the replication of the viral genome and the expression of viral proteins. Coronaviruses modify host organelles to generate viral replication factories—so-called double-membrane vesicles (DMVs) that act as hubs for viral RNA synthesis (Wolff *et al*, 2020). SARS-CoV-2 viral budding occurs at ER-to-Golgi intermediate compartments (ERGIC) and newly formed

1   Department of Pathology, University of Cambridge, Cambridge, UK
2   Laboratory of Immune Systems Biology, National Institute of Allergy and Infectious Diseases, National Institutes of Health, Bethesda, MD, USA
3   Department of Biomedical Science, University of Sheffield, Firth Court, Sheffield, UK
4   Department of Veterinary Medicine, University of Cambridge, Cambridge, UK
5   Department of Infectious Disease, Imperial College London, London, UK
6   Royal Brompton Hospital, Guy's and St Thomas' NHS Foundation Trust, London, UK
7   UCL Institute of Ophthalmology, University College London, London, UK
    *Corresponding author. Tel: +44 1223 762607; E-mail: je333@cam.ac.uk

viral particles traffic through secretory vesicles to the plasma membrane where they are released to the extracellular space.

Tetherin (*Bst2*) is an interferon-inducible restriction factor that limits egress of several types of enveloped viruses, reducing the infection of neighbouring cells. Tetherin is a type II integral membrane protein with a short cytosolic tail, a single pass transmembrane domain, and an extracellular coiled-coil domain that is anchored to the membrane via a C terminal GPI anchor. Cysteine residues in the extracellular domain mediate homodimer formation via disulphide bonds, linking tetherin molecules on virus and cell membrane, leading to the retention of nascent viral particles on the surface of infected cells (Perez-Caballero *et al*, 2009). The cell surface retention of virions enhances their reinternalisation to endosomal compartments, limiting the extent of virus spread (Neil *et al*, 2006). Tetherin is expressed from a type-I Interferon-Stimulated Gene (ISG), and so although it is constitutively expressed by many cell types, expression of tetherin is enhanced by the presence of type-I interferon (IFN; Neil *et al*, 2007).

Enveloped viruses including human immunodeficiency virus 1 (genus *Lentivirus*; Neil *et al*, 2007, 2008), Ebola virus (genus *Ebolavirus*; Neil *et al*, 2007; Bates *et al*, 2009), Kaposi's sarcoma-associated herpesvirus (KSHV; genus *Rhadinovirus*; Mansouri *et al*, 2009) and human coronavirus 229E (HCoV-229E; *Alphacoronavirus*; Wang *et al*, 2014) undergo tetherin-dependent restriction. For tetherin-restricted enveloped viruses to produce fully released progeny, they have evolved means to counteract tetherin activity. Although the molecular mechanisms by which each virus downregulates tetherin differ, they all reduce tetherin abundance from the organelle in which respective viruses bud, and/or reduce tetherin levels upon the plasma membrane.

Of the previously described coronaviruses, HCoV-229E and SARS-CoV-1 have been shown to undergo viral restriction by tetherin (Wang *et al*, 2014; Taylor *et al*, 2015). Two SARS-CoV-1 proteins have been shown to antagonise tetherin resulting in a concomitant increase in virion spread—the ORF7a protein and spike glycoprotein (Taylor *et al*, 2015; Wang *et al*, 2019). In order for tetherin to tether coronaviruses, tetherin must be incorporated in the virus envelope during budding which occurs in intracellular organelles. The mechanism by which sarbecoviruses dysregulate tetherin remains unclear.

Here, we show that tetherin is directly responsible for tethering of nascent enveloped SARS-CoV-2 virions to infected cell surfaces. We demonstrate using primary cells and immortalised cell lines that SARS-CoV-2 infection causes a dramatic downregulation of tetherin and that loss of tetherin aids SARS-CoV-2 viral spread and infection. We examine the effect of expression of individual SARS-CoV-2 proteins on tetherin antagonism and find that ORF3a redirects tetherin away from the biosynthetic pathway and towards the endolysosomal pathway by defective retrograde traffic. The reduction of tetherin within the biosynthetic pathway limits its incorporation to forming virions, with subsequent enhancement in virus release. We also demonstrate that Spike expression causes tetherin downregulation, as has previously been described for SARS-CoV-1.

# Results

## SARS-CoV-2 downregulates tetherin

Airway epithelial cells are one of the initial sites of viral contact during SARS-CoV-2 infection. Differentiation of nasal primary human airway epithelial (HAE) cells results in a pseudostratified epithelium consisting of ciliated, goblet and basal cells, and SARS-CoV-2 readily infects ciliated cells (Pinto *et al*, 2022). Differentiated, mock-infected HAE cells displayed negligible levels of tetherin (Fig EV1A), reflecting low basal tetherin expression in the absence of IFN stimulation. Differentiated HAE cells were infected with SARS-CoV-2 and sections analysed by immunofluorescence microscopy to detect changes in tetherin. Infected cells, confirmed by anti-spike labelling, displayed lower tetherin fluorescence than neighbouring, uninfected cells (Figs 1A and EV1B and C). The levels of tetherin in infected HAE cells is lower than observed in uninfected neighbours in infected wells suggesting that infected HAE cells are able to generate IFN to act upon uninfected neighbouring cells, enhancing tetherin expression.

A robust IFN response is considered a key first line of defence against viral infection. Coronaviruses have developed multiple strategies to dampen IFN responses, and a hallmark of SARS-CoV-2 infection is the very weak IFN response (Sa Ribero *et al*, 2020). As an IFN stimulated gene, tetherin transcription is likely antagonised

**Figure 1. SARS-CoV-2 infection downregulates tetherin in primary human airway epithelial and HeLa + ACE2 cells.**

A   Nasal primary human airway epithelial (HAE) cells were infected with SARS-CoV-2 (MOI 0.01). Cells were fixed at 48 hpi and embedded to OCT. Cryostat sections were collected and prepared for confocal microscopy. Sections were immunolabelled with antibodies against SARS-CoV-2 spike (green)—to reveal SARS-CoV-2 infected cells, tetherin (red), and with phalloidin-647 (grey) and DAPI (blue).

B   ACE2 expressing stable HeLa cell lines were generated by lentiviral transduction. To confirm ACE2 expression, mock and ACE2 transduced cells were lysed and immunoblotted with anti-ACE2 antibodies. Anti-α-tubulin was used as a loading control.

C   HeLa + ACE2 cells were infected with SARS-CoV-2 (MOI 0.5). Cells were fixed at 24 hpi and stained with antibodies against SARS-CoV-2 spike (green), and with DAPI (blue). Uninfected cells are shown with asterisks.

D   HeLa + ACE2 cells were infected with SARS-CoV-2 (MOI 0.5). Cells were fixed at 24 hpi and stained with antibodies against SARS-CoV-2 spike (green), tetherin (red), and with DAPI (blue). Uninfected cells are shown by asterisks.

E   Electron micrographs showing plasma membrane-associated SARS-CoV-2 virions, and virus filled intracellular organelles. SARS-CoV-2 infected HeLa + ACE2 cells (MOI 0.5) were fixed at 24 hpi and processed for TEM. Micrographs show (i) plasma membrane-associated virions, (ii) virus-filled compartments and virions associated with the plasma membrane, (iii) virions within intracellular organelles and DMVs. Plasma membrane-associated virions (some, but not all) are highlighted by arrowheads. DMVs (some, but not all) are highlighted by asterisks. Virions can be seen budding within internal organelles, shown by white arrow.

F   Surface immunogold electron microscopy of SARS-CoV-2 infected HeLa + ACE2 cells. Cells were infected with SARS-CoV-2 (MOI 0.5), fixed at 24 hpi and immunogold surface labelled using an anti-tetherin antibody.

G   As in (F) but cells were labelled with an anti-SARS-CoV-2 spike antibody.

Source data are available online for this figure.

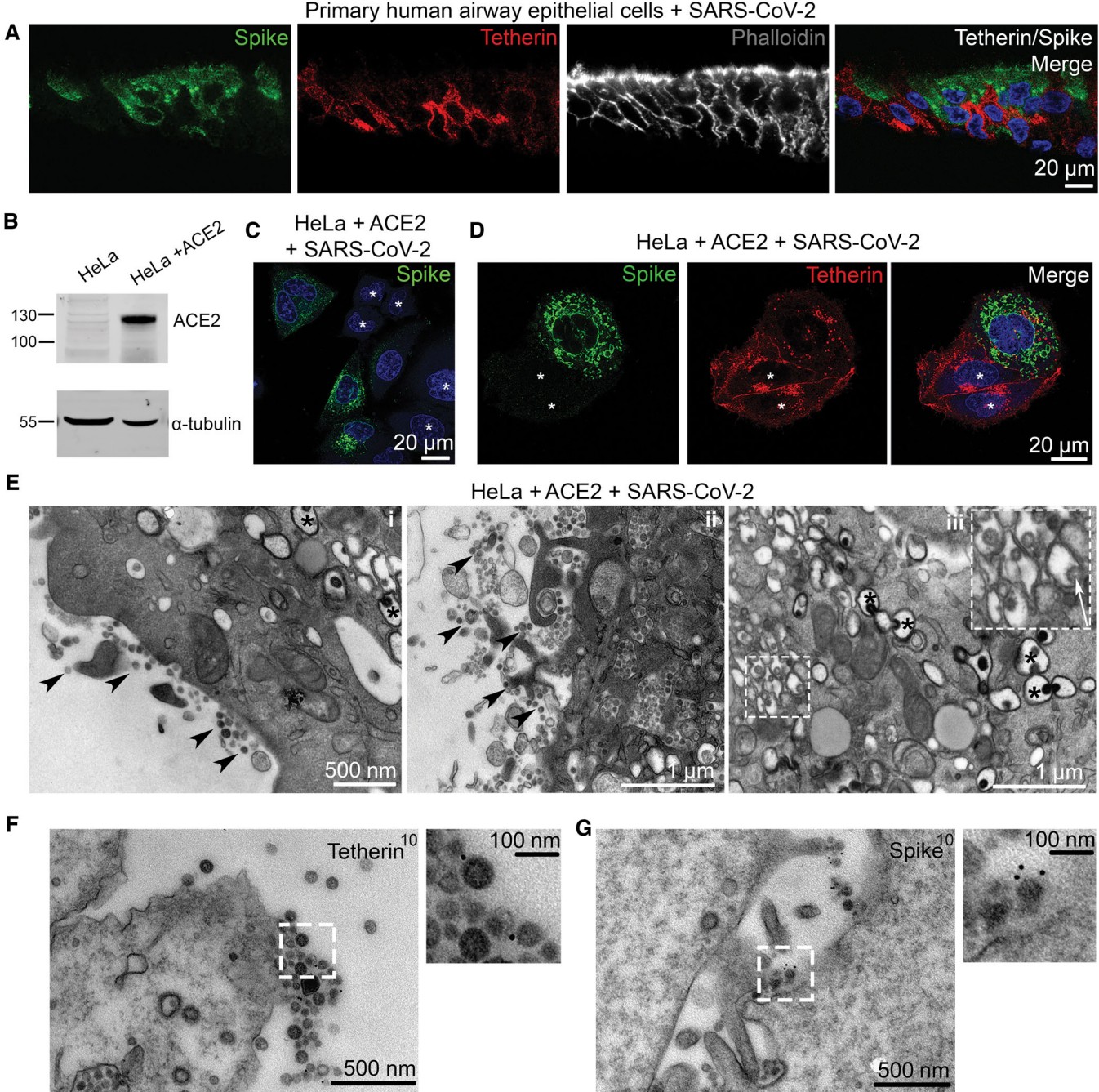

**Figure 1.**

by SARS-CoV-2. To determine whether SARS-CoV-2 reduces tetherin independently of IFN antagonism, we assessed the impact of SARS-CoV-2 infection on HeLa cells which constitutively express high levels of tetherin, without the need of induction through IFN. To allow SARS-CoV-2 infection, we generated a HeLa cell line stably expressing ACE2 by lentiviral transduction, designated as HeLa + ACE2, and ACE2 protein expression was confirmed by western blotting (Fig 1B). Using a clinical isolate of SARS-CoV-2 (isolate BetaCoV/Australia/VIC01/2020; Caly *et al*, 2020), we performed viral infection assays and fixed the cells 24 h post

infection (hpi). Infected cells were confirmed by anti-spike labelling (Fig 1C, uninfected cells shown with asterisk).

Uninfected HeLa + ACE2 cells display tetherin localised to the plasma membrane, perinuclear region and to peripheral punctate organelles. Upon SARS-CoV-2 infection, tetherin signal was decreased (Figs 1D and EV1D and E—uninfected cells shown with asterisk) and quantification of immunofluorescence images revealed a 47% decrease in total tetherin fluorescence in SARS-CoV-2 infected cells (Fig EV1E). Whilst tetherin appeared to be broadly downregulated from the plasma membrane, some cell surface staining remained in

these cells that often colocalised with spike labelling (Fig EV1D), likely representing areas of surface-tethered SARS-CoV-2 virus.

To examine whether SARS-CoV-2 virions were tethered to the cell surface, we performed transmission electron microscopy. In infected HeLa + ACE2 cells, SARS-CoV-2 virions could be found clustered on the plasma membrane of cells, although tethered virions were frequently polarised to discrete areas, rather than distributed evenly along the plasma membrane (Fig 1E, arrowheads). Electron microscopy also verified the presence of DMVs (Fig 1E, asterisks) in infected cells, and intracellular virions were also observed (Fig 1E, white arrow).

To check whether SARS-CoV-2 infection caused a global downregulation of surface proteins, we stained infected cells for the surface protein beta2microglobulin but no obvious loss in plasma membrane staining was observed (Fig EV1F). Surface labelling immunogold electron microscopy was performed (see Materials and Methods) and revealed tetherin molecules to be found between SARS-CoV-2 virions (Fig 1F), and virions were verified as SARS-CoV-2 virions rather than other forms of extracellular vesicle using an anti-SARS-CoV-2 spike antibody (Fig 1G).

The human alveolar epithelial cell line, A549, expresses low levels of ACE2 endogenously, and ectopic overexpression of ACE2 has been used to facilitate betacoronavirus entry (Jia et al, 2005; Yang et al, 2020). A549 cells stably expressing ACE2, designated A549 + ACE2, were generated by lentiviral transduction (Fig 2A) and these cells were amenable to SARS-CoV-2 infection (Fig 2B, uninfected cells shown with asterisk). A549 cells do not express tetherin at steady state, although its expression can be induced through stimulation with IFN alpha (IFNα; Neil et al, 2007; Giese & Marsh, 2014; Fig EV1G), following which tetherin is localised to the plasma membrane and to intracellular compartments. Immunofluorescence analysis of A549 + ACE2 cells infected with SARS-CoV-2 revealed a dramatic loss of tetherin as revealed by the loss of tetherin in SARS-CoV-2 infected cells (Fig 2C, uninfected cells shown with asterisk). Tetherin levels in infected cells were quantified to be 35% of uninfected levels by immunofluorescence (Fig EV1H). To examine what effect this near-total loss of tetherin had on virus tethering, we again performed electron microscopy. SARS-CoV-2 infected, IFNα treated A549 + ACE2 cells were characterised by significant intracellular remodelling, but very few

surface-associated virions were present, likely due to the significant tetherin downregulation (Figs 2D and EV1I). Virion-containing DMVs were frequently observed in the perinuclear region of infected cells, and these were associated with dramatic membrane remodelling, including a loss of typical Golgi cisternae from cells (Figs 2E and F).

Although SARS-CoV-2 infections predominantly cause pathology to the respiratory system, gastrointestinal epithelial cells, where ACE2 is highly expressed, are also infected. The human colonic adenocarcinoma epithelial cell line, T84, which express endogenous ACE2 (Nie et al, 2004) were examined for their ability to be infected by SARS-CoV-2 and to tether virions. SARS-CoV-2 infection of T84 cells was accompanied by a decrease in total levels of tetherin (Figs 2G and EV1J). Electron microscopy analysis of infected T84 cells revealed significant virus tethering at microvilli (Fig 2H) along flat regions of the plasma membrane, and formation of virus-filled intracellular compartments (Fig EV1K). The differences in the amount of tetherin downregulation between these cell lines may reflect either differences in kinetics of infection, differences in resting levels of tetherin, or differences in host machinery involved in the process of downregulation. These data are consistent with previous observations suggesting that other coronaviruses downregulate tetherin (Wang et al, 2014, 2019; Taylor et al, 2015).

## Tetherin loss aids SARS-CoV-2 viral spread

To determine whether tetherin plays a functional role in viral tethering, we performed both one-step (MOI of 5) and multi-step (MOI of 1 and 0.01) growth curves, to measure both released and intracellular virus production from infected WT HeLa + ACE2 and Bst2KO HeLa + ACE2 cells (Fig 3A). HeLa cells were used due to their high, homogenous and IFN-independent expression of tetherin. Bst2KO HeLa cells used were previously described (Edgar et al, 2016). Both WT HeLa + ACE2 and Bst2KO HeLa + ACE2 could be infected with SARS-CoV-2 (Fig 3B). WT HeLa + ACE2 and Bst2KO HeLa + ACE2 cells were infected at the respective MOI and released and intracellular virus was harvested at the indicated time points. The virus titre of all samples was measured by plaque assay (Fig EV2) and the amount of released virus, as a percentage of total virus, was calculated (Fig 3C).

**Figure 2. SARS-CoV-2 infection of epithelial cell lines.**

A A549 cells were transduced with ACE2 lentivirus to generate stable cell lines. Mock and ACE2 transduced cells were lysed and immunoblotted with anti-ACE2 antibodies. Anti-alpha-tubulin served as a loading control.

B A549 + ACE2 cells were infected with SARS-CoV-2 (MOI 0.5). Cells were fixed at 24 hpi and stained for SARS-CoV-2 spike (green) to reveal infected cells, and DAPI (blue). Uninfected cells are shown by asterisks.

C A549 + ACE2 cells were treated with IFNα (1,000 U/ml, 24 h) to upregulate tetherin expression. Cells were then infected with SARS-CoV-2 (MOI 0.5) and fixed at 24 hpi. Cells were immunolabelled using antibodies against SARS-CoV-2 spike (green), tetherin (red), and with DAPI (blue). Uninfected cells shown with asterisks.

D A549 + ACE2 cells were treated with IFNα (1,000 U/ml) and infected with SARS-CoV-2 (MOI 0.5), fixed at 24 hpi and processed for TEM. In a small proportion of infected cells, virion clusters were observed at the plasma membrane (shown by arrowheads) and significant DMV formation (some, but not all DMVs are highlighted with asterisks). Further DMV examples shown in Fig EV1I.

E Electron micrographs of the perinuclear region of mock A549 + ACE2 cell. Zoomed area highlights typical Golgi morphology with mock cells comprising parallel cisternae (arrowheads).

F Electron micrographs of the perinuclear region of SARS-CoV-2 infected A549 + ACE2 cells. Infected cells were infected at an MOI of 0.5 and fixed at 24 hpi. Zoomed area highlights the loss of typical Golgi cisternae, and the appearance of DMVs.

G T84 cells were infected with SARS-CoV-2 (MOI 0.5) and fixed at 24 hpi and stained for spike (green) and tetherin (red).

H T84 cells were infected with SARS-CoV-2 (MOI 0.5) and fixed at 24 hpi and processed for TEM. Tethered virions were frequently present at the plasma membrane and are highlighted by arrowheads.

Source data are available online for this figure.

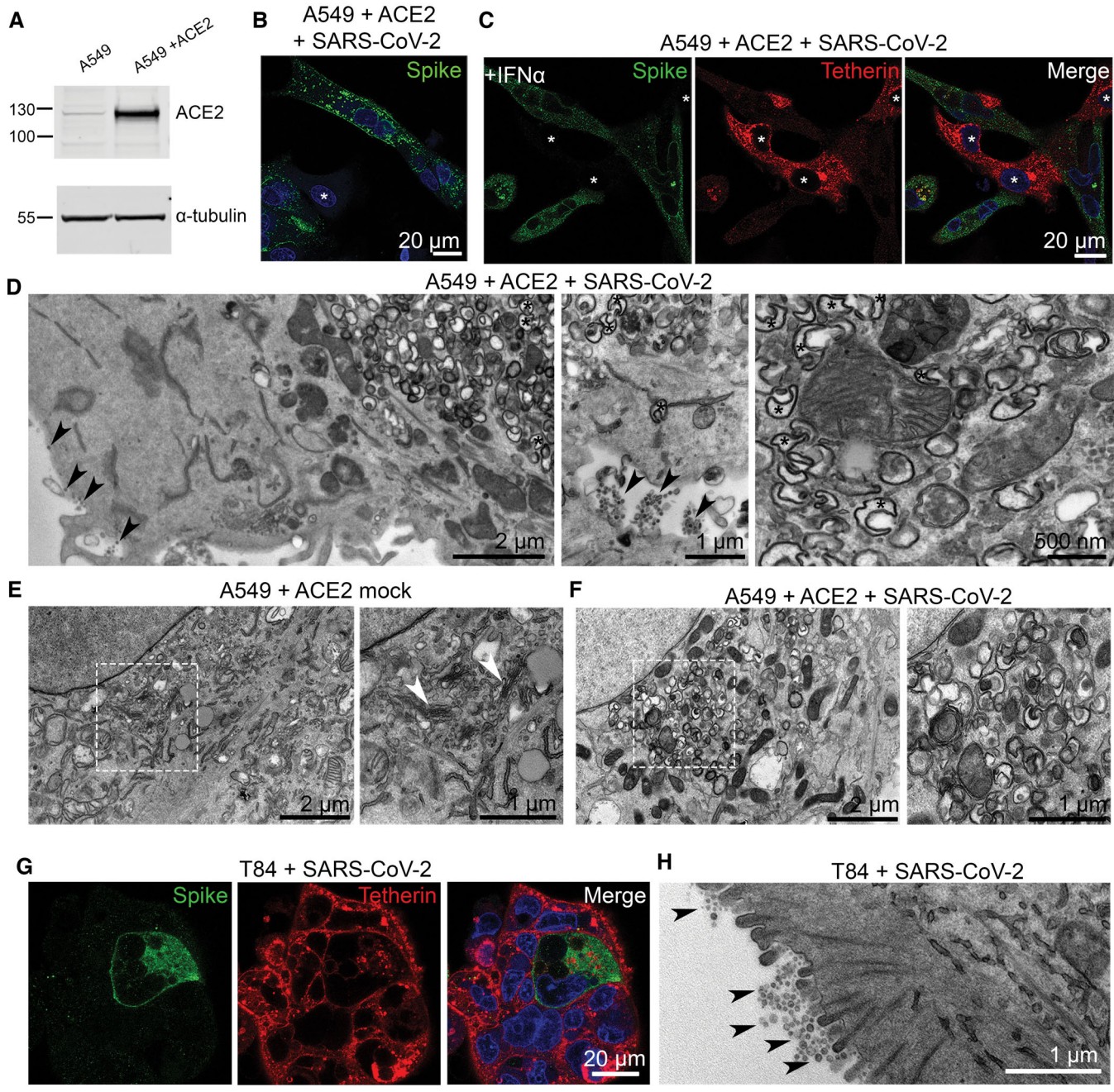

**Figure 2.**

For all MOIs tested, intracellular virions accumulated primarily in the first 24 h post infection. WT HeLa + ACE2 cells showed significantly higher intracellular titres than Bst2KO cells in the lowest MOI (0.01) experiments (Fig EV2B), whilst the released virus titres were similar (Fig EV2A). This provides evidence of tetherin-mediated restriction, whereby virions accumulate in WT cells. In contrast, at the higher MOIs of 1 and 5, the intracellular viral titres were similar regardless of the cell genotype whilst the Bst2KO cells released reproducibly higher virion numbers.

Irrespective of the MOI used, when the released virions were viewed as a proportion of total infectious particles, the Bst2KO

HeLa + ACE2 cells consistently released significantly more than WT HeLa + ACE2 cells at both 24 and 48 hpi, due to their inability to tether nascent virions (Fig 3C).

Together, these data demonstrate that tetherin acts to limit SARS-CoV-2 infection and that SARS-CoV-2 acts to downregulate tetherin. These data support the notion that tetherin exerts a broad restriction against numerous enveloped viruses, regardless of whether budding occurs at the plasma membrane or within intracellular compartments. As previously studied coronaviruses, including HCoV-229E and SARS-CoV-1, have been demonstrated to downregulate tetherin (Wang *et al*, 2014, 2019; Taylor *et al*, 2015) we next aimed to

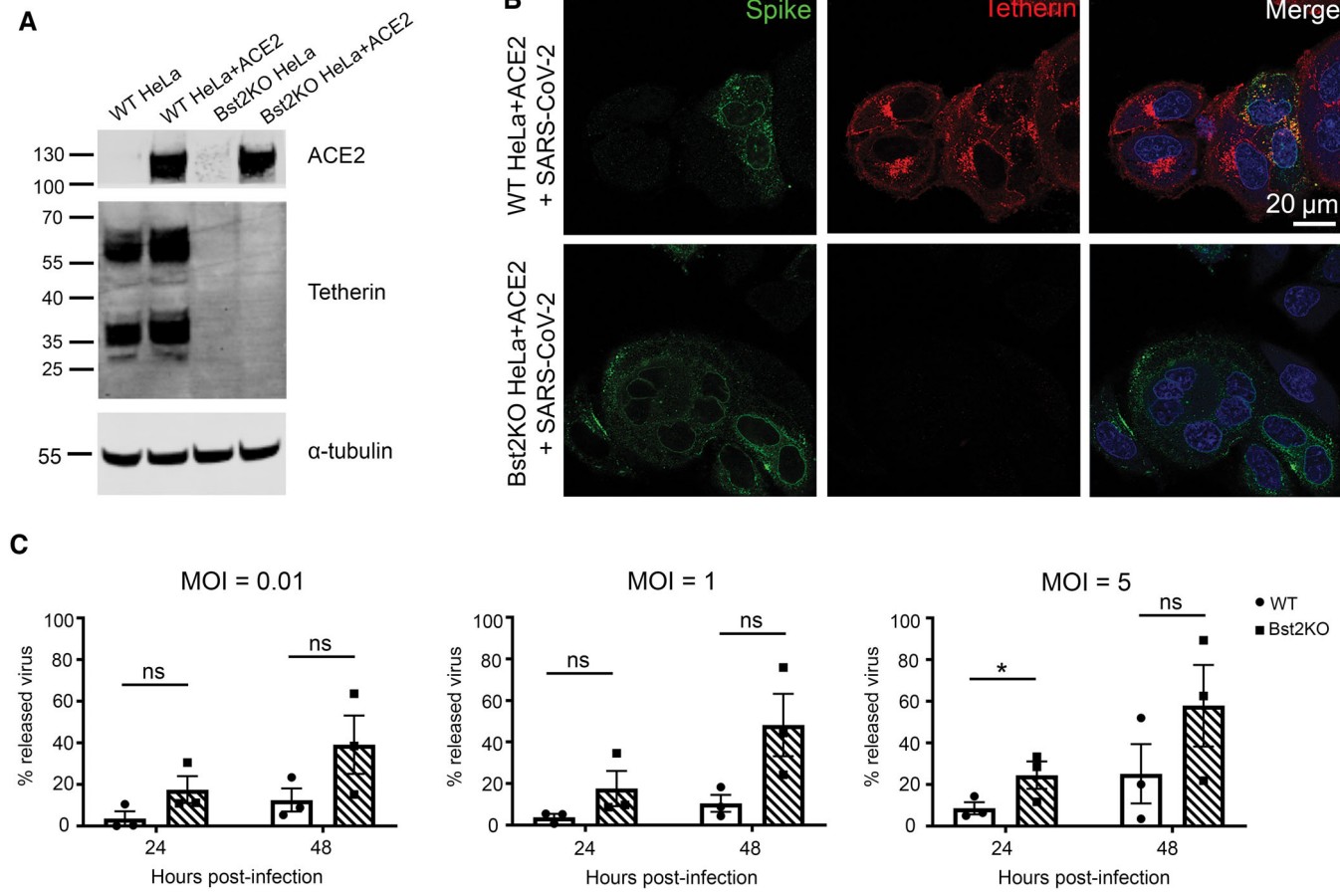

**Figure 3. Viral growth curves reveal tetherin loss enhances viral spread.**

A Lentiviral ACE2 was used to generate stable WT HeLa + ACE2 and Bst2KO HeLa + ACE2 cells. Tetherin KO and ACE2 expression were verified by western blotting. GAPDH served as a loading control.

B WT HeLa + ACE2 and Bst2KO HeLa + ACE2 cells were infected with SARS-CoV-2 (MOI 0.5) and fixed at 24 hpi. Cells were stained for spike (green) to demonstrate infection with SARS-CoV-2, and tetherin (red).

C Viral growth curves were performed by infecting WT HeLa + ACE2 and Bst2KO HeLa + ACE2 cells with SARS-CoV-2 at the indicated MOI (0.01, 1 or 5). Released and intracellular viral titres were measured by plaque assay. The amount of virus released into the supernatant is displayed as a percentage of total virus for each sample (released/released + intracellular), taking into account volumes and dilution factors. Data represents the mean $\pm$ SEM of three independent biological replicates. Ratio paired $t$-tests were performed ($ns$ [not significant] $P > 0.05$, *$P < 0.05$).

Source data are available online for this figure.

determine which SARS-CoV-2 protein is responsible for tetherin downregulation.

## ORF7a protein does not alter endogenous tetherin abundance, glycosylation or dimer formation

ORF7a has been shown to antagonise tetherin in both SARS-CoV-1 (Taylor et al, 2015) and SARS-CoV-2 (Martin-Sancho et al, 2021), although the mechanism of antagonism is not well understood. SARS-CoV-2 ORF7a has acquired a number of mutations that exist in and around the transmembrane domain which plays a role in SARS-CoV-1 ORF7a localisation. We found that SARS-CoV-1 ORF7a colocalises with the trans-Golgi marker TGN46, but that SARS-CoV-2 ORF7a localises additionally to small puncta (Fig EV3A and B). Golgi markers TGN46 and ZFPL1 were both found to be less

reticular and fragmentation of Golgi markers was observed in HeLa cells stably expressing SARS-CoV-2 ORF7a (Fig EV3C and D). Such Golgi fragmentation is similarly observed in both HeLa and A549 cells infected with SARS-CoV-2 (Figs 2E and 2F, and EV3E).

To determine if ORF7a expression affected tetherin localisation, abundance, glycosylation or ability to form dimers, HeLa cells stably expressing SARS-CoV-1 ORF7a-FLAG or SARS-CoV2-ORF7a-FLAG were analysed by immunofluorescence, western blotting and flow cytometry. By immunofluorescence ORF7a-FLAG and tetherin were not found to significantly overlap, but we did observe that the two proteins were both localised to some degree in adjacent perinuclear organelles (Fig 4A). Antagonism of tetherin by ORF7a could occur by interfering with tetherins ability to form homodimers. SARS-CoV-1 ORF7a was previously shown to alter glycosylation of ectopic tetherin without altering protein abundance, and the glycosylation-defective

tetherin showed impaired ability to restrict SARS-CoV-1 egress (Taylor *et al*, 2015). To determine whether SARS-CoV-2 ORF7a impaired endogenous glycosylation or affected tetherin dimer formation, we performed western blotting of WT HeLa, Bst2KO HeLa, KO + C3A-tetherin, and WT HeLa cells stably expressing either SARS-CoV-1 ORF7a-FLAG or SARS-CoV-2-ORF7a-FLAG (Fig 4B). C3A-tetherin stable cells (expressed in a tetherin KO HeLa background), are unable to form tetherin dimers and this can be observed by western blotting. The stable expression of either SARS-CoV-1 ORF7a or SARS-CoV-2 ORF7a did not impact upon tetherin abundance, glycosylation, or the ability of tetherin to form dimers (Fig 4B). Finally, flow cytometry determined that expression of SARS-CoV-1 or SARS-CoV-2 ORF7a did not impact upon cell surface levels of tetherin (Fig 4C). Overall, these data show that ORF7a does not directly influence tetherin localisation, abundance, glycosylation or dimer formation.

### Tetherin is downregulated by SARS-CoV-2 spike

SARS-CoV-1 Spike causes tetherin downregulation via lysosomal degradation (Wang *et al*, 2019). To interrogate the impact of SARS-CoV-2 Spike on tetherin, we generated an epitope tagged Spike construct to aid our imaging studies. We placed a HA epitope immediately after the native signal peptide sequence (ss-HA-Spike), rendering the HA tag at the N terminus of the mature protein. Transient transfection of cells with ss-HA-Spike caused a 32% decrease in tetherin as observed by immunofluorescence (Fig EV4A and B), with tetherin being primarily lost from the plasma membrane. By flow cytometry, we found no impairment in surface levels of ss-HA-Spike (Fig EV4C), or in the ability of ss-HA-Spike to bind ACE2 (Fig EV4D) versus untagged Spike.

Attempts to generate stable cell lines constitutively expressing Spike were unsuccessful, most likely due to the accumulation of non-viable multinucleated cells (despite the lack of ACE2 expression), so we generated an inducible ss-HA-Spike cell line using the lentiviral TetOne system. Tetracycline-inducible ss-HA-Spike stable HeLa cells were generated, and expression of Spike was analysed by immunofluorescence microscopy following induction through Doxycycline treatment (Fig 4D). Induction of Spike expression with

Doxycycline in TetOne stable cells caused a decrease in cellular levels to 60% that of untreated cells (Fig EV4E). Doxycycline-induced ss-HA-Spike expression resulted in the rapid formation of numerous multi-nucleated syncytia (Fig EV4F), as reported by others (Daly *et al*, 2020; Hoffmann *et al*, 2020a,b). Flow cytometry demonstrated a loss of surface tetherin upon by expression of Spike (Fig 4E), and western blotting confirmed a loss of total cellular tetherin protein upon Spike expression (Figs 4F and EV4G). To determine whether Spike specifically downregulates tetherin, or affects other host proteins, we analysed the surface levels of another integral transmembrane protein, CD71, and found that the expression of Spike had a very minor impact on CD71 (Fig EV4H).

To explore whether the Spike-induced tetherin downregulation altered virus release, we performed experiments with virus like particles (VLPs) in HEK293T cells which do not express endogenous tetherin. Western blots of lysates showed that the transient expression of Spike caused cellular tetherin levels to be reduced by 19% (Figs 4G and EV4I). VLPs were isolated from supernatants of cells expressing FLAG-tagged SARS-CoV-2 structural proteins (Nucleocapsid (N), Membrane (M), Envelope (E). VLP yields were impaired by the additional expression of tetherin (Fig 4G), confirming tetherins ability to retain SARS-CoV-2 VLPs. N-FLAG was most abundantly expressed in cells and recovered best in VLP fractions, and quantification of VLP blots demonstrated expression of tetherin reduced VLP release by 74% (Fig EV4J). These data demonstrate that tetherin can impair the release of SARS-CoV-2 VLPs and that SARS-CoV-2 Spike does not downregulate tetherin sufficiently to overcome this restriction.

### ORF3a alters tetherin localisation by impairing retrograde traffic and enhances VLP release

Given the dramatic loss and relocalisation of tetherin in SARS-CoV-2 virally infected cells, yet only minor downregulation by Spike, we performed a miniscreen of SARS-CoV-2 ORFs to identify additional SARS-CoV-2 proteins involved in tetherin downregulation or antagonism. We transiently transfected HeLa cells with: ORF3a-Strep, ORF6-Strep, ORF7a-Strep, Strep-ORF7b, ORF8-Strep, ORF9b-Strep, Strep-ORF9c, ORF10-Strep and confirmed their expression by

---

**Figure 4. Spike but not ORF7a expression downregulates tetherin.**

A   Stable SARS-CoV-1 ORF7a-FLAG and SARS-CoV-2 ORF7a-FLAG HeLa cell lines were generated and tetherin localisation was analysed by immunofluorescence. Representative confocal immunofluorescence microscopy images of fixed wild-type, SARS-CoV-1 ORF7a-FLAG or SARS-CoV-2 ORF7a-FLAG HeLa cells were stained using anti-FLAG (green) and anti-tetherin (red) antibodies.

B   Western blotting was performed to confirm the abundance, glycosylation and dimer formation of tetherin upon stable expression of SARS-CoV-1 ORF7a-FLAG and SARS-CoV-2 ORF7a-FLAG. Bst2KO cells rescued with C3A-Tetherin-HA express tetherin that is unable to form homodimers. Blots were performed in non-reducing conditions to highlight the loss of dimer formation by C3A-Tetherin.

C   Flow cytometry of wild-type (red), stable SARS-CoV-1 ORF7a-FLAG (blue) and stable SARS-CoV-2 ORF7a-FLAG (green) HeLa cells to analyse surface tetherin levels. Two biological replicates were performed, representative data shown.

D   Stable tetracycline-inducible (TetOne) ss-HA-Spike HeLa cell lines were generated. Cells were incubated with Doxycycline for 48 h to induce the expression of ss-HA-Spike. Representative confocal immunofluorescence image showing tetherin loss correlates with HA expression. Cells were stained using anti-HA (green) and anti-tetherin (red) antibodies.

E   Flow cytometry was performed on tetracycline-inducible ss-HA-Spike cells in resting conditions (no Doxycycline—red) or following induction (plus Doxycycline—blue). Surface tetherin levels were analysed. Two biological replicates were performed, representative data shown.

F   Western blot analysis of tetherin in TetOne ss-HA-Spike HeLa cells in the absence of Doxycycline or following induction with Doxycycline for 48 h.

G   Virus-like particle (VLP) experiments were performed using HEK293T cells. Cells were co-transfected with plasmids encoding N-FLAG, M-FLAG, E-FLAG, Tetherin, Spike. Overall expression of FLAG was used to confirm transfection and VLP production. Whole cell lysates were collected and blotted, and VLPs were isolated and purified from the culture supernatants. Blots were analysed using anti-FLAG, anti-tetherin, anti-Spike and anti-GAPDH (loading control) antibodies.

Source data are available online for this figure.

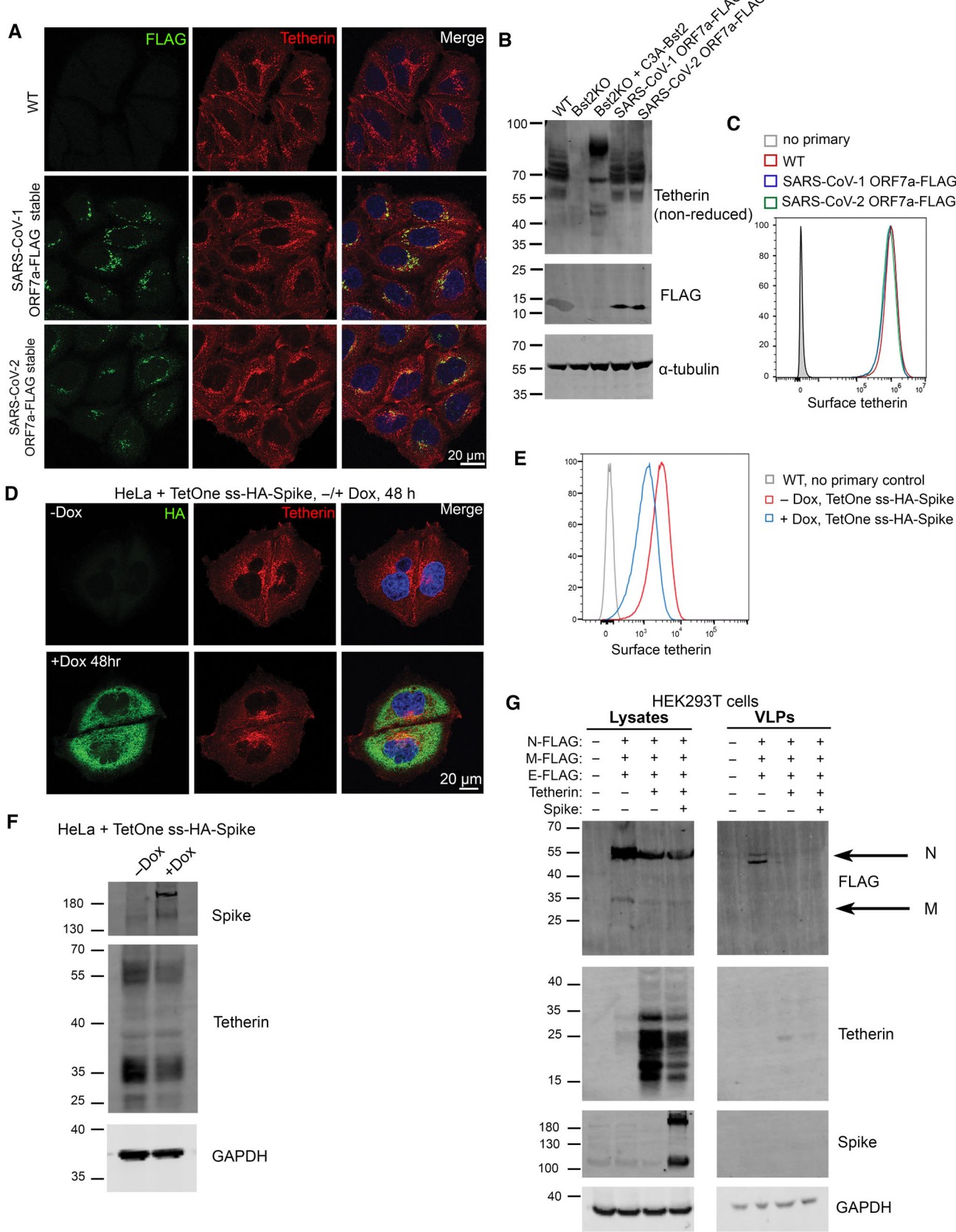

**Figure 4.**

intracellular flow cytometry using an anti-Strep antibody (Appendix Fig S1A). Surface tetherin levels were analysed by flow cytometry and no significant tetherin downregulation was observed upon expression of any ORF (Fig 5A). The intracellular tetherin localisation was analysed by confocal microscopy (Appendix Fig S1B) and we noticed a redistribution of tetherin towards punctate organelles upon expression of ORF3a (Fig 5B).

SARS-CoV-2 ORF3a is an accessory protein that localises to and perturbs endosomes and lysosomes (Miao *et al*, 2021). It may do so by acting either as a viroporin (Zhang *et al*, 2022a) or by interacting with, and possibly interfering with the function of VPS39, a component of the HOPS complex which facilitates tethering of late endosomes or autophagosomes with lysosomes (Miao *et al*, 2021; Miller *et al*, 2023). Given ORF3a likely impairs lysosome function, the observed increased presence of tetherin puncta following ORF3a expression may be due to decreased lysosomal degradation. Western blotting showed that expression of ORF3a had no impact on total levels of endogenous tetherin protein (Fig 5C). Flow cytometry confirmed minimal differences in tetherin levels upon transient expression of ORF3a (Fig EV5A). We noticed that tetherin localisation appeared altered upon ORF3a expression and observed a decrease in perinuclear tetherin that appeared in and around the Golgi (Fig 5D), and that tetherin localised to ORF3a-Strep positive punctate organelles. We quantified the levels of colocalisation between tetherin and the trans-Golgi marker, TGN46, and found a significant loss of tetherin from this region upon expression of ORF3a (Fig 5E). The loss of tetherin from the peri-Golgi area in ORF3a transfected cells was associated with an increase in tetherin within LAMP1 positive punctate organelles (Fig 5F and G). Expression of ORF3a also disrupted the distribution of numerous endosome-related markers including CIMPR, VPS35, CD63, which all localised to larger and less peripheral puncta (Fig EV5B), and the mixing of early (EEA1) and late endosomal markers (cathepsin D; Fig EV5C). By TEM, ORF3a expressing cells contained enlarged endolysosomes, consistent with defects in endolysosomal homeostasis (Fig EV5D).

The ORF3a-mediated increase in tetherin abundance within endolysosomes could be due to defective lysosomal degradation but could also be due to altered intracellular trafficking. In addition to the plasma membrane, tetherin is found in several intracellular organelles, including ERGIC, Golgi and endosomes. The persistence of proteins within biosynthetic organelles at steady state can be due to the presence of retention motifs or retrograde trafficking (Bonifacino & Rojas, 2006)—where proteins are recovered from endosomes back to the biosynthetic pathway.

Tetherin has recently been shown to undergo some retromer-dependent retrograde trafficking (Ding *et al*, 2021). We investigated whether ORF3a could impair the retrograde traffic of tetherin. Antibody uptake experiments were performed and the localisation of the internalised antibody assessed by immunofluorescence. In mock cells, fed anti-tetherin antibodies were found at the cell periphery and in the peri-Golgi region (Fig 5H), confirming retrograde recycling of tetherin. However, in cells expressing ORF3a, endocytosed tetherin was not recovered to the peri-Golgi region (Fig 5H). Colocalisation analysis confirmed a significant loss of anti-tetherin labelling overlapping with TGN46 upon the expression of ORF3a (Fig 5I). Internalised anti-tetherin antibodies were clearly found to colocalise with LAMP1 in ORF3a-Strep transfected cells (Fig EV5E), demonstrating that tetherin recycling was inhibited.

To establish whether the loss of retrograde traffic of tetherin was cargo specific, we similarly performed antibody pulse-chase experiments using the well-established retrograde cargo, CIMPR (Fig EV5F and G). Similar phenotypes were observed, indicating that ORF3a-Strep expression impairs global retrograde recycling and is not specific to tetherin.

To determine whether the ORF3a-mediated loss of tetherin from biosynthetic organelles altered SARS-CoV-2 egress, we again performed VLP experiments in HEK293T cells. HEK239T cells were transiently transfected with SARS-CoV-2 structural proteins, Tetherin and ORF3a and lysates and VLPs collected. The expression of ORF3a increased the cellular levels of tetherin in HEK293T cells

**Figure 5.   SARS-CoV-2 ORF3a redistributes tetherin away from the biosynthetic pathway via defective retrograde trafficking.**

A   A miniscreen was performed to analyse the ability of other SARS-CoV-2 ORFs to downregulate surface tetherin. HeLa cells were transiently transfected with Strep-tagged plasmids encoding: ORF3a-Strep, ORF6-Strep, ORF7a-Strep, Strep-ORF7b, ORF8-Strep, ORF9b-Strep, Strep-ORF9c or ORF10-Strep. 48 h post transfection, cells were stained for surface tetherin. Two biological replicates were performed. Representative data shown.

B   ORF3a-Strep transfected cells displayed intracellular tetherin accumulation. Tetherin accumulated in ORF3a-Strep positive compartments. Representative confocal immunofluorescence microscopy image of fixed HeLa cells transiently transfected with SARS-CoV-2 ORF3a-Strep. Cells were fixed and stained with anti-Strep (green) and anti-tetherin (red) antibodies and DAPI (blue). Non-transfected cells are shown with asterisks.

C   HeLa cells were transiently transfected with SARS-CoV-2 ORF3a-Strep. Mock and transfected cells were lysed 48 h post transfection and analysed by western blot. Blots were analysed using anti-tetherin, anti-Strep and anti-EF2 (loading control) antibodies.

D   Confocal immunofluorescence microscopy was performed on mock or ORF3a-Strep transiently transfected HeLa cells to analyse tetherin overlap with TGN46. Cells were fixed and stained using anti-TGN46 (green), anti-tetherin (red) antibodies, and DAPI (blue). Representative images are shown.

E   Colocalisation analysis was performed to quantify the Mander's overlap coefficient of tetherin overlapping TGN46. At least 20 cells per condition from three biological replicates were analysed. Individual data points are plotted with mean and standard deviation. Two-tailed, unpaired *t*-tests were performed ($****P < 0.0001$).

F   Confocal immunofluorescence microscopy was performed on mock or ORF3a-Strep transiently transfected HeLa cells to analyse tetherin overlap with the lysosomal marker, LAMP1. Cells were fixed and stained using anti-LAMP1 (green), anti-tetherin (red) antibodies, and DAPI (blue). Representative images are shown.

G   Colocalisation analysis was performed to quantify the Mander's overlap coefficient of tetherin overlapping LAMP1. At least 30 cells per condition from three biological replicates were analysed. Individual data points are plotted with mean and standard deviation. Two-tailed, unpaired *t*-tests were performed ($****P < 0.0001$).

H   Antibody uptake experiments were performed to investigate the fate of endocytosed tetherin. Transient transfections were performed 48 h prior to uptake experiments. Anti-tetherin antibodies were bound to live cells on ice for 30 min before a 2 h chase at 37°C. Cells were fixed and immunolabelled using anti-TGN46 (green), anti-Strep (white), secondary anti-rabbit555 (red) antibodies, and DAPI (blue). Representative images are shown.

I   Colocalisation analysis was performed to quantify the Mander's overlap coefficient of endocytosed anti-tetherin overlapping TGN46. At least 26 cells per condition from three biological replicates were analysed. Individual data points are plotted with mean and standard deviation. Two-tailed, unpaired *t*-tests were performed ($****P < 0.0001$).

Source data are available online for this figure.

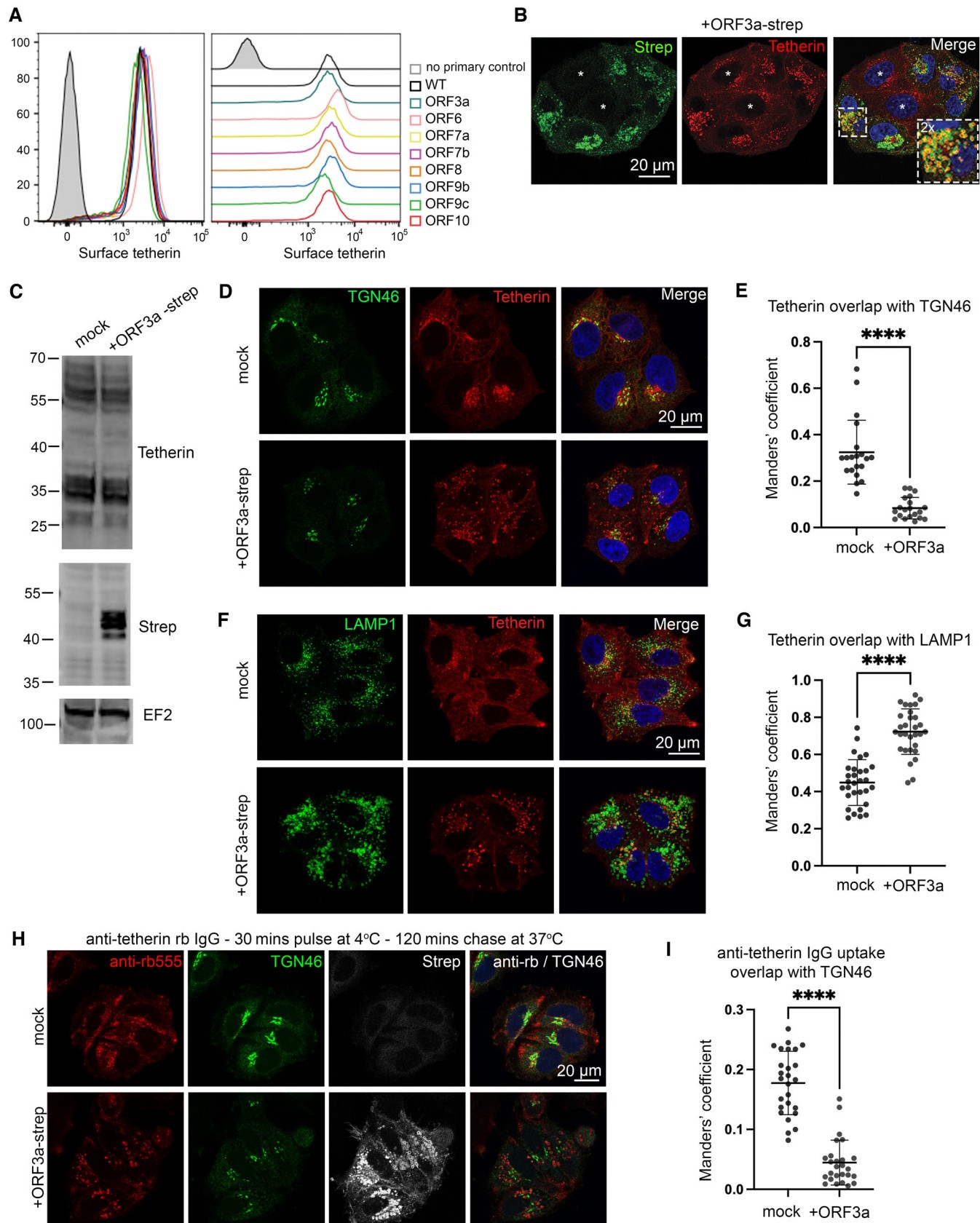

**Figure 5.**

(Fig 6A, Appendix Fig S2A). VLPs were collected and purified from the supernatants of cells and ran on western blots. VLP yields were reduced by 67% by the additional expression of tetherin (Fig 6A, Appendix Fig S2B), consistent with our previous data (Figs 4G and EV4I). The VLP yield was increased 4.3-fold by the expression of ORF3a alone, indicating that ORF3a may promote VLP release independently of its antagonism of tetherin. The co-expression of tetherin and ORF3a resulted in a 1.9-fold increase in VLP release, compared to expression of structural proteins alone (Fig 6A, Appendix Fig S2B).

We reproducibly found that the expression of tetherin reduced cellular levels of structural proteins (Figs 4G and 6A), and we hypothesised that tetherin expression could promote endocytosis and subsequent degradation of VLPs. Tetherin restriction of enveloped viruses prevents their egress, but also promotes internalisation of tethered virions which are redirected to lysosomal degradation. To test this, we expressed either wild-type tetherin or C3A-tetherin which is unable to form homodimers and therefore unable to promote viral retention and reinternalisation. Both WT and C3A-tetherin gave similar reductions in cellular levels of structural proteins (Fig 6B), indicating that reduced structural protein levels were independent of tethering, reinternalisation and subsequent degradation. As expected, the yield of VLPs from C3A-tetherin expressing cells was similar to that when no tetherin was expressed, confirming that tetherin homodimers are required for SARS-CoV-2 restriction (Fig 6C).

## Discussion

Tetherin has the ability to restrict numerous different enveloped viruses that bud at distinct organelles. Tetherin forms disulphide-linked homodimers that can link opposing membranes (e.g., plasma membrane and viral envelope). For tetherin to restrict SARS-CoV-2, tetherin molecules must be incorporated into the virus during viral budding, which occurs in modified ERGIC organelles. Many enveloped viruses antagonise tetherin by altering its localisation and removing it from the respective site of virus budding. Our data demonstrate that tetherin molecules become incorporated to SARS-CoV-2 virions and act to restrict virus release. SARS-CoV-2 counteracts this restriction in a variety of ways, ensuring its continued transmission.

Tetherin is an ISG (Neil et al, 2007), and many cell types express low levels of tetherin in the absence of stimulation. Weak type I IFN responses appear as a hallmark of sarbecovirus infections—SARS-CoV-1 is a poor inducer of type I IFN (Ziegler et al, 2005), and SARS-CoV-2 weaker still. By using cell lines that ubiquitously express high levels of tetherin and cell lines that require induction with IFN, we have demonstrated that SARS-CoV-2-mediated tetherin downregulation is not solely dependent upon dysregulation of IFN responses, and that other mechanisms exist to antagonise tetherin. In cells where tetherin is ubiquitously expressed and does not rely on IFN induction, SARS-CoV-2 infection downregulates and relocalises tetherin to punctate organelles (Fig 1D).

We identified SARS-CoV-2 ORF3a as a tetherin antagonist. Expression of ORF3a redistributed tetherin away from biosynthetic organelles and towards late endosomes and lysosomes (Fig 5D–G), and we demonstrated that this was caused by defective retrograde traffic (Fig 5H and I). Tetherin appears similarly redistributed during SARS-CoV-2 infection (Fig 1D). The resultant loss of tetherin from the biosynthetic pathway impaired its ability to retain SARS-CoV-2 VLPs (Fig 6A).

In this study, we used VLP assays to investigate virus egress. VLPs are advantageous in analysing virus egress over genuine SARS-CoV-2 virus because they are not impacted by factors relating to virulence or virus uptake, and instead allow one to assess egress alone. ORF3a-deletion viruses are severely attenuated (Castaño-Rodriguez et al, 2018; Liu et al, 2022) and do not infect cells at equal rates to wild-type SARS-CoV-2 virus.

The ORF3a-mediated defective retrograde trafficking is not specific to tetherin (Fig EV5F and G), and likely affects numerous other protein cargos, some of which may aid the infectivity and transmissibility of SARS-CoV-2. Our findings highlight a global defect in retrograde traffic as a novel mechanism of tetherin antagonism. Whether other enveloped viruses which bud within the biosynthetic pathway employ similar strategies remains to be investigated.

How ORF3a impairs late endocytic organelles is unclear. Expression of ORF3a impairs endolysosomal acidification (Ghosh et al, 2020) and promotes lysosome fusion with the plasma membrane (Chen et al, 2021). ORF3a has been proposed to dimerise and form an ion channel (Kern et al, 2021; Zhang et al, 2022a), although recent structural and functional experiments suggest that ORF3a is not an ion channel and instead sequestrates VPS39, impairing fusion of late endosomes with lysosomes (Miao et al, 2021; Miller et al, 2023).

We found that the expression of ORF3a enhanced VLP independently of its ability to relocalise tetherin (Fig 6A). This may be due to either the ability of ORF3a to induce Golgi fragmentation (Arshad et al, 2023) which facilitates viral trafficking (preprint: Zhang et al, 2022b), or due to enhanced lysosomal exocytosis (Chen et al, 2021). Tetherin was also found in VLPs upon co-expression with ORF3a (Fig 6A) which may also indicate enhanced release via lysosomal exocytosis (Chen et al, 2021).

**Figure 6. SARS-CoV-2 ORF3a enhances VLP release despite enhancing cellular tetherin.**

A  Virus-like particle (VLP) experiments were performed using HEK293T cells. Cells were co-transfected with plasmids encoding N-FLAG, M-FLAG, E-FLAG, Tetherin, ORF3a-Strep. Overall expression of FLAG was used to confirm structural protein transfection and VLP production. Whole cell lysates were collected, and VLPs were isolated from the culture supernatants. Blots were analysed using anti-FLAG, anti-tetherin, anti-Strep and anti-GAPDH (loading control) antibodies.

B  Experiments were performed to analyse the effect of tetherin-dependent reuptake and degradation of structural proteins. Cells were co-transfected with plasmids encoding N-FLAG, M-FLAG, E-FLAG, Tetherin, C3A-Tetherin. Whole cell lysates were collected. Blots were analysed using anti-FLAG, anti-tetherin and anti-GAPDH (loading control) antibodies.

C  VLPs were collected from (B) and the levels of VLP release was analysed using anti-FLAG antibodies.

Source data are available online for this figure.

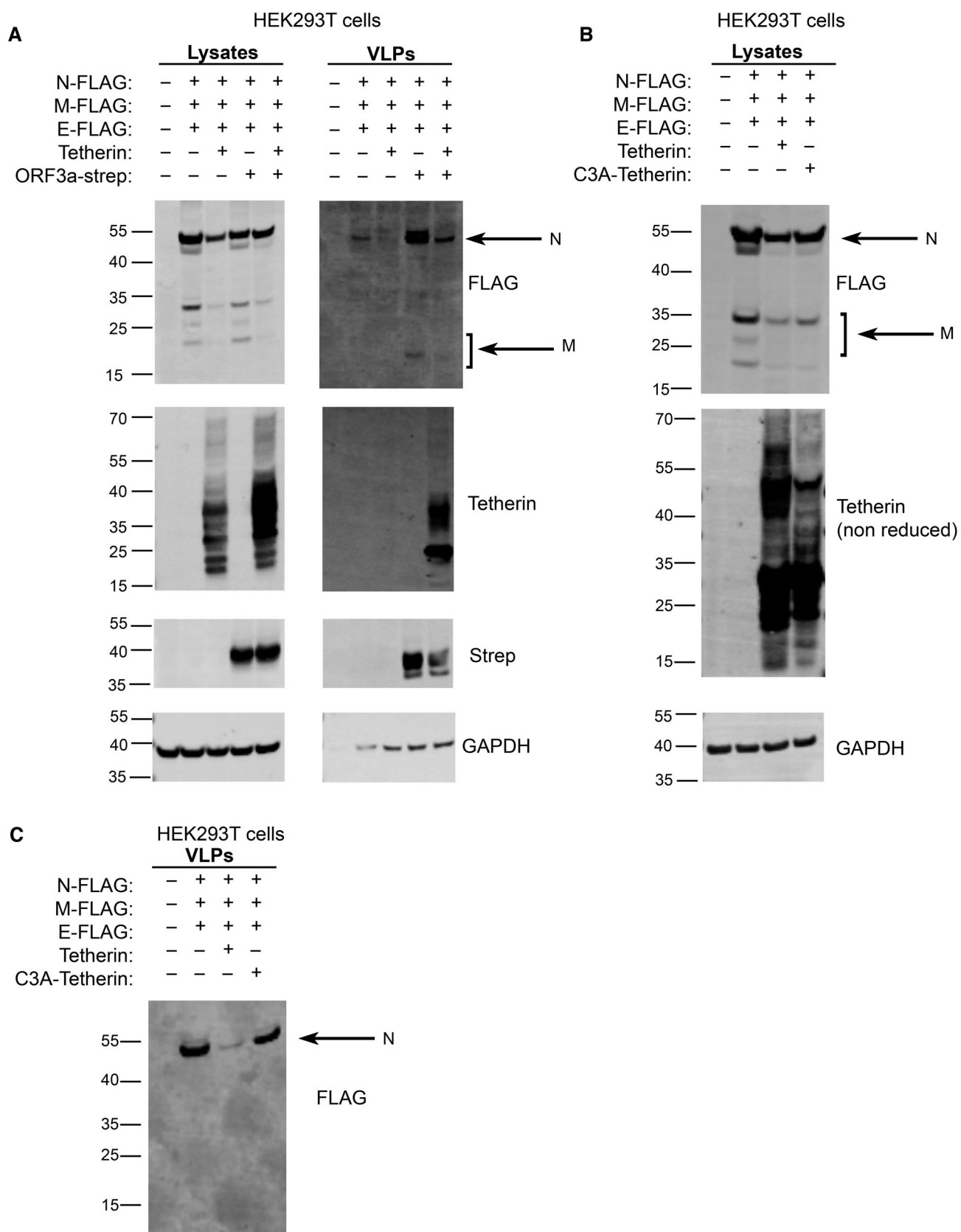

**Figure 6.**

The secretion of lysosomal hydrolases has been reported upon expression of ORF3a (Chen *et al*, 2021) and whilst this may in-part be due to enhanced lysosome-plasma membrane fusion, our data highlights that ORF3a impairs the retrograde trafficking of CIMPR (Fig EV5B, F and G), which may similarly increase hydrolase secretion.

Precisely how ORF3a impedes retrograde traffic is unclear. Recruitment of VPS35, a core retromer component, to membranes does not appear impeded by ORF3a expression (Fig EV5B), indicating that tubule formation or scission may be impaired. Endosomal acidification has been proposed to regulate tubule formation, as a homeostatic mechanism to ensure tubulation does not occur too early in the endocytic pathway or from incorrect organelles (Harbour *et al*, 2012), and so ORF3a-impaired endolysosomal acidification (Ghosh *et al*, 2020) may prevent proper retrograde tubules from forming on endosomes.

We also investigated the roles ORF7a and Spike—two proteins from SARS-CoV-1 which has previously been identified as tetherin antagonists. SARS-CoV-1 ORF7a is reported to inhibit tetherin glycosylation and localise to the plasma membrane in the presence of tetherin (Taylor *et al*, 2015). We did not observe any difference in total tetherin levels, tetherin glycosylation, ability to form dimers, or surface tetherin upon expression of either SARS-CoV-1 or SARS-CoV-2 ORF7a (Fig 4A–C). Others groups have demonstrated a role for ORF7a in sarbecovirus infection and both SARS-CoV-1 and SARS-CoV-2 virus lacking ORF7a show impaired virus replication in the presence of tetherin (Taylor *et al*, 2015; Martin-Sancho *et al*, 2021). A direct interaction between SARS-CoV-1 ORF7a and SARS-CoV-2 ORF7a and tetherin have been described (Taylor *et al*, 2015; Martin-Sancho *et al*, 2021), although the precise mechanism(s) by which ORF7a antagonises tetherin remains enigmatic. We cannot exclude that ORF7a requires other viral proteins to antagonise tetherin, or that ORF7a antagonises tetherin via another mechanism. For example, ORF7a can potently antagonise IFN signalling (Xia *et al*, 2020) which would impair tetherin induction in many cell types. In this study, we chose to use HeLa cells to examine the effect of ORF7a due to their high endogenous expression of tetherin, rather than using other cell types where tetherin needs to be ectopically expressed. The different cell lines and endogenous/ectopically tagged tetherin may account for differences between our results and others.

SARS-CoV-2 Spike has well-documented roles in facilitating SARS-CoV-2 viral entry, and in syncytia formation (Buchrieser *et al*, 2020). We found that SARS-CoV-2 Spike expression caused a mild downregulation of tetherin from cells, consistent with reports from SARS-CoV-1 (Wang *et al*, 2019), although in our experiments the downregulation was not sufficient to promote VLP release (Fig 4G). Overexpression of SARS-CoV-2 Spike can induce the unfolded protein response (UPR; Echavarría-Consuegra *et al*, 2021), and this may explain minor levels of tetherin loss observed.

SARS-CoV-2 can spread be transmitted between cells by cell-free dissemination through the extracellular space and through cell-to-cell transmission. Cell-to-cell transmission of SARS-CoV-2 occurs by a process mediated by Spike, but that does not require the host receptor ACE2 (Zeng *et al*, 2022). The cell-to-cell transmission of SARS-CoV-2 is sensitive to endosomal entry inhibitors, implicating endosomal membrane fusion as a mechanism for infection. Although tetherin impairs the egress of viruses from cells, for many viruses it also impairs cell-to-cell transmission. Tethered HIV-1 aggregates are not efficiently endocytosed by cells and the virus fusion capacities reduced (Casartelli *et al*, 2010; Kuhl *et al*, 2010; Giese & Marsh, 2014). Whether tetherin promotes or impairs cell-to-cell transmission of SARS-CoV-2 remains unknown.

The multiple mechanisms by which tetherin is antagonised by different enveloped viruses highlights the importance of overcoming restriction in determining the success of enveloped viruses.

# Materials and Methods

### Antibodies

#### *Primary antibodies used in the study were*

Rat anti-DYKDDDDK (L5) (BioLegend, WB 1:1,000, IF 1:200); rat anti-HA (Roche, 3F10, IF 1:500); rabbit monoclonal anti-tetherin (Abcam, ab243230, WB 1:2,000, IF 1:400, surface EM 1:200); mouse anti-SARS-CoV-2 Spike antibody 1A9 (GeneTex, GTX632604, WB 1:1,000, IF 1:300); rabbit anti-TGN46 (Abcam, ab50595, 1:300); rabbit anti-ZFPL1 (Sigma-Aldrich, HPA014909, 1:500); rabbit anti-Beta2microglobulin (Dako 1:500); mouse anti-ACE2 (Proteintech, 66699, WB 1:1,000), Phycoerythrin-conjugated anti-human tetherin (BioLegend, RS38E, FC 2 µg/100 µl/$10^6$ cells), Pacific Blue-conjugated anti-CD71 (Exbio, MEM-75, FC 1 µg/100 µl/$10^6$ cells), StrepMAB-Classic (IBA LifeSciences, WB 1:2,000, IF 1:500), StrepMAB-Classic DY-549 (IBA LifeSciences, FC 1:500), Strep-Tactin-DY649 (IBA LifeSciences, IF 1:2,000), mouse anti-CIMPR (2G11) (Abcam, ab2733, IF 1:200), mouse anti-VPS35 (Santa Cruz, B-5, IF 1:300), mouse anti-CD63 (BioLegend, H5C6, IF 1:300), mouse anti-EEA1 (BD Transduction, 610457, IF 1:200), rabbit anti-Cathepsin D (Calbiochem, 219361, IF 1:500), rabbit anti-GAPDH (Cell Signalling 14C10, WB 1:2,000), goat anti-EF2 (Santa Cruz, C-14, WB 1:10,000), mouse anti-tubulin (Proteintech, 66031, WB 1:10,000).

#### *Secondary antibodies used in this study were*

Goat anti-Mouse IgG Alexa488/555 and Goat anti-rabbit IgG Alexa 488/555 (ThermoFisher) secondary antibodies were used for confocal microscopy.

Goat IRDye 680 anti-mouse, anti-rabbit, anti-goat, anti-rat and Goat IRDye 800 anti-mouse, anti-rabbit antibodies (Li-Cor) were used for western blotting.

### Cloning

pcDNA6B SARS-CoV-1 and SARS-CoV-2 ORF7a-FLAG constructs were a gift from Professor Peihui Wang (Shandong University, China). To generate stable cell lines, ORF7a-FLAG cDNA fragments were subcloned into pQCXIH retroviral vectors.

ss-HA-Spike was generated by cloning an HA epitope plus a Serine-Glycine linker between residues S13 and Q14 of SARS-CoV-2 spike. Following translocation to the ER lumen, cleavage of the signal sequence will render the HA tag at the N terminus of the mature protein. Codon optimised SARS-CoV-2 spike was a gift from Dr Jerome Cattin/Professor Sean Munro (LMB, Cambridge, UK). ss-HA-Spike was originally cloned to pcDNA6B for transient transfection, and subsequently subcloned into pLVX-TetOne. Gene expression

from TetOne stable cell lines was induced by treatment with Doxycycline (1 µg/ml final concentration—Caymen Chemical). All cloning was verified by Sanger sequencing (GeneWiz).

The soluble ACE2 plasmid (pcDNA3-sACE2(WT)-Fc(IgG1) was a gift from Dr Erik Procko (Addgene plasmid: 145163; Chan et al, 2020) The StrepTactin tagged SARS-CoV-2 M protein plasmid (pLVX-EF1alpha-nCoV2019-M-IRES-Puro), and StrepTactin tagged SARS-CoV-2 ORF library was a gift from Dr David Gordon (Gordon et al, 2020).

Bst2-C3A-HA was a gift from Prof George Banting (University of Bristol) and the cDNA was cloned to pQCXIH for stable expression in Bst2KO HeLa cells.

### Immortalised cell lines

A549 cells were a gift from Dr Brian Ferguson, University of Cambridge, UK and were cultured in DMEM supplemented with 10% fetal bovine serum, L-glutamine, and Penicillin/Streptomycin, in 5% $CO_2$ at 37°C. T84 cells were purchased from ATCC and were cultured in DMEM: F-12 medium containing 5% fetal bovine serum, L-glutamine, and Penicillin/Streptomycin, in 5% $CO_2$ at 37°C. HeLa cells were a gift from Professor Scottie Robinson (CIMR, University of Cambridge, UK) and were cultured in DMEM supplemented with 10% fetal bovine serum, L-glutamine, and Penicillin/Streptomycin, in 5% $CO_2$ at 37°C. Bst2KO HeLa cells were previously described (Edgar et al, 2016). All cells were tested for mycoplasma tested using MycoAlert Mycoplasma Detection Kit (Lonza).

### Biosafety

Infection of HAE cells with SARS-CoV-2 were performed in a BSL-3 lab and the work was approved by the local genetic manipulation safety committee of Imperial College London, St. Mary's Campus (centre number GM77), and the Health and Safety Executive of the United Kingdom, under reference CBA1.77.20.1.

Infection of HeLa, A549 or T84 cells with SARS-CoV-2 were performed in a Biosafety Level 3 (BSL3) facility within the Department of Veterinary Medicine, University of Cambridge (the High Security Zone in the Laboratory of Viral Zoonotics, Rooms 231 and 229) approved by the United Kingdom Health and Safety Executive. The Departmental Biosafety Committee approved standard operating procedures and risk assessments before any experiments commenced. All samples were inactivated and decontaminated according to experimentally validated protocols, which had been previously approved by the Departmental Biosafety Committee.

### Human airway epithelium (HAE) cells

Nasal brushing samples were taken from healthy participants turbinate using a 3-mm bronchial cytology brush under Health Research Authority study approval (REC ref: 20/SC/0208; IRAS: 282739). The nasal brushings were placed in PneumaCult-Ex Plus medium (STEMCELL Technologies, Cambridge, UK) and cells extracted from the brush by gentle agitation. The cells were seeded into a single well of a collagen (PureCol from Sigma Aldrich) coated plate and once confluent, they were passaged and expanded further in a T25 flask. The cells were passaged a second time and seeded onto

Transwell inserts (6.5 mm diameter, 0.4 µm pore size, Corning) at a density of 24,000 cells per insert. Cells were cultured in PneumaCult-Ex Plus medium (STEMCELL Technologies, Cambridge, UK) until confluent before replacing with PneumaCult-ALI medium in the basal chamber and the apical surface exposed, giving an air–liquid interface to stimulate cilia biogenesis.

### SARS-CoV-2 infections of HAE cells

SARS-CoV-2 isolate hCoV-19/England/204501206/2020 (EPI_ISL_660791) was isolated from swabs as described in (preprint: Brown et al, 2021). The isolate was passaged twice in Vero cells before being used to infect HAE cells. To remove the mucus layer from the apical surface of HAE cells prior to infection, 200 µl of DMEM was added to the HAE cells at 37°C, 5% $CO_2$ for 10 min. The cells were infected at a MOI of 0.01 with inocula added to the apical chamber and incubated for 1 h at 37°C, 5% $CO_2$ before removal of the inoculum and incubating for a further 48 h.

### SARS-CoV-2 infections of immortalised cell lines

WT HeLa + ACE2, Bst2KO HeLa + ACE2, A549 + ACE2 or T84 cells were infected with isolate BetaCoV/Australia/VIC01/2020 (Caly et al, 2020), which had been passaged once on Vero cells following receipt from Public Health England. All cells were washed with PBS before being infected with a single virus stock, diluted to the desired MOI with sera-free DMEM (supplemented with 25 mM HEPES, penicillin (100 U/ml), streptomycin (100 g/ml), 2 mM L-glutamine, 1% non-essential amino acids). After 1 h, the inoculum was removed, and cells washed again with PBS. Infected cells were maintained in DMEM, supplemented with the above-described additions plus 2% FCS (virus growth media).

For immunofluorescence, cells were plated to glass-bottomed 24-well plates (E0030741021, Eppendorf) and infected at an MOI of 0.5 and incubated for 24 h, after which plates were submerged in 4% PFA/PBS for 20 min.

For conventional electron microscopy, cells were plated to plastic Thermanox (Nunc) coverslips in 24-well plates and infected at an MOI of 0.5 and incubated for 24 h, after which plates were submerged in 2% PFA/2.5% glutaraldehyde/0.1 M cacodylate buffer for 20 min.

For surface labelling immunoEM, cells were plated to Thermanox coverslips, infected at an MOI of 0.5 and incubated for 24 h, and fixed with 4% PFA/0.1 M cacodylate buffer for 20 min.

### Transient transfections

HeLa cells were transfected with 2.5 µg of DNA using TransIT-HeLa Monster (Mirus Bio) according to the manufacturer's instructions. Cells were analysed 48 h after transfection.

### SARS-CoV-2 ORF miniscreen

HeLa cells were transiently transfected with pLVX-EF1alpha-nCoV2019 2xStrep-tagged SARS-CoV2 ORF constructs (Gordon et al, 2020) using TransIT-HeLa Monster (Mirus Bio). After 48 h transfection, cells were detached and the cell population equally split. Half of the cells were permeabilised and intracellular Strep

levels were measured by flow cytometry. Surface tetherin staining was performed on the remaining half of the cells.

## Generation of stable cell lines

### Lentiviral constructs

ACE2 stable HeLa and A549 cell lines were generated using the lentiviral pLVX-ACE2-Blasticidin construct from Dr Yohei Yamauchi (University of Bristol). Following transduction, cells were selected with 10 μg/ml blasticidin for 18 days.

Expression-inducible stable cell lines were generated using the pLVX-TetOne system. pLVX-TetOne-Puro-ORF7a-2xStrep and pLVX-TetOne-Puro-2xStrep-ORF9c were a gift from Dr David Gordon (UCSF, USA). pLVX-TetOne-Puro-ss-HA-spike was generated as described above. Following transduction, cells underwent antibiotic selected with 1 μg/ml puromycin for 5 days.

HEK293T cells were transfected with lentiviral vectors (pLVX/pLVX-TetOne) plus packaging plasmids pCMVR8.91 and pMD.VSVG using TransIT-293 (Mirus Bio). Viral supernatants were collected 48 h after transfection, passed through 0.45 μm filters and recipient cells transduced by 'spinfection'—viral supernatants were centrifuged at 1,800 rpm in a benchtop centrifuge at 37°C for 3 h to enhance viral transduction.

### Retroviral constructs

pQCXIH-SARS-CoV-1-ORF7a-FLAG and pQCXIH-SARS-CoV-2-ORF7a-FLAG were transfected to HEK293T cells with the packaging plasmids pMD.GagPol and pMD.VSVG using Trans-IT293 (Mirus Bio). Viral supernatants were collected 48 h after transfection, passed through 0.45 μm filters and recipient cells transduced by 'spinfection'—viral supernatants were centrifuged at 1,800 rpm in a benchtop centrifuge at 37°C for 3 h to enhance viral transduction. Stable cells were selected using 400 μg/ml Hygromycin B for 10 days.

## Conventional electron microscopy

Cells were fixed (described above) before being washed with 0.1 M cacodylate buffer. Cells were stained using 1% osmium tetroxide +1.5% potassium ferrocyanide for 1 h before staining was enhanced with 1% tannic acid/0.1 M cacodylate buffer for 45 min. Cells were washed, dehydrated and infiltrated with Epoxy propane (CY212 Epoxy resin:propylene oxide) before being embedded in Epoxy resin. Epoxy was polymerised at 65°C overnight before Thermanox coverslips were removed using a heat-block. 70 nm sections were cut using a Diatome diamond knife mounted to an ultramicrotome. Ultrathin sections were stained with UA Zero (Agar scientific) and lead citrate. An FEI Tecnai transmission electron microscope at an operating voltage of 80 kV was used to visualise samples, mounted with a Soft Imaging System Megaview III digital camera.

## Surface immunogold labelling

To enable lumenal surface epitopes to be labelled, cells were fixed with 4% PFA/0.1 M cacodylate. They were washed with 0.1 M cacodylate buffer before being blocked with 1% BSA/PBS. Coverslips were inverted over drops of either rabbit anti-tetherin or rabbit anti-spike antibodies diluted in 1% BSA/PBS. Coverslips were

washed before being incubated with protein A gold. Following gold labelling, cells were re-fixed using 2% PFA/2.5% glutaraldehyde/0.1 M cacodylate before being processed for conventional electron microscopy as described above.

## Immunofluorescence microscopy

A549, T84 or HeLa cells were grown on glass coverslips and fixed using 4% PFA/PBS. Cells were quenched with 15 mM glycine/PBS and permeabilised with 0.1% saponin/PBS. Blocking and subsequent steps were performed with 1% BSA, 0.01% saponin in PBS. Cells were mounted on slides with mounting medium containing DAPI (Invitrogen). Cells were imaged using a LSM700 confocal microscope (63×/1.4 NA oil immersion objective; ZEISS).

## Immunofluorescence of cryostat sections

HAE cells were fixed in 4% PFA/PBS for 1 h before embedding and freezing in OCT (optimal cutting temperature) compound. ~ 15 μm thick sections were cut using a cryostat and these permeabilised using 0.2% saponin in PBS for 20 min at room temperature before incubating in blocking solution containing 0.02%, 1% BSA in PBS for 30 min at room temperature. The cryostat sections were incubated with primary antibodies in blocking solution for 2 h at room temperature. Subsequently, Alexa Fluor bound secondary antibodies and phalloidin 647 (Abcam, ab176759) were applied to the sections for 1 h at room temperature. Images were acquired using a Leica SP8 confocal.

## Fluorescence intensity quantification

Changes in fluorescence intensity were measured using ImageJ software. Cells were traced and mean fluorescence intensity calculated on a per cell basis. The mean, standard deviation, number of cells analysed and number of independent experiments are expressed in figure legends. Values are normalised to mock or control conditions per experiment. Due to variability in absolute values between independent experiments, values are normalised to the mean values. Unpaired, two-tailed $t$-tests were performed. *$P < 0.05$, **$P < 0.01$, ***$P < 0.001$, and not significant ($P > 0.05$).

## Immunofluorescence colocalisation analysis

Appropriate threshold values were manually applied to each channel and the Manders' overlap coefficient between two channels was quantified using the JACoP plugin of ImageJ Fiji software (Bolte & Cordelières, 2006). The same threshold values were applied to all the cells quantified. For analysis of ORF3a transient transfected cells, transfected cells were identified by anti-Strep-647 labelling and cell masks were manually drawn and applied to each channel for analysis.

## Antibody uptake experiments

Cells were seeded to glass coverslips in 24-well plates a day before uptake experiments were performed. Antibody was diluted in pre-chilled 1% BSA/PBS (to 5 μg/ml) and 50 μl drops pipetted to parafilm-covered blocks on ice. Coverslips were removed from

24-well plates are placed cell-side down to diluted antibody drops on ice. Cells were incubated on ice for 30 min to allow antibodies to bind to proteins at the cell surface. Coverslips were then removed and placed back in 24-well plates, and transferred back to a 37°C, 5% CO$_2$ incubator to allow antibody–antigen complexes to be endocytosed and trafficked for 2 h. Coverslips were then fixed and processed for immunofluorescence microscopy using secondary antibodies specific for internalised antibodies to detect antibody localisation.

## Western blotting

Cells were washed with PBS and scraped in pre-chilled lysis buffer (1% Triton-x100, 1 mM EDTA, 150 mM NaCl, 20 mM Tris pH 7.5) supplemented with 1x cOmplete™ EDTA-free Protease Inhibitor Cocktail (11836170001, Roche). To visualise tetherin monomer and tetherin dimers (Fig 4B), lysates were mixed with Laemmli sample buffer and run in non-reducing conditions as previously described (Giese & Marsh, 2014). For all other blots, lysates were mixed with 4× NuPage LDS sample buffer (ThermoFisher). Gels were loaded to NuPage 4–12% Bis-Tris precast gels (ThermoFisher) and transferred to PVDF membranes before being blocked using 5% milk/PBS/0.1% Tween. Primary antibodies and secondary antibodies were diluted in PBS-tween. Blots were imaged using an Odyssey CLx (Li-Cor).

Western blots were quantified by densitometry using ImageJ software. Background readings were measured and subtracted and values normalised to protein loading. Values were normalised to mock or control conditions and mean and standard deviation calculated. Graphs and statistical analysis were generated using GraphPad Prism 9 software.

## Virus growth curves

Subconfluent 6 well plates of WT HeLa + ACE2 and Bst2KO HeLa + ACE2 cells were each infected with SARS-CoV-2 (BetaCoV/Australia/VIC01/2020), at an MOI of 0.01, 1 or 5. After 1 h of infection, cells were washed with PBS and maintained in growth media until harvested. Harvests were conducted at 30 min (0 h), 24 and 48 h post infection. At each time point, the supernatant (containing released virions) was collected, clarified and stored at −80°C. The cell monolayer was scraped into PBS and subjected to three freeze–thaw cycles to release intracellular virions. Following clarification, the cell debris was discarded, and the remaining supernatant was stored at −80°C. The infectious titre of all virus samples was determined by plaque assay. Three independent biological replicates, using different virus stocks, were conducted. Statistical significance was determined using multiple $t$ tests and the Holm–Sidak method ($\alpha$ = 0.05). Each time point was analysed individually, without assuming a consistent SD. Analysis was conducted using Prism 9 (GraphPad). The percentage of virus released into the supernatant was calculated (released/released + intracellular), taking into account volumes and dilution factors. Statistical significance of released percentages was determined by a ratio paired $t$-test conducted in Prism 9 (GraphPad).

## Plaque assays

Plaque assays were performed as previously described for SARS-CoV-1, with minor amendments (van den Worm et al, 2012; Ogando

et al, 2020). Briefly, subconfluent Vero cells in 6-well plates were infected with serial dilutions of the virus sample, diluted in sera-free media, for 1 h with constant rocking. After removal of the inocula and washing with PBS, 3 ml of 0.2% agarose in virus growth media was overlaid and the cells were incubated at 37°C for 48 h. At this time the overlay media was removed, cells were washed with PBS and fixed with 10% formalin, before being stained with toluidine blue. Plaques were counted manually.

## Flow cytometry

Cells were gently trypsinised, and surface stained for flow cytometry in PBS with 0.5% BSA + 1 mM EDTA (FACS buffer) for 30 min on ice. For intracellular staining, surface stained cells were fixed and stained with Foxp3/Transcription Factor Staining Buffer Set (eBioscience) according to manufacturer's protocol. Samples were acquired on a four laser Cytoflex S (Beckman coulter, 488 nm, 640 nm, 561 nm, 405 nm).

## Flow cytometry method to assess ACE2 binding

To generate soluble ACE2-Fc, approximately 5 million HEK-293T cells were transfected with 10 µg of pcDNA3-sACE2(WT)-Fc(IgG1) complexed to 50 µg of PEI. 72 h post transfection the tissue culture supernatant was collected from the cells and any debris removed via centrifugation. The media was then aliquoted and frozen at −20°C.

To determine the cell surface levels of the recombinant spike constructs, approximately 200,000 HEK-293T cells were transfected with 1 µg of plasmid DNA complexed to 3 µl of FuGENE HD (Promega, Cat: E2311). 16–20 h post transfection the cells were dissociated from the plastic using cell dissociation buffer (Gibco Cat: 13151014) and resuspend in complete media. The cells were incubated with a SARS-CoV-2 spike antibody (GeneTex Cat: GTX632604; 1:400 diluted in media) for 1 h on ice. Unbound antibody was removed by washing the cells with ice cold media and the bound antibody detected using a goat anti-mouse secondary antibody conjugated to Cy5 (Jakson ImmunoResearch Cat: 115-175-146; diluted 1:500 in media) for 1 h on ice. The cells were then washed three times with ice cold media and their fluorescent intensity measured using a BD FACSCalibur flow cytometer. Live cells were gated using forward/side scatter and approximately 10,000 events collected per sample.

To measure the ability of the recombinant spike constructs to bind ACE2, transfected cells (see previous section) were incubated with tissue culture supernatant containing ACE2-Fc for 1 h on ice. Unbound ACE2-Fc was removed by washing the cells with ice cold media and the bound ACE2 detected using goat anti-human secondary antibody conjugated to Alexa647 (ThermoFisher Scientific Cat: A21445; diluted in media at 1:500) for 30 min on ice. The cells were then washed three times with ice cold media and their fluorescent intensity measured as outlined previously. To determine the specificity of the ACE2-Fc binding non-transfected cells were also incubated with the ACE2-Fc and the anti-human secondary antibody.

## VLP assays

HEK293T cells were seeded to 9 cm dishes. The plasmid constructs were co-transfected with TransIT-293 (Mirus Bio) and 5 µg of each

plasmid (pcDNA3.1 N-FLAG, pcDNA3.1 M-FLAG, pcDNA3.1 E-FLAG, pQCXIH Bst2-HA, pQCXIH Bst2-C3A-HA, pcDNA3.1 Spike [codon optimised], pLVX ORF3a-Strep). SARS-CoV-2 VLPs were harvested 48 h post transfection. VLPs were purified from culture supernatants by centrifugation at 500 *g*, 10 min, followed by a second centrifugation at 2,000 *g* for 20 min. Collected supernatants were filtered through 0.45 μm filters before filtrates were layered on top of 20% sucrose cushions, and then centrifuged at 100,000 *g* for 3 h. The final pellets were resuspended in pre-chilled lysis buffer.

# Data availability

The datasets produced in this study are available in the following databases: Confocal microscopy data from this publication have been deposited to the BioImage Archive database, https://www.ebi.ac.uk/biostudies and assigned the identifier S-BIAD687 (https://www.ebi.ac.uk/biostudies/bioimages/studies/S-BIAD687).

**Expanded View** for this article is available online.

## Acknowledgements

We wish to thank the microscopy and flow cytometry facilities at the Department of Pathology, University of Cambridge and the electron microscopy facility at Cambridge Institute for Medical Research, University of Cambridge. We also thank Ranjit K. Rai and Paul Griffin for assistance with HAE cell culture and Anand Shah for help to gain ethical approval to acquire HAE cells for this study. Viral infection of HAE cells was done in the Wendy S. Barclay lab (Imperial College London) supported by G2P-UK National Virology Consortium funded by UKRI (to Jonathan C. Brown and Wendy S. Barclay). We wish to thank Melbourne Health and Public Health England for nCoV/Victoria/1/2020 Coronavirus strain. JRE and RP are supported by a Sir Henry Dale Fellowship jointly funded by the Wellcome Trust and the Royal Society (216370/Z/19/Z). HS and AEF are supported by Wellcome Trust (106207; 220814) and European Research Council (646891) grants. KHJ's PhD was supported by the Wellcome Trust (200925/Z/16/Z). NMcG is supported by a Sir Henry Dale Fellowship jointly funded by the Wellcome Trust and the Royal Society (204464/Z/16/Z). HKJ is funded by Exosis Inc, Florida, USA. GWC and JLH are funded by Innovate UK. AAP and OSM are supported by the BBSRC (BB/S009566/1).

## Author contributions

**Hazel Stewart:** Resources; data curation; formal analysis; supervision; validation; investigation; visualization; methodology; writing – original draft; project administration; writing – review and editing. **Roberta Palmulli:** Resources; formal analysis; validation; investigation; methodology. **Kristoffer H Johansen:** Data curation; formal analysis; investigation. **Naomi McGovern:** Data curation; formal analysis; investigation. **Ola M Shehata:** Formal analysis; investigation. **George W Carnell:** Formal analysis; investigation; project administration. **Hannah K Jackson:** Formal analysis; investigation. **Jin S Lee:** Formal analysis; investigation. **Jonathan C Brown:** Supervision; validation; investigation; methodology; project administration. **Thomas Burgoyne:** Resources; formal analysis; validation; investigation; methodology; project administration; writing – review and editing. **Jonathan L Heeney:** Supervision; investigation; project administration. **Klaus Okkenhaug:** Supervision; writing – review and editing. **Andrew E Firth:** Formal analysis; supervision; validation; project administration; writing – review and editing. **Andrew A Peden:** Supervision; validation; investigation; methodology; writing – review and editing. **James R Edgar:** Conceptualization; resources; data curation; formal analysis; supervision; funding acquisition; validation; investigation; visualization; methodology; writing – original draft; project administration; writing – review and editing.

## Disclosure and competing interests statement

The authors declare that they have no conflict of interest.

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
