## [Review Process File · EMBO Reports]

Tetherin antagonism by SARS-CoV-2 ORF3a and Spike protein enhances virus release

Hazel Stewart, Roberta Palmulli, Kristoffer H. Johansen, Naomi McGovern, Ola M. Shehata, George W. Carnell, Hannah K. Jackson, Jin S. Lee, Jonathan C. Brown, Thomas Burgoyne, Jonathan L. Heeney, Klaus Okkenhaug, Andrew E. Firth, Andrew A. Peden and James R. Edgar

DOI: [10.15252/embr.202357224](https://doi.org/10.15252/embr.202357224)

Corresponding author(s): James R. Edgar (je333@cam.ac.uk)

Review Timeline:

Transfer from Review Commons:	21st Mar 23
Editorial Decision:	30th Mar 23
Revision Received:	19th May 23
Editorial Decision:	18th Jul 23
Revision Received:	23rd Aug 23
Accepted:	21st Sep 23

Review
COMMONS

Transaction Report: This manuscript was transferred to **EMBO reports** following peer review at Review Commons.

Review #1

1. Evidence, reproducibility and clarity:

Evidence, reproducibility and clarity (Required)

I summarise the major findings of the work below. In my opinion the range and application of approaches has provided a broad evidence base that, in general, supports the authors conclusions. However, there are, in my opinion, particular failures to utilise and communicate this evidence. The manuscript may be much improved with attention in the following areas. In each case I will give general criticism with a few examples, but the principals of my comments could be applied throughout the work.

1. Insufficient quantification.

The investigation combines various sources of qualitative data (EM, fluorescence microscopy, western blotting) to generate a reasonably strong evidence base. However, the work is over-reliant on representative images and should include more quantification from repeat experiments. When there are multiple fluorescence micrographs with intensity changes (not necessarily just representative images) (e.g. Figure 1 or 2) the authors should consider making measurements of these. Also the VLP production assays, which are assessed by western blotting would particularly benefit from a quantitative assessment (either by densitometry or, if samples remain, ELISA/similar approach).

2. Insufficient explanation.

I found some of the images and legends contained insufficient annotation and/or description for a non-expert reader to appreciate the result(s). Particularly if the authors want to draw attention to features in micrographs they should consider using more enlarged/inset images and annotations (e.g. arrows) to point out structures (e.g DMVs etc.). This short coming exacerbates the lack of quantification.

3. Insufficient exploration of the data.

I had a sense that some aspects of the data seem unconsidered or ignored, and the discussion lacks depth and reflection. For example the tetherin down-regulation apparent in Figures 1 and 2 is not really explained by the spike/ORF3a antagonism described later on, but this is not explicitly addressed. Also, Figure 6 suggests that ORF3a results in high levels of incorporation of tetherin in to VLPs, but I don't think this is even described(?). The discussion should also include more comparison with previous studies on the relationship between SARS-2 and tetherin.

I have no minor comments on this draft of the manuscript.

2. Significance:

Significance (Required)

Tetherin, encoded by the BST2 gene, is an antiviral restriction factor that inhibits the release of enveloped viruses by creating tethers between viral and host membranes. It also has a capacity for sensing and signalling viral infection. It is most widely understood in the context of HIV-1, however, there is evidence of restriction in a wide variety of enveloped viruses, many of which have evolved strategies for antagonising tetherin. This knowledge informs on viral interactions with the innate immune system, with implications for basic virology and translational research.

This study investigates tetherin in the context of SARS-CoV-2. The authors use a powerful collection of tools (live virus, gene knock out cells, recombinant viral and host expression systems) and a variety of approaches (microscopy, western blotting, infection assays), which is, itself, a strength. The study provides evidence to support a series of conclusions: I) BST2/tetherin restricts SARS-CoV-2 II) SARS-CoV-2 ablates tetherin expression III) spike protein can modestly down-regulate tetherin IV) ORF3A dysregulates tetherin localisation by altering retrograde trafficking. These conclusions are broadly supported by the data and this study make significant contributions to our understanding of SARS-CoV-2/tetherin interactions.

My enthusiasm is reduced by, in my opinion, a failure of the authors to fully quantify, explain and explore their data. I expect the manuscript could be significantly improved without further experimentation by strengthening these aspects.

This manuscript will be of interest to investigators in virology and/or cellular intrinsic immunity. Given the focus on SARS-CoV-2 it is possible/likely that it will find a slightly broader readership.

I have highly appropriate skills for evaluating this work being experienced in virology, SARS-CoV-2, cell biology and microscopy.

3. How much time do you estimate the authors will need to complete the suggested revisions:

Estimated time to Complete Revisions (Required)

(Decision Recommendation)

Between 1 and 3 months

Yes

Review #2

1. Evidence, reproducibility and clarity:

Evidence, reproducibility and clarity (Required)

BST2/tetherin can restrict the release and transmission of many enveloped viruses, including coronaviruses. In many cases, restricted viruses have developed mechanisms to abrogate tetherin-restriction by expressing proteins that antagonize tetherin; HIV-1 Vpu-mediated antagonism of tetherin restriction is a particularly well studied example. In this paper, Stewart et al. report their studies of the mechanism(s) underlying SARS-CoV-2 antagonism of tetherin restriction. They conclude that Orf3a is the primary virally encoded protein involved and that Orf3a manipulates endo-lysosomal trafficking to decrease tetherin cycling and divert the protein away from putative assembly sites.

****Major comments:****

In my view some of the claims made by the authors are not fully supported by the data. For example, the bystander effect discussed in line 162 may suggest that infected cells can produce IFN but does not 'indicate' that they do. Most of the EM images show part of a cell profile, so statements such as (line 192) 'virus containing tubulovesicular organelles were often polarised towards sites of significant surface-associated virus' should be backed up with appropriate images, or indicated as 'not shown', or removed (the observation is not so important for this story). Line 196,

DMVs can't be seen in these micrographs. Line 391, I can't see much change in CD63 distribution.

Line 321, the authors show that ORF7a does not affect tetherin localization, abundance, glycosylation or dimer formation, but they don't show that it doesn't restrict SARS-CoV-2. Can they be sure that epitope tagging this molecule does not abrogate function (or the functions of any of the other tagged proteins for that matter), or that ORF7a works in conjunction with one of the other viral proteins? In the ORF screen, a number of the constructs are expressed at low level, is it possible they are missing something?

Line 376, the authors refer to ORF3a being a viroporin. A recent eLife paper (doi: 10.7554/eLife.84477; initially published in BioRxiv) refutes this claim and builds on other evidence that ORF3a interacts with the HOPS complex. The authors should at least mention this work, especially in the discussion, as it would seem to provide a molecular mechanism to support their conclusions.

Fig 3, the growth curves illustrated in Fig3 C and D do not have errors bars; how many times were these experiments repeated?

Line 396, the authors show increased co-localization with LAMP1. As LAMP1 is found in late endosomes as well as lysosomes, they cannot claim the redistributed tetherin is specifically in lysosomes.

There seems to be a marked difference in the anti-rb555 signal in the 'mock' cells in panels 5H and Suppl 6E. Is there a good reason for this, or does this indicate variability between experiments?

Fig 6a, why is there negligible VLP release from cells lacking BST2 and ORF3a-strep? How many times were these experiments performed? Is this a representative image? I think it confusing to refer to the same protein by two different names in the same figure (i.e. BST2 and tetherin). Do the authors know how the levels of ORF3a expressed in cells in these experiments compares to those seen in infected cells?

My final point is, perhaps, the trickiest to answer, but nevertheless needs to be considered. As far as we know, SARS-CoV-2 and at least some other coronaviruses, bud into organelles of the early secretory pathway, often considered to be ERGIC. In the experiments shown here the authors provide evidence that ORF3a can influence tetherin recycling, but the main way of showing this is through its increased association with endocytic organelles. Do the authors have any evidence that Orf3a reduces tetherin levels in the ERGIC or whether the tetherin cycling pathway(s)

involve the ERGIC?

****Minor comments:****

Overall, the manuscript should be carefully edited to ensure the text reads clearly. A few examples of things that need to be fixed are:-

Line 53, delete 'shell' its redundant and confusing when the authors have said coronaviruses have a membrane.

Line 61, delete 'the'

Line 72, delete 'enveloped'; coronaviruses already described as enveloped viruses (line 53)

Lines 93 - 100, lop-sided discussion of the viral life cycle; this paragraph is mostly about entry, which is not relevant to this paper, and does not really deal with the synthesis and assembly side of the cycle.

Line 103, why are the neighbouring cells 'naive'?

Line 112 - 113, delete last phrase; tetherin is described as an IFN stimulated gene in line 111; to be accurate, the beginning of the sentence should be 'Tetherin is expressed from a type 1 Interferon stimulated gene ...'

Line 118 - 119, should say 'For tetherin-restricted enveloped viruses' as not all enveloped viruses are restricted by tetherin.

Line 131, coronaviruses are not the only family of tetherin-restricted viruses that assemble on intracellular membranes, e.g. bunyaviruses.

Line 192, there is no EM data in Supplemental Fig 1C.

Line 251, 'a synchronous infection event' should be 'synchronous infection' as there will be multiple infection events

Page 13 (and elsewhere), unlike Southern, 'Western' should not have a capital letter, except at the start of a sentence.

Lines 330 and 352, can the authors quantitate S protein-induced reduction in cell surface tetherin rather than using the somewhat subjective 'mild'?

Line 379, OFR, should be ORF.

Line 448, 'Tetherin retains the ability' - did it ever lose it?

Line 451, 'luminal' is confusing in this context.

Line 453, the process of virus envelopment is likely to be more than a 'single step'

Line 457, in my view the notion that Vpu abrogation of tetherin restriction is just due to redistribution of tetherin to the TGN is somewhat simplistic and disregards a lot of other work.

Line 472, what is meant by 'resting states'?

Line 1204, how were 'mock infected cells infected'?

2. Significance:

Significance (Required)

This study builds on published work supporting the notion that SARS-Cov-2 ORF3a is an antagonist for the restriction factor tetherin. Importantly, it provides insights to the mechanism of ORF3a mediated tetherin antagonism, specifically to ORF3a inhibits tetherin cycling, diverting the protein to lysosomes and away from compartment(s) where virions assemble. Overall, the authors provide good supporting evidence for these conclusions, however there are issues that the authors need to address.

3. How much time do you estimate the authors will need to complete the suggested revisions:

Estimated time to Complete Revisions (Required)

(Decision Recommendation)

Less than 1 month

4. Review Commons values the work of reviewers and encourages them to get credit for their work. Select 'Yes'

below to register your reviewing activity at Web of Science Reviewer Recognition Service (formerly Publons); note that the content of your review will not be visible on Web of Science.

Yes

Review #3

1. Evidence, reproducibility and clarity:

Evidence, reproducibility and clarity (Required)

Restriction factors are major barriers against viral infections. A prime example is Tetherin (aka BST2), which is able to physically tether budding virions to the plasma membrane preventing release of the infectious particles. Of note, tetherin has broad anti-viral activity and has been established as a crucial innate immune defense factor against HIV, IAV, SARS-CoV-2 and other important human pathogens. However, successful viruses like SARS-CoV-2 evolved strategies to counteract restriction factors and promote their replication. Important restriction factors, such as tetherin, may often be targeted by multiple viral strategies to ensure complete suppression of their anti-viral activities by the pathogen. Of note, it was previously published that the accessory protein ORF7a of SARS-CoV-2 binds to (Petrosino et al, Chemistry Europe, 2021) and antagonizes it (Martin-Sancho et al, Molecular Cell, 2021). Previous data on SARS-CoV also revealed that ORF7a promotes cleavage of tetherin (Taylor et al, 2015, J Virol).

In this manuscript, the authors show that tetherin restricts SARS-CoV-2 by tethering virions to the plasma membrane and propose that tetherin is targeted by two proteins of SARS-CoV-2. Whereas the Spike protein promotes degradation of tetherin, the accessory protein ORF3a redirects tetherin away from newly forming SARS-CoV-2 virions.

While the overall findings that both S and ORF3a are additionally targeting tetherin is

both novel and intriguing, additional evidence is needed to support this. In addition, the authors show that in their experimental setups ORF7a does not induce cleavage of tetherin. This is in direct contrast to previously published data both on SARS-CoV(-1) and -2 (Taylor et al, 2015, J Virol; Petrosino et al, Chemistry Europe, 2021; Martin-Sancho et al, Molecular Cell, 2021). From my point of view that needs further experimental confirmation. While the authors state that the impact of Spike on tetherin is mild, the experiments should still allow the conclusion whether there is a (mild) effect or not. The mechanism of ORF3a is fortunately more robustly assessed and provides some novel insights.

Unfortunately, the whole manuscript suffers from a striking lack of quantifications. In addition, it is not clear whether and how many times experiments were repeated to the same results. Overall, the data in this manuscript seem very speculative and preliminary and thus do not support the authors conclusions.

****Major****

Much of the data seems like it was only done once. As I am sure that this is a writing issue, please clearly state how many times the individual assays were repeated, provide the quantification graphs and appropriate statistics. Some experiments may need additional quantification and confirmation by other methods to be convincing. For example, Figure 1A, C and D: Please quantify the levels of tetherin and use an alternative readout, e.g. Western blotting of infected cells. Figure 2A: Please quantify. Fig 3A: Please show and confirm successful tetherin KO in the cell lines that are used not only in microscopy. Figure 4C: Please quantify. Figure 4D: Please quantify the effects are not obvious from the images provided. Figure 4E,F Please provide a quantification of multiple independent repeats, the claimed differences are neither striking nor obvious. Figure 5A: Please quantify.

Figure 3C and D: At timepoint 0 the infection input levels are different. The initial infection levels have to be the same to draw the conclusion that tetherin KO affects virion release and not the initial infection efficiency. Can the authors either normalize or ensure that the initial infection is the same in all conditions and that variations in the initial infection efficiency do not correlated with the impact of tetherin on replication/release ? How often were those experiments repeated? Are the marginal differences in infectious titre significant? Overall the impact of tetherin on SARS-CoV-2 is very underwhelming but that may be due to efficient viral tetherin-counteraction strategies. Why is the phenotype inverted at 72 h?

Figure 4B and C: Can the authors provide an explanation why SARS-CoV ORF7a is not inducing cleavage/removes glycosylation of tetherin. To show that the assays work, an independent positive control needs to be included. The FACS data in C is unfortunately not quantified.

Fig 4G: The rationale and result of this experiment are not clear.

Fig 6: What is the benefit of doing the VLP assays as opposed to genuine virus experiments? To me it rather seems to be making the data unnecessarily complex. Again, no quantifications or repeats are provided.

****Minor****

Fig 1D: How do the authors explain the mainly intracellular Spike staining?

Please add statistical analyses on the data e.g. Fig. 3 C and D

Fig. 4B and F: Why do the annotated sizes of tetherin differ between the blots?

Fig. 5A: What is ORF6a? Do the authors mean ORF6?

An MOI of 1 is NOT considered a low or relevant MOI. Can the authors either rephrase or repeat experiments with an actual low or relevant MOI i.e. 0.01 ?

Why were the cell models switched between Figure 1 and 2 and essentially the same experiments repeated?

The manuscript may benefit a lot from streamlining and removing unessential deviations from the main message (e.g. discussions why multistep/single step growth curves are used/not relevant; why are they shown if the authors conclude that a single step is not relevant?). The discussion is extremely lengthy and does not provide sufficient discussion of the presented data.

According to my opinion, the current manuscript does not provide significant advancement for the field. While the intention was to update and expand our existing knowledge about tetherin restriction by SARS-CoV-2, the experiments do not support this yet. However, the general premise and approach/concept of the manuscript would be appealing to a broader audience. I especially like the notion that multiple proteins of SARS-CoV-2 could synergistically counteract an important innate immune defense factor, tetherin.

My expertise is on SARS-CoV-2 and the interplay between the virus and host cell restriction factors.

2. Significance:

Significance (Required)

According to my opinion, the current manuscript does not provide significant advancement for the field. While the intention was to update and expand our existing knowledge about tetherin restriction by SARS-CoV-2, the experiments do not support this yet. However, the general premise and approach/concept of the manuscript would be appealing to a broader audience. I especially like the notion that multiple proteins of SARS-CoV-2 could synergistically counteract an important innate immune defense factor, tetherin.

My expertise is on SARS-CoV-2 and the interplay between the virus and host cell restriction factors.

3. How much time do you estimate the authors will need to complete the suggested revisions:

Estimated time to Complete Revisions (Required)

(Decision Recommendation)

More than 6 months

Yes

Revision Plan

Manuscript number: RC-2022-01808

Corresponding author(s): James Edgar

1. General Statements

We are delighted to have received such thorough reviews of our manuscript. The major aim of this study was to determine whether SARS-CoV-2 antagonises the anti-viral restriction factor tetherin, and, if so, to determine the mechanism by which this occurs.

Our data demonstrate that SARS-CoV-2 antagonises tetherin and that loss of tetherin enhances SARS-CoV-2 spread. We find that ORF3a impairs retrograde traffic, and consequently tetherin is depleted from the biosynthetic pathway.

The reviewers agree with our findings and conclusions;

“The study provides evidence to support a series of conclusions: I) BST2/tetherin restricts SARS-CoV-2 II) SARS-CoV-2 ablates tetherin expression III) spike protein can modestly down-regulate tetherin IV) ORF3A dysregulates tetherin localisation by altering retrograde trafficking. These conclusions are broadly supported by the data and this study make significant contributions to our understanding of SARS-CoV-2/tetherin interactions.” – Reviewer #1.

“Importantly, it provides insights to the mechanism of ORF3a mediated tetherin antagonism, specifically to ORF3a inhibits tetherin cycling, diverting the protein to lysosomes and away from compartment(s) where virions assemble.” – Reviewer #2.

The reviewers have commented that the study would be improved by quantification of various immunofluorescence, western blot and VLP experiments. These analysis have been performed using existing data and are now included in the revised manuscript. A few figures have been modified, and extra annotation has been added to a number of figures. We have also modified parts of the text in response to suggestions from reviewers. All reviewer comments are addressed in the following sections.

2. Description of the planned revisions

We wish to address the point which is raised by both Reviewer #2 and #3, that infection and plaque assays need to be repeated in order to add statistics to Figures 3C and 3D.

Point-by-point response:

Reviewer #2:

Fig 3, the growth curves illustrated in Fig3 C and D do not have errors bars; how many times were these experiments repeated?

- These experiments require more repeats to include error bars. Infection and plaque assay (Figure 3C, 3D) are currently ongoing and we plan to complete them in the next 6-8 weeks. In the current experiments, infections will additionally be performed at MOI 0.01, in addition to the previous MOIs (1 and 5).

Reviewer #3:

Please add statistical analyses on the data e.g. Fig. 3 C and D

- **See above – these experiments are in progress.**

An MOI of 1 is NOT considered a low or relevant MOI. Can the authors either rephrase or repeat experiments with an actual low or relevant MOI i.e. 0.01 ?

- **See above – we are currently repeating these experiments and are including MOIs of 0.01, 1 and 5.**

Figure 3C and D: At timepoint 0 the infection input levels are different. The initial infection levels have to be the same to draw the conclusion that tetherin KO affects virion release and not the initial infection efficiency. Can the authors either normalize or ensure that the initial infection is the same in all conditions and that variations in the initial infection efficiency do not correlated with the impact of tetherin on replication/release ? How often were those experiments repeated? Are the marginal differences in infectious titre significant? Overall the impact of tetherin on SARS-CoV-2 is very underwhelming but that may be due to efficient viral tetherin-counteraction strategies. Why is the phenotype inverted at 72 h?

- **See above.**

Revision Plan

- **Equal amounts of virus, as measured by plaque-forming units (PFU), were used for both HeLa cell lines and thus at 0 hpi the variation seen is within the parameters of the assay used. It remains possible that tetherin affects virus entry but this is unlikely and this assay was not designed to investigate that effect.**
- **Growth curve assays are currently being repeated using an MOI of 0.01, 1 and 5. We are removing the 72 hpi sample from future experiments. At this time point, we find that the extensive cell death caused by viral replication (especially at higher MOIs) makes it difficult to accurately separate the released from intracellular fractions and conclusions cannot be accurately drawn from the data.**

3. Description of the revisions that have already been incorporated in the transferred manuscript

The suggestions from reviewers to strengthen this manuscript focused on increasing the quantification throughout the manuscript and to change parts of the text and figures to better explain the data. We thank the reviewers for these suggestions and have included quantification for many experiments in our revised manuscript. We have been able to use our existing data to quantify immunofluorescence, western blot and VLP experiments. The final page of this section contains a table of new quantification, and additional and modified data. We have added annotations to several figures and have replaced a few images to support or clarify points (details below). We have also made some modifications to the text (details below).

Point-by-point

Reviewer #1:

1) Insufficient quantification.

The investigation combines various sources of qualitative data (EM, fluorescence microscopy, western blotting) to generate a reasonably strong evidence base. However, the work is over-reliant on representative images and should include more quantification from repeat experiments. When there are multiple fluorescence micrographs with intensity changes (not necessarily just representative images) (e.g. Figure 1 or 2) the authors should consider making measurements of these. Also the VLP production assays, which are assessed by western blotting would particularly benefit from a quantitative assessment (either by densitometry or, if samples remain, ELISA/similar approach).

- **We have performed quantification for immunofluorescence, western blotting and VLP experiments. A simplified table is shown on the final page of this form.**
- **Quantification of the following immunofluorescence experiments are now included in our updated manuscript:**
 - **Related to Figure 1A – tetherin mean fluorescence intensity changes in Spike negative vs positive HAE cells. These data are shown in Supplemental Figure 1C.**
 - **Related to Figure 1D – tetherin mean fluorescence intensity changes in Spike negative vs positive HeLa cells. These data are shown in Supplemental Figure 1E.**

- Related to Figure 2C – tetherin mean fluorescence intensity changes in Spike negative vs positive A549 cells. These data are shown in Supplemental Figure 2B.
- Related to Figure 2G – tetherin mean fluorescence intensity changes in Spike negative vs positive T84 cells. These data are shown in Supplemental Figure 2D.
- Related to Figure 4D – tetherin mean fluorescence intensity changes +/- Dox in TetOne ss-HA-Spike HeLa cells. These data are shown in Supplemental Figure 4E.
- Related to Supplemental Figure 4F – tetherin mean fluorescence intensity changes in Spike negative vs positive in HeLa cells transiently transfected with ss-HA-Spike. These data are shown in Supplemental Figure 4B.

The mean, standard deviation, number of cells analysed and number of independent experiments are expressed in the figure legends. Statistical analysis is also detailed in figure legends. Methods for the quantification of fluorescence intensity is included in the Methods section.

- We have performed densitometry as suggested to quantify western blots and VLP production. These data are included in our revised manuscript:
 - Related to Figure 4F – tetherin abundance in lysates from TetOne ss-HA-Spike, +/- Dox. These data are shown in Supplemental Figure 4H.
 - Related to Figure 4G – tetherin abundance in lysates from Spike VLP experiments. These data are shown in Supplemental Figure 4I.
 - Related to Figure 4G – N-FLAG abundance in VLP fraction from Spike VLP experiments. These data are shown in Supplemental Figure 4J.
 - Related to Figure 6A – tetherin abundance in lysates from ORF3a VLP experiments These data are shown in Supplemental Figure 7A.
 - Related to Figure 6A – N-FLAG abundance in VLP fraction from ORF3a VLP experiments. These data are shown in Supplemental Figure 7B.

The mean, standard deviation and number of independent experiments analysed are expressed in the figure legends. Methods for densitometry quantification is now included at Line 956.

2) Insufficient explanation.

I found some of the images and legends contained insufficient annotation and/or description for a non-expert reader to appreciate the result(s). Particularly if the authors want to draw attention to features in micrographs they should consider using more enlarged/inset images and annotations (e.g. arrows) to point out structures (e.g DMVs etc.). This short coming exacerbates the lack of quantification.

- **Additional detail has been given to the figure legends, and to draw attention to features in micrographs, we have added the following;**
 - **Black arrowheads have been added to Figures 1E, 2D, 2H to highlight plasma membrane-associated virions, and asterisks to highlight DMVs in Figures 1E, 2D and Supplemental Figures 2C, 2E.**
 - **Similarly, typical Golgi cisternae are highlighted by white arrowheads in micrographs in Figure 2E.**
- **These figure legends have also been modified to highlight these additions.**

3) Insufficient exploration of the data.

I had a sense that some aspects of the data seem unconsidered or ignored, and the discussion lacks depth and reflection. For example the tetherin down-regulation apparent in Figures 1 and 2 is not really explained by the spike/ORF3a antagonism described later on, but this is not explicitly addressed.

- **We have made changes throughout the manuscript, but the Discussion especially has been modified. We now discuss the ORF3a data in more depth, discuss possible mechanisms by which ORF3a alone enhances VLP release, and discuss our ORF7a data in the context of previous reports.**
- **See discussion section of revised manuscript.**

Also, Figure 6 suggests that ORF3a results in high levels of incorporation of tetherin in to VLPs, but I don't think this is even described(?). The discussion should also include more comparison with previous studies on the relationship between SARS-2 and tetherin.

- **We have added a section to discuss how ORF3a may enhance VLP release, 'We found that the expression of ORF3a enhanced VLP independently of its ability to relocalise tetherin (Figure 6A). This may be due to either the ability of ORF3a to induce Golgi fragmentation [38] which facilitates viral trafficking [39], or due to enhanced lysosomal exocytosis [37]. Tetherin was also found in VLPs upon co-expression with ORF3a (Figure 6A) which may also indicate enhanced release via lysosomal exocytosis [37].**

The secretion of lysosomal hydrolases has been reported upon expression of ORF3a [31] and whilst this may in-part be due to enhanced lysosome-plasma membrane fusion, our data highlights that ORF3a impairs the retrograde trafficking of CIMPR (Supplemental Figures 6B, 6F, 6G), which may similarly increase hydrolase secretion.
– (Line 625-654).

- **The Discussion has been developed to compare the relationship between SARS-CoV-2 and tetherin in previous studies,**
'SARS-CoV-1 ORF7a is reported to inhibit tetherin glycosylation and localise to the plasma membrane in the presence of tetherin [18]. We did not observe any difference in total tetherin levels, tetherin glycosylation, ability to form dimers, or surface tetherin upon expression of either SARS-CoV-1 or SARS-CoV-2 ORF7a (Figures 4A, 4B, 4C). Others groups have demonstrated a role for ORF7a in sarbecovirus infection and both SARS-CoV-1 and SARS-CoV-2 virus lacking ORF7a show impaired virus replication in the presence of tetherin [18,41]. A direct interaction between SARS-CoV-1 ORF7a and SARS-CoV-2 ORF7a and tetherin have been described [18,41], although the precise mechanism(s) by which ORF7a antagonises tetherin remains enigmatic. We cannot exclude that ORF7a requires other viral proteins to antagonise tetherin, or that ORF7a antagonises tetherin via another mechanism. For example, ORF7a can potentially antagonise IFN signalling [42] which would impair tetherin induction in many cell types.'
– (Line 667-704).

Reviewer #2:

Major comments:

In my view some of the claims made by the authors are not fully supported by the data. For example, the bystander effect discussed in line 162 may suggest that infected cells can produce IFN but does not 'indicate' that they do.

- This text has now been edited,
'The levels of tetherin in uninfected HAE cells is lower than observed in uninfected neighbours in infected wells demonstrating that infected HAE cells are able to generate IFN to act upon uninfected neighbouring cells, enhancing tetherin expression.' - (Lines 163-172).

Most of the EM images show part of a cell profile, so statements such as (line 192) 'virus containing tubulovesicular organelles were often polarised towards sites of significant surface-associated virus' should be backed up with appropriate images, or indicated as 'not shown', or removed (the observation is not so important for this story). Line 196, DMVs can't be seen in these micrographs.

- The statement 'virus containing tubulovesicular organelles were often polarised towards sites of significant surface-associated virus' has been removed. The micrographs in Figure 1E have been re-cropped, and image iii replaced with an image showing DMVs

Revision Plan

and budding virions. Plasma membrane-associated virions are highlighted by black arrowheads, DMVs by black asterisks, and intracellular virion by a white arrow.

Line 391, I can't see much change in CD63 distribution.

- CD63 reproducibly appears clustered towards the nuclei in ORF3a expressing cells, whilst CD63 positive puncta are abundant in the periphery of mock cells. CD63 puncta are also larger, and the staining of CIMPR and VPS35 also appears to be associated with larger organelles. We have amended the text to now read, 'Expression of ORF3a also disrupted the distribution of numerous endosome-related markers including CIMPR, VPS35, CD63, which all localised to larger and less peripheral puncta (Supplemental Figure 6B), and the mixing of early and late endosomal markers' - (Line 469).
- Quantification of the diameter of CD63 puncta indicate that they are larger in ORF3a expressing cells than in mock cells. Mock cells - $0.71\mu\text{m}$ (SD; 0.19), ORF3a - $1.15\mu\text{m}$ (SD;0.35). At least 75 organelles per sample, from 10 different cells. We have not included this data as we do not wish to labor this point but are happy to include this quantification if required to do so.

Line 321, the authors show that ORF7a does not affect tetherin localization, abundance, glycosylation or dimer formation, but they don't show that it doesn't restrict SARS-CoV-2. Can they be sure that epitope tagging this molecule does not abrogate function (or the functions of any of the other tagged proteins for that matter), or that ORF7a works in conjunction with one of the other viral proteins?

- We are careful in the manuscript not to claim that ORF7a has no effect on tetherin. Our data indicate that 'ORF7a does not directly influence tetherin localisation, abundance, glycosylation or dimer formation' - (Line 361-362).
- We were unable to reproduce an effect of ORF7a on tetherin glycosylation. Our data conflicts with that presented by Taylor et al, 2015, where ORF7a impaired tetherin glycosylation and ORF7a localised to the plasma membrane in tetherin expressing cells. The experiments performed by Taylor et al used HEK293 cells and ectopically expressed tagged tetherin. The differences in results may be attributed to the differences between cell lines or due to differences between endogenous or ectopic / tagged tetherin.
- The study by Taylor et al uses SARS-CoV-1 ORF7a-HA from Kopecky-Bromberg et al., 2007 (DOI: [10.1128/JVI.01782-06](https://doi.org/10.1128/JVI.01782-06)), where the -HA tag is positioned at the C-terminus. Our ORF7a-FLAG constructs have a C-terminal epitope tag. While we cannot exclude

the possibility that tagged proteins may act differently from untagged ones, the differences between our findings and previous work appear unlikely to be due to epitope tags.

- Our manuscript states that although we cannot find any effect of ORF7a on tetherin localisation, abundance, glycosylation, or dimer formation, we cannot exclude that ORF7a impacts tetherin by another mechanism. For example, ORF7a has been found to antagonise interferon responses. Tetherin is abundantly expressed in HeLa cells and expression does not require induction through interferon. None of our experiments above would be impacted by interferon antagonism yet this could impact other cell types besides infection *in vivo*. These possibilities may explain the reported differential impact of ORF7a by different labs. An addition comment has been added to the discussion to reflect this,
'We cannot exclude that ORF7a requires other viral proteins to antagonise tetherin, or that ORF7a antagonises tetherin via another mechanism. For example, ORF7a potently antagonises IFN signalling [38], which would impair tetherin induction in many cell types.'
- (Line 701-704).

Note - Reference 38 has been added to the manuscript – Xia et al., Cell Reports
DOI: [10.1016/j.celrep.2020.108234](https://doi.org/10.1016/j.celrep.2020.108234)

In the ORF screen, a number of the constructs are expressed at low level, is it possible they [the authors] are missing something?

- Some of the ORFs expressed in the miniscreen appear poorly expressed. We accept that in the use of epitope tagged constructs expression levels of individual viral proteins may impact upon a successful screen. However, this screen was performed to identify any potential changes in tetherin abundance or localisation, and the screen did successfully identify ORF3a, which we were able to follow-up and verify.

Line 376, the authors refer to ORF3a being a viroporin. A recent eLife paper (doi: 10.7554/eLife.84477; initially published in BioRxiv) refutes this claim and builds on other evidence that ORF3a interacts with the HOPS complex. The authors should at least mention this work, especially in the discussion, as it would seem to provide a molecular mechanism to support their conclusions.

- This paper had not been peer reviewed at the time of our initial submission. We have now included the following text,
'SARS-CoV-2 ORF3a is an accessory protein that localises to and perturbs endosomes and lysosomes [29]. It may do so by acting either as a viroporin [30] or by interacting with, and possibly interfering with the function of VPS 39, a component of the HOPS complex which facilitates tethering of late endosomes or autophagosomes with lysosomes [29,31]. Given ORF3a likely impairs lysosome function, the observed increased....' - (Lines 444-449).

Fig 3, the growth curves illustrated in Fig3 C and D do not have errors bars; how many times

were these experiments repeated?

- **See comment above – these experiments are ongoing.**

Line 396, the authors show increased co-localization with LAMP1. As LAMP1 is found in late endosomes as well as lysosomes, they cannot claim the redistributed tetherin is specifically in lysosomes.

- We have altered the text to now say:
'The ORF3a-mediated increase in tetherin abundance within endolysosomes could be due to defective lysosomal degradation.' - (Line 475).

There seems to be a marked difference in the anti-rb555 signal in the 'mock' cells in panels 5H and Suppl 6E. Is there a good reason for this, or does this indicate variability between experiments?

- Antibody uptake experiments in Figure 5H and Supp Figure 6E were performed and acquired on different days. Relatively low levels of signal are available in these antibody uptake experiments, and the disperse labelling seen in the mocks does not aid this.

Fig 6a, why is there negligible VLP release from cells lacking BST2 and ORF3a-strep? How many times were these experiments performed? Is this a representative image? I think it confusing to refer to the same protein by two different names in the same figure (i.e. BST2 and tetherin). Do the authors know how the levels of ORF3a expressed in cells in these experiments compares to those seen in infected cells?

- We have changed the blot in Figure 6A for one with clearer FLAG bands. Three independent experiments were performed for Figure 6A. Quantification of VLPs is now included in Supplemental Figure 7B.
- We have changed 'Bst2' to 'tetherin' in all previous figures relating to protein; Figure 4G, Figure 6A, B, C.
- We have no current information to compare ORF3a levels in these experiments versus in infected cells. We can investigate quantifying this if necessary.

My final point is, perhaps, the trickiest to answer, but nevertheless needs to be considered. As far as we know, SARS-CoV-2 and at least some other coronaviruses, bud into organelles of the early secretory pathway, often considered to be ERGIC. In the experiments shown here the authors provide evidence that ORF3a can influence tetherin recycling, but the main way of

Revision Plan

showing this is through its increased association with endocytic organelles. Do the authors have any evidence that Orf3a reduces tetherin levels in the ERGIC or whether the tetherin cycling pathway(s) involve the ERGIC?

- This is an interesting point, and as the reviewer concedes, this is tricky to answer. Expression of ORF3a causes the redistribution or remodeling of various organelles (Figures 1E, 2D, 2F, Supp Figures 2C, 2E, 3E, 6B, 6C, 6D). We have been unable to test the direct involvement of ERGIC, despite attempts with a number of commercial antibodies. Given the huge rearrangements of organelles during SARS-CoV-2 infection, it is unclear exactly what will happen to the distribution of ERGIC.

Minor comments:

Line 53, delete 'shell' its redundant and confusing when the authors have said coronaviruses have a membrane.

- **Deleted.**

Line 61, delete 'the'

- **Deleted.**

Line 72, delete 'enveloped'; coronaviruses already described as enveloped viruses (line 53)

- **Removed.**

Lines 93 - 100, lop-sided discussion of the viral life cycle; this paragraph is mostly about entry, which is not relevant to this paper, and does not really deal with the synthesis and assembly side of the cycle.

- We have now added the following text, '...liberating the viral nucleocapsid to the cytosol of the cell. Upon uncoating, the RNA genome is released into the host cytosol and replication-transcription complexes assemble to drive the replication of the viral genome and the expression of viral proteins. Coronaviruses modify host organelles to generate viral replication factories - so-called DMVs (double-membrane vesicles) that act as hubs for viral RNA synthesis [10]. SARS-CoV-2 viral budding occurs at ER-to-Golgi intermediate compartments (ERGIC) and newly formed viral particles traffic through secretory vesicles to the plasma membrane where they are released to the extracellular space.' - (Lines 95-104).

Line 103, why are the neighbouring cells 'naive'?

- 'naïve' removed.

Line 112 - 113, delete last phrase; tetherin is described as an IFN stimulated gene in line 111; to be accurate, the beginning of the sentence should be 'Tetherin is expressed from a type 1 Interferon stimulated gene ...'

- Amended.

Line 118 - 119, should say 'For tetherin-restricted enveloped viruses' as not all enveloped viruses are restricted by tetherin.

- Amended.

Line 131, coronaviruses are not the only family of tetherin-restricted viruses that assemble on intracellular membranes, e.g. bunyaviruses.

- This has been modified and now reads, 'In order for tetherin to tether coronaviruses, tetherin must be incorporated in the virus envelope during budding which occurs in intracellular organelles.' - (Lines 133-135).

Line 192, there is no EM data in Supplemental Fig 1C.

- **This has now been removed.**

Line 251, 'a synchronous infection event' should be 'synchronous infection' as there will be multiple infection events.

- **This has been changed.**

Page 13 (and elsewhere), unlike Southern, 'Western' should not have a capital letter, except at the start of a sentence.

- Have been updated throughout the manuscript (Lines 183, 341, 3549, 356, 392, 509, 763, 1330, 1399).

Lines 330 and 352, can the authors quantitate S protein-induced reduction in cell surface tetherin rather than using the somewhat subjective 'mild'?

- These are now changed to,

Revision Plan

'Transient transfection of cells with ss-HA-Spike caused a 32% decrease in tetherin as observed by immunofluorescence (Supplemental Figure 4A, 4B), with...' – (Line 370).

'To explore whether the Spike-induced tetherin downregulation altered virus release, we performed experiments with virus like particles (VLPs) in HEK293T ...' – (Line 399).

Line 379, OFR, should be ORF.

- Yes, changed.

Line 448, 'Tetherin retains the ability' - did it ever lose it?

- This has been rephrased to, 'Tetherin has the ability to restrict a number of different enveloped viruses that bud at distinct organelles.' - (Line 547).

Line 451, 'luminal' is confusing in this context.

This has been modified to, 'Tetherin forms homodimers between opposing membranes (e.g., plasma membrane and viral envelope) that are linked via disulphide bonds.' - (Line 549).

Line 453, the process of virus envelopment is likely to be more than a 'single step'

- Removed. This now reads, '...virus during viral budding, which occurs in modified ERGIC organelles.' - (Line 552).

Line 457, in my view the notion that Vpu abrogation of tetherin restriction is just due to redistribution of tetherin to the TGN is somewhat simplistic and disregards a lot of other work.

- We have removed mention of mechanisms of tetherin antagonism by other viruses. The key point we wish to make here is that tetherin is lost from the budding compartment. This now reads, 'Many enveloped viruses antagonise tetherin by altering its localisation and removing it from the respective site of virus budding.' – (Line 552-553).

Line 472, what is meant by 'resting states'?

- This should have been 'in the absence of stimulation' and have now been re-written,

Revision Plan

'Tetherin is an IFN-stimulated gene (ISG) [13], and many cell types express low levels of tetherin in the absence of stimulation.' - (Line 577).

Line 1204, how were 'mock infected cells infected'?

- This has now been re-written,
'Differentiated nasal primary human airway epithelial (HAE) cells were embedded to OCT....' - (Line 1385).

Reviewer #3:

Much of the data seems like it was only done once. As I am sure that this is a writing issue, please clearly state how many times the individual assays were repeated, provide the quantification graphs and appropriate statistics. Some experiments may need additional quantification and confirmation by other methods to be convincing.

- **Quantification is provided throughout the revised manuscript. Figure legends have also been updated to provide information on quantification and statistical analysis.**

For example, Figure 1A, C and D: Please quantify the levels of tetherin and use an alternative readout, e.g. Western blotting of infected cells.

- **Quantification has been performed and included in our revised manuscript in Supplemental Figures 1C, 1E. Tetherin is not shown in Figure 1C. A table is provided on the last page of this document to highlight the additional quantification.**

Figure 2A: Please quantify.

- **We are not sure we understand this point. The western blot shown in Figure 2A demonstrates the ectopic expression of ACE2 in our A549 cell line. A549 cells have been used by many labs to study SARS-CoV-2 infection, but express negligible ACE2.**

Fig 3A: Please show and confirm successful tetherin KO in the cell lines that are used not only in microscopy.

- **A new blot is now shown in Figure 3A, including a blot demonstrating tetherin loss in both KO lines.**

Figure 4D: Please quantify the effects are not obvious from the images provided.

- **Quantification is now provided in Supplemental Figure 4E.**

Figure 4E, F Please provide a quantification of multiple independent repeats, the claimed differences are neither striking nor obvious.

- **Quantification of 4F is now provided in Supplemental Figure 4G. Tetherin levels were quantified to be reduced by 25% (SD: 8%) by addition of Doxycycline and induction of ss-HA-Spike.**
- **Information for quantification is provided in figure legends.**

Figure 4B and C: Can the authors provide an explanation why SARS-CoV ORF7a is not inducing cleavage/removes glycosylation of tetherin. To show that the assays work, an independent positive control needs to be included.

- **See above comments regarding discussion on ORF7a. Additional text has been included to discuss ORF7a data,**
'We cannot exclude that ORF7a tetherin requires other viral proteins to antagonise tetherin, or that ORF7a antagonises tetherin via another mechanism. For example, ORF7a potentially antagonises IFN signalling [40], which would impair tetherin induction in many cell types.' - (Line 701-704).

Fig 4G: The rationale and result of this experiment are not clear.

- **The rationale for Spike VLP experiments is explained at Line 403. Given that Spike caused a reproducible decrease in cellular tetherin, we examined whether this downregulation was sufficient to antagonise tetherin and increase VLP yield.**

Fig 6: What is the benefit of doing the VLP assays as opposed to genuine virus experiments? To me it rather seems to be making the data unnecessarily complex. Again, no quantifications or repeats are provided.

- **VLPs are used to separate the budding and release process from the replication process of RNA viruses. VLPs have been used in a number of SARS-CoV (DOI: [10.1002/jmv.25518](https://doi.org/10.1002/jmv.25518)) and HIV-1 (DOI: <https://doi.org/10.1186/1742-4690-7-51>) studies to analyse the impact of tetherin (and tetherin mutants) on release.**

- **VLP experiment quantification are now included throughout.**

Minor:

Fig 1D: How do the authors explain the mainly intracellular Spike staining?

- **We do not understand this point. Spike staining is intracellular, whether expressed alone or in the context of infected cells.**

Fig. 4B and F: Why do the annotated sizes of tetherin differ between the blots?

- **Figures 4B and 4F are run in non-reduced and reduced conditions respectively. In order to best show the dimer deficient C3A-Tetherin, blots are typically run in non-reduced conditions to exemplify dimer formation and to highlight any defects in dimer formation. The rest of the blots in the manuscript are run in denaturing conditions to aid blotting of other proteins. (Lines 957-958) and now (Lines 1356-1357).**

Fig. 5A: What is ORF6a? Do the authors mean ORF6?

- **Yes, this has been changed.**

Why were the cell models switched between Figure 1 and 2 and essentially the same experiments repeated?

- **HeLa cells express high levels of tetherin at steady state, whilst A549 cells require IFN stimulation. HeLa cells demonstrate that tetherin downregulation occurs via an IFN-independent manner. A549 and T84 cells are more physiologically relevant cell types for SARS-CoV-2 infection. These points are stated in Lines 230 and 261.**

The manuscript may benefit a lot from streamlining and removing unessential deviations from the main message (e.g. discussions why multistep/single step growth curves are used/not relevant; why are they shown if the authors conclude that a single step is not relevant?). The discussion is extremely lengthy and does not provide sufficient discussion of the presented data.

- **The multistep/single step growth curve text will be adapted, but it will be re-written after additional infection experiments.**

Revision Plan

- **We have removed from the Discussion a small section discussing ORF7a mutants, given that the emphasis of our manuscript is not on ORF7a.**
- **We have also removed a small section describing the rearrangements of intracellular organelles by SARS-CoV-2 as it does not directly relate to the central message of our manuscript.**

Summary of new data included in revised manuscript

Quantification of existing data:

Data shown	Quantification now shown in	Method	Analysis
Figure 1A	Supp F1C	IF	HAE (-/+ SARS-CoV-2) - Tetherin total fluorescence intensity
Figure 1D	Supp F1E	IF	HeLa+ACE2 (-/+ SARS-CoV-2) - Tetherin total fluorescence intensity
Figure 2C	Supp F2B	IF	A549+ACE2 (-/+ SARS-CoV-2)- Tetherin total fluorescence intensity
Figure 2G	Supp F2D	IF	T84 (-/+ SARS-CoV-2) - Tetherin total fluorescence intensity
Supp F4A	Supp F4B	IF	HeLa + ss-HA-Spike transients (-/+ HA stained cells) - Tetherin total fluorescence intensity
Figure 4D	Supp F4E	IF	HeLa + TetOne ss-HA-Spike stables (-/+ Dox) - Tetherin total fluorescence intensity
Figure 4F	Supp F4G	W blot	HeLa + TetOne ss-HA-Spike stables (-/+ Dox) – Tetherin abundance
Figure 4G	Supp F4I	W blot – lysates	Spike VLP experiments – tetherin abundance
Figure 4G	Supp F4J	W blot - VLPs	Spike VLP experiments - N-FLAG abundance
Figure 6A	Supp F7A	W blot – lysates	ORF3a VLP experiments – tetherin abundance
Figure 6A	Supp F7B	W blot - VLPs	ORF3a VLP experiments - N-FLAG

New figures:

* Supplemental Figure 2C: More TEM images provided of SARS-CoV-2 infected A549+ACE2 cells. Arrowheads and asterisks highlighting virions at the cell surface and DMVs respectively.

* Figure 3A: Western blot updated to include tetherin for WT/Bst2KO HeLa -/+ ACE2 cell lines.

Updated / modified data:

* Figure 1E: Micrograph (i) has been uncropped to show DMV. Micrograph in (iii) has been changed to show clear DMVs and budding virus.

* Figure 2D: Micrographs (ii) and (iii) have been added to replace previous images which were blowups from part (i).

Revision Plan

* Additional annotation added to Figures 1E, 2D, 2E, 2H, Supp F2C, Supp F2E.

4. Description of analyses that authors prefer not to carry out

Currently flow cytometry experiments have been performed twice each and this is now detailed in the figure legends. The data shown in each panel is representative and the data has been explored using analogous approaches. For example, Figure 4C is complemented by Figures 4A and 4B, Figures 4E is complemented by 4D and 4F. We do not feel that repeating these flow cytometry analysis will significantly improve the manuscript.

Reviewer #3:

Figure 4E, F Please provide a quantification of multiple independent repeats, the claimed differences are neither striking nor obvious.

Figure 5A: Please quantify.

The FACS data in (Figure 4C) is unfortunately not quantified.

Dear Dr. Edgar,

Thank you for the transfer of your research manuscript to EMBO reports. As discussed, we would like to invite you to revise your study along the lines suggested in your revision plan.

Please address all referee concerns in a complete point-by-point response. Acceptance of the manuscript will depend on a positive outcome of the re-review round. It is EMBO reports policy to allow a single round of revision only and acceptance or rejection of the manuscript will therefore depend on the completeness of your responses included in the next, final version of the manuscript.

You indicated in your letter that you would need another 6 - 8 weeks for the revision. I have now entered our 'standard' revision duration of 3 months in the system. Please discuss the revision progress ahead of this time with me, in case it turns out that you require more time to complete the revisions.

I am also happy to discuss the revision further via e-mail or a video call, if you wish.

Below I will list further information on how to format your manuscript and on editorial policies that I ask you to consider when you submit the revised manuscript. Please also note that we require a statement on the safety of work with SARS-CoV-2 (BSL3).

Kind regards,

Martina Rembold

Further information:

*******IMPORTANT NOTE:**

We perform an initial quality control of all revised manuscripts before re-review. Your manuscript will FAIL this control and the handling will be DELAYED IF the following APPLIES:

- 1) A data availability section providing access to data deposited in public databases is missing. If you have not deposited any data, please add a sentence to the data availability section that explains that.
- 2) Your manuscript contains statistics and error bars based on $n=2$. Please use scatter blots in these cases. No statistics should be calculated if $n=2$.

When submitting your revised manuscript, please carefully review the instructions that follow below. Failure to include requested items will delay the evaluation of your revision. *****

- 1) a .docx formatted version of the manuscript text (including legends for main figures, EV figures and tables). Please make sure that the changes are highlighted to be clearly visible.
- 2) individual production quality figure files as .eps, .tif, .jpg (one file per figure).
Please download our Figure Preparation Guidelines (figure preparation pdf) from our Author Guidelines pages <https://www.embopress.org/page/journal/14693178/authorguide> for more info on how to prepare your figures.
- 3) a .docx formatted letter INCLUDING the reviewers' reports and your detailed point-by-point responses to their comments. As part of the EMBO Press transparent editorial process, the point-by-point response is part of the Review Process File (RPF), which will be published alongside your paper.
- 4) a complete author checklist, which you can download from our author guidelines (). Please insert information in the checklist that is also reflected in the manuscript. The completed author checklist will also be part of the RPF.
- 5) Please note that all corresponding authors are required to supply an ORCID ID for their name upon submission of a revised manuscript (). Please find instructions on how to link your ORCID ID to your account in our manuscript tracking system in our Author guidelines
()
- 6) We replaced Supplementary Information with Expanded View (EV) Figures and Tables that are collapsible/expandable online.

A maximum of 5 EV Figures can be typeset. EV Figures should be cited as 'Figure EV1, Figure EV2' etc... in the text and their respective legends should be included in the main text after the legends of regular figures.

7) Please note that a Data Availability section at the end of Materials and Methods is now mandatory. In case you have no data that requires deposition in a public database, please state so instead of refereeing to the database. See also < <https://www.embopress.org/page/journal/14693178/authorguide#dataavailability>>. Please note that the Data Availability Section is restricted to new primary data that are part of this study.

Additional information on source data and instruction on how to label the files are available .

10) Figure legends and data quantification:

- the name of the statistical test used to generate error bars and P values,
 - the number (n) of independent experiments (please specify technical or biological replicates) underlying each data point,
 - the nature of the bars and error bars (s.d., s.e.m.)
- If the data are obtained from n {less than or equal to} 5, show the individual data points in addition to the SD or SEM.
- If the data are obtained from n {less than or equal to} 2, use scatter blots showing the individual data points.

11) Our journal encourages inclusion of *data citations in the reference list* to directly cite datasets that were re-used and obtained from public databases. Data citations in the article text are distinct from normal bibliographical citations and should directly link to the database records from which the data can be accessed. In the main text, data citations are formatted as follows: "Data ref: Smith et al, 2001" or "Data ref: NCBI Sequence Read Archive PRJNA342805, 2017". In the Reference list, data citations must be labeled with "[DATASET]". A data reference must provide the database name, accession number/identifiers and a resolvable link to the landing page from which the data can be accessed at the end of the reference. Further instructions are available at .

12) As part of the EMBO publication's Transparent Editorial Process, EMBO reports publishes online a Review Process File to accompany accepted manuscripts. This File will be published in conjunction with your paper and will include the referee reports, your point-by-point response and all pertinent correspondence relating to the manuscript.

Yours sincerely,

Response to Reviewers

Author responses are given in indented bold text.
Major text edits are shown, and the Line numbers given.

Reviewer #1 (Evidence, reproducibility and clarity (Required)):

I summarise the major findings of the work below. In my opinion the range and application of approaches has provided a broad evidence base that, in general, supports the authors conclusions. However, there are, in my opinion, particular failures to utilise and communicate this evidence. The manuscript may be much improved with attention in the following areas. In each case I will give general criticism with a few examples, but the principals of my comments could be applied throughout the work.

1) Insufficient quantification.

The investigation combines various sources of qualitative data (EM, fluorescence microscopy, western blotting) to generate a reasonably strong evidence base. However, the work is over-reliant on representative images and should include more quantification from repeat experiments. When there are multiple fluorescence micrographs with intensity changes (not necessarily just representative images) (e.g. Figure 1 or 2) the authors should consider making measurements of these. Also the VLP production assays, which are assessed by western blotting would particularly benefit from a quantitative assessment (either by densitometry or, if samples remain, ELISA/similar approach).

- **We have performed quantification for immunofluorescence, western blotting and VLP experiments. A few additional immunofluorescence experiments were performed to allow three biological replicates to be analysed. A simplified table showing new quantification, new data or amended data is also shown on the final page of this document.**
- **Quantification of the following immunofluorescence experiments are now included in our updated manuscript:**
 - **Related to Figure 1A – tetherin mean fluorescence intensity changes in Spike negative vs positive HAE cells. These data are shown in Expanded View Figure 1C.**
 - **Related to Figure 1D – tetherin mean fluorescence intensity changes in Spike negative vs positive HeLa cells. These data are shown in Expanded View Figure 1E.**

- Related to Figure 2C – tetherin mean fluorescence intensity changes in Spike negative vs positive A549 cells. These data are shown in Expanded View 1H.
- Related to Figure 2G – tetherin mean fluorescence intensity changes in Spike negative vs positive T84 cells. These data are shown in Expanded View Figure 1J.
- Related to Figure 4D – tetherin mean fluorescence intensity changes +/- Dox in TetOne ss-HA-Spike HeLa cells. These data are shown in Expanded View Figure 34.
- Related to Expanded View Figure 4A – tetherin mean fluorescence intensity changes in Spike negative vs positive in HeLa cells transiently transfected with ss-HA-Spike. These data are shown in Expanded View Figure 4B.

The mean, standard deviation, number of cells analysed and number of independent experiments are expressed in the figure legends. Statistical analysis is also detailed in figure legends. Methods for the quantification of fluorescence intensity is included in the Methods section.

The confocal images used for the above analyses have now been uploaded to the BiImage Archive database, <https://www.ebi.ac.uk/biostudies> and assigned the identifier S-BIAD687.

- We have performed densitometry as suggested to quantify western blots and VLP production. These data are included in our revised manuscript:
 - Related to Figure 4F – tetherin abundance in lysates from TetOne ss-HA-Spike, +/- Dox. These data are shown in Expanded View Figure 4G.
 - Related to Figure 4G – tetherin abundance in lysates from Spike VLP experiments. These data are shown in Expanded View 4I.
 - Related to Figure 4G – N-FLAG abundance in VLP fraction from Spike VLP experiments. These data are shown in Expanded View 4J.
 - Related to Figure 6A – tetherin abundance in lysates from ORF3a VLP experiments These data are shown in Appendix Figure 2A.
 - Related to Figure 6A – N-FLAG abundance in VLP fraction from ORF3a VLP experiments. These data are shown in Appendix Figure S2.

The mean, standard deviation and number of independent experiments analysed are expressed in the figure legends. Methods for densitometry quantification is now included at Line 808.

- **Additional biological replicates have been performed for growth curve assays. Three independent biological replicates were performed for each MOI. These data are shown in Figure 3C and EV Figure 2A and 2B in our revised manuscript.**

2) Insufficient explanation.

I found some of the images and legends contained insufficient annotation and/or description for a non-expert reader to appreciate the result(s). Particularly if the authors want to draw attention to features in micrographs they should consider using more enlarged/inset images and annotations (e.g. arrows) to point out structures (e.g DMVs etc.). This short coming exacerbates the lack of quantification.

- **Additional detail has been given to the figure legends, and to draw attention to features in micrographs, we have added the following;**
 - **Black arrowheads have been added to Figures 1E, 2D, 2H, Expanded View Figure 1K to highlight plasma membrane-associated virions.**
 - **Images showing DMVs in HeLa and A549 cells have been replaced or recropped. Asterisks have been added to Figures 1E, 2D and Expanded View Figure 1E to highlight DMVs. Additional examples of DMVs in infected A549 cells is now included as Expanded View Figure 1I.**
 - **White arrowheads have been added to Figure 2E to highlight typical Golgi cisternae.**
- **These figure legends have also been modified to highlight the additional annotations.**

3) Insufficient exploration of the data.

I had a sense that some aspects of the data seem unconsidered or ignored, and the discussion lacks depth and reflection. For example the tetherin down-regulation apparent in Figures 1 and 2 is not really explained by the spike/ORF3a antagonism described later on, but this is not explicitly addressed. Also, Figure 6 suggests that ORF3a results in high levels of incorporation of tetherin in to VLPs, but I don't think this is even described(?). The discussion should also include more comparison with previous studies on the relationship between SARS-2 and tetherin.

- **We have made changes throughout the manuscript, but the Discussion especially has been modified to expand on the above points. We now discuss the ORF3a data in more depth, discuss possible mechanisms by**

which ORF3a alone enhances VLP release, and discuss our ORF7a data in the context of previous reports. See results and discussion section of revised manuscript.

- **We have added a section to discuss how ORF3a may enhance VLP release,**

'We found that the expression of ORF3a enhanced VLP independently of its ability to relocalise tetherin (**Figure 6A**). This may be due to either the ability of ORF3a to induce Golgi fragmentation (Arshad *et al*, 2023) which facilitates viral trafficking (Zhang *et al*, 2022b), or due to enhanced lysosomal exocytosis (Chen *et al*, 2021). Tetherin was also found in VLPs upon co-expression with ORF3a (**Figure 6A**) which may also indicate enhanced release via lysosomal exocytosis (Chen *et al*, 2021).' - (**Line 509 - 514**).

'We found that the expression of ORF3a enhanced VLP independently of its ability to relocalise tetherin (**Figure 6A**). This may be due to either the ability of ORF3a to induce Golgi fragmentation (Arshad *et al*, 2023) which facilitates viral trafficking (Zhang *et al*, 2022b), or due to enhanced lysosomal exocytosis (Chen *et al*, 2021). Tetherin was also found in VLPs upon co-expression with ORF3a (**Figure 6A**) which may also indicate enhanced release via lysosomal exocytosis (Chen *et al*, 2021).' - (**Line 516 - 520**).

- **The Discussion has been developed to compare the relationship between SARS-CoV-2 and tetherin in previous studies,**

'SARS-CoV-1 ORF7a is reported to inhibit tetherin glycosylation and localise to the plasma membrane in the presence of tetherin (Taylor *et al*, 2015). We did not observe any difference in total tetherin levels, tetherin glycosylation, ability to form dimers, or surface tetherin upon expression of either SARS-CoV-1 or SARS-CoV-2 ORF7a (**Figures 4A, 4B, 4C**). Others groups have demonstrated a role for ORF7a in sarbecovirus infection and both SARS-CoV-1 and SARS-CoV-2 virus lacking ORF7a show impaired virus replication in the presence of tetherin (Taylor *et al*, 2015; Martin-Sancho *et al*, 2021a). A direct interaction between SARS-CoV-1 ORF7a and SARS-CoV-2 ORF7a and tetherin have been described (Taylor *et al*, 2015; Martin-Sancho *et al*, 2021a), although the precise mechanism(s) by which ORF7a antagonises tetherin remains enigmatic. We cannot exclude that ORF7a requires other viral proteins to antagonise tetherin, or that ORF7a antagonises tetherin via another mechanism. For example, ORF7a can potentially antagonise IFN signalling (Xia *et al*, 2020) which would impair tetherin induction in many cell types.' - (**Line 532 - 546**).

I have no minor comments on this draft of the manuscript.

Reviewer #1 (Significance (Required)):

Tetherin, encoded by the BST2 gene, is an antiviral restriction factor that inhibits

the release of enveloped viruses by creating tethers between viral and host membranes. It also has a capacity for sensing and signalling viral infection. It is most widely understood in the context of HIV-1, however, there is evidence of restriction in a wide variety of enveloped viruses, many of which have evolved strategies for antagonising tetherin. This knowledge informs on viral interactions with the innate immune system, with implications for basic virology and translational research.

This study investigates tetherin in the context of SARS-CoV-2. The authors use a powerful collection of tools (live virus, gene knock out cells, recombinant viral and host expression systems) and a variety of approaches (microscopy, western blotting, infection assays), which is, itself, a strength. The study provides evidence to support a series of conclusions: I) BST2/tetherin restricts SARS-CoV-2 II) SARS-CoV-2 ablates tetherin expression III) spike protein can modestly down-regulate tetherin IV) ORF3A dysregulates tetherin localisation by altering retrograde trafficking. These conclusions are broadly supported by the data and this study make significant contributions to our understanding of SARS-CoV-2/tetherin interactions.

My enthusiasm is reduced by, in my opinion, a failure of the authors to fully quantify, explain and explore their data. I expect the manuscript could be significantly improved without further experimentation by strengthening these aspects.

This manuscript will be of interest to investigators in virology and/or cellular intrinsic immunity. Given the focus on SARS-CoV-2 it is possible/likely that it will find a slightly broader readership.

I have highly appropriate skills for evaluating this work being experienced in virology, SARS-CoV-2, cell biology and microscopy.

- **We wish to thank Reviewer 1 for their comments and suggestions. They have been immensely helpful in strengthening our manuscript.**

Reviewer #2 (Evidence, reproducibility and clarity (Required)):

BST2/tetherin can restrict the release and transmission of many enveloped viruses, including coronaviruses. In many cases, restricted viruses have developed mechanisms to abrogate tetherin-restriction by expressing proteins that antagonize tetherin; HIV-1 Vpu-mediated antagonism of tetherin restriction is a particularly well studied example. In this paper, Stewart et al. report their studies of the mechanism(s) underlying SARS-CoV-2 antagonism of tetherin restriction. They conclude that Orf3a is the primary virally encoded protein involved and that Orf3a manipulates endo-lysosomal trafficking to decrease tetherin cycling and divert the protein away from putative assembly sites.

Major comments:-

In my view some of the claims made by the authors are not fully supported by the data. For example, the bystander effect discussed in line 162 may suggest that infected cells can produce IFN but does not 'indicate' that they do.

- **This text has now been edited,**
'The levels of tetherin in uninfected HAE cells is lower than observed in uninfected neighbours in infected wells demonstrating that infected HAE cells are able to generate IFN to act upon uninfected neighbouring cells, enhancing tetherin expression.' - (Lines 167 - 170).

Most of the EM images show part of a cell profile, so statements such as (line 192) 'virus containing tubulovesicular organelles were often polarised towards sites of significant surface-associated virus' should be backed up with appropriate images, or indicated as 'not shown', or removed (the observation is not so important for this story). Line 196, DMVs can't be seen in these micrographs.

- **The statement '*virus containing tubulovesicular organelles were often polarised towards sites of significant surface-associated virus*' has been removed. The micrographs in Figure 1E have been re-cropped, and image iii replaced with an image showing clearer DMVs and budding virions. Additional annotation has been included with TEM data throughout and specifically plasma membrane-associated virions have been highlighted by black arrowheads, DMVs by black asterisks, and intracellular virion by a white arrow. These are highlighted in the figure legends to make features clearer for readers.**

CD63 organelle diameter

Line 391, I can't see much change in CD63 distribution.

- **CD63 reproducibly appears clustered towards the nuclei in ORF3a expressing cells, whilst CD63 positive puncta are abundant in the periphery of mock cells. CD63 puncta are also larger, and the staining of CIMPR and**

VPS35 also appears to be associated with larger organelles. We have amended the text to now read,

‘Expression of ORF3a also disrupted the distribution of numerous endosome-related markers including CIMPR, VPS35, CD63, which all localised to larger and less peripheral puncta (Figure EV 5B), and the mixing of early (EEA1) and late endosomal markers (cathepsin D) (Figure EV 5C)’ - (Line 399 - 405).

- **Quantification of the diameter of CD63 puncta indicate that they are larger in ORF3a expressing cells than in mock cells. Mock cells - 0.71µm (SD; 0.19), ORF3a - 1.15µm (SD;0.35). At least 75 organelles per sample, from 10 different cells. We have not included this data as we do not wish to labor this point but are happy to include this quantification if required to do so.**

Line 321, the authors show that ORF7a does not affect tetherin localization, abundance, glycosylation or dimer formation, but they don't show that it doesn't restrict SARS-CoV-2. Can they be sure that epitope tagging this molecule does not abrogate function (or the functions of any of the other tagged proteins for that matter), or that ORF7a works in conjunction with one of the other viral proteins?

- **We are careful in the manuscript not to claim that ORF7a has no effect on tetherin. When discussing our data we state,**
‘ORF7a does not directly influence tetherin localisation, abundance, glycosylation or dimer formation’ - (Line 320 - 321).
- **We were unable to reproduce an effect of ORF7a on tetherin glycosylation. Our data conflicts with that presented by Taylor et al, 2015, where ORF7a impaired tetherin glycosylation and ORF7a localised to the plasma membrane in tetherin expressing cells. The experiments performed by Taylor et al used HEK293 cells and ectopically expressed tagged tetherin. The differences in results may be attributed to the differences between cell lines or due to differences between endogenous or ectopic / tagged tetherin.**
- **The study by Taylor et al uses SARS-CoV-1 ORF7a-HA from Kopecky-Bromberg et al., 2007 (DOI: [10.1128/JVI.01782-06](https://doi.org/10.1128/JVI.01782-06)), where the -HA tag is positioned at the C-terminus. Our ORF7a-FLAG constructs have a C-terminal epitope tag. While we cannot exclude the possibility that tagged proteins may act differently from untagged ones, the differences between our findings and previous work appear unlikely to be due to epitope tags.**
- **Our manuscript states that although we cannot find any effect of ORF7a on tetherin localisation, abundance, glycosylation, or dimer formation, we cannot exclude that ORF7a impacts tetherin by another mechanism. For example, ORF7a has been found to antagonise interferon responses. Tetherin is abundantly expressed in HeLa cells and expression does not require induction through interferon. None of our experiments above would be impacted by interferon antagonism yet this could impact other cell types besides infection *in vivo*. These possibilities may explain the reported differential impact of ORF7a by different labs. An addition comment has been added to the discussion to reflect this,**

'We cannot exclude that ORF7a requires other viral proteins to antagonise tetherin, or that ORF7a antagonises tetherin via another mechanism. For example, ORF7a potentially antagonises IFN signalling [38], which would impair tetherin induction in many cell types.' - (Line 543 - 546).

Note - Reference 38 has been added to the manuscript – Xia et al., Cell Reports DOI: [10.1016/j.celrep.2020.108234](https://doi.org/10.1016/j.celrep.2020.108234)

In the ORF screen, a number of the constructs are expressed at low level, is it possible they are missing something?

- **Some of the ORFs expressed in the miniscreen appear poorly expressed. We accept that in the use of epitope tagged constructs expression levels of individual viral proteins may impact upon a successful screen. However, this screen was performed to identify any potential changes in tetherin abundance or localisation, and the screen did successfully identify ORF3a, which we were able to follow-up and verify.**

Line 376, the authors refer to ORF3a being a viroporin. A recent eLife paper (doi: 10.7554/eLife.84477; initially published in BioRxiv) refutes this claim and builds on other evidence that ORF3a interacts with the HOPS complex. The authors should at least mention this work, especially in the discussion, as it would seem to provide a molecular mechanism to support their conclusions.

- **This paper had not been peer reviewed at the time of our initial submission. We have now included the following text, 'SARS-CoV-2 ORF3a is an accessory protein that localises to and perturbs endosomes and lysosomes [29]. It may do so by acting either as a viroporin [30] or by interacting with, and possibly interfering with the function of VPS 39, a component of the HOPS complex which facilitates tethering of late endosomes or autophagosomes with lysosomes [29,31]. Given ORF3a likely impairs lysosome function, the observed increased....'** - (Lines 382 - 387).

Fig 3, the growth curves illustrated in Fig3 C and D do not have errors bars; how many times were these experiments repeated?

- **These experiments have now been repeated to give data from three biologically independent experiments, using different virus stocks. We have additionally now performed these experiments at MOI 0.01, in addition to the previous MOIs (1 and 5). The data for these experiments is now shown in Figure 3C and EV Figure 2A and 2B. Data represents the mean +/- SEM of three biological replicates.**

Line 396, the authors show increased co-localization with LAMP1. As LAMP1 is found in late endosomes as well as lysosomes, they cannot claim the redistributed tetherin is specifically in lysosomes.

- **We have altered the text to now say:**
'The ORF3a-mediated increase in tetherin abundance within endolysosomes could be due to defective lysosomal degradation.' - **(Line 407)**.

There seems to be a marked difference in the anti-rb555 signal in the 'mock' cells in panels 5H and Suppl 6E. Is there a good reason for this, or does this indicate variability between experiments?

- **Antibody uptake experiments in Figure 5H and Supp 6E (now EV Figure 5E) were performed and acquired on different days. Relatively low levels of signal are available in these antibody uptake experiments, and the disperse labelling seen in the mocks does not aid this.**
- **For image acquisition, the height was selected by the widest DAPI stain as to not bias the image analysis towards the brightest part of the cell (which may differ between conditions).**

Fig 6a, why is there negligible VLP release from cells lacking BST2 and ORF3a-strep? How many times were these experiments performed? Is this a representative image? I think it confusing to refer to the same protein by two different names in the same figure (i.e. BST2 and tetherin). Do the authors know how the levels of ORF3a expressed in cells in these experiments compares to those seen in infected cells?

- **We have changed the blot in Figure 6A for one with clearer FLAG bands. Three independent experiments were performed for Figure 6A. Quantification of VLPs is now included in Appendix Figure S2. Western blots used are now provided as Source Data.**
- **We have changed 'Bst2' to 'tetherin' in all previous figures relating to protein; Figure 4G, Figure 6A, B, C.**
- **We have no current information to compare ORF3a levels in these experiments versus in infected cells.**

My final point is, perhaps, the trickiest to answer, but nevertheless needs to be considered. As far as we know, SARS-CoV-2 and at least some other coronaviruses, bud into organelles of the early secretory pathway, often considered to be ERGIC. In the experiments shown here the authors provide

evidence that ORF3a can influence tetherin recycling, but the main way of showing this is through its increased association with endocytic organelles. Do the authors have any evidence that Orf3a reduces tetherin levels in the ERGIC or whether the tetherin cycling pathway(s) involve the ERGIC?

- **This is an interesting point, and as the reviewer concedes, this is tricky to answer. Infection of cells with SARS-CoV-2 or ectopic expression of ORF3a causes the redistribution or remodeling of various organelles (Figures 1E, 2D, 2F, 2D-I, EV Figures 1I, 1K, 3E, 3B-G). We have been unable to test the direct involvement of ERGIC, despite attempts with a number of commercial antibodies (ProteinTec #13364-1-AP, Santa Cruz #sc-398777). Given the huge rearrangements of organelles during SARS-CoV-2 infection, it is unclear exactly what will happen to the distribution of ERGIC.**

Minor comments:-

Overall, the manuscript should be carefully edited to ensure the text reads clearly.

A few examples of thing that need to be fixed are:-

Line 53, delete 'shell' its redundant and confusing when the authors have said coronaviruses have a membrane.

- **Thanks. This has been deleted.**

Line 61, delete 'the'

- **Thanks. This has been deleted.**

Line 72, delete 'enveloped'; coronaviruses already described as enveloped viruses (line 53)

- **Agree, this has been deleted.**

Lines 93 - 100, lop-sided discussion of the viral life cycle; this paragraph is mostly about entry, which is not relevant to this paper, and does not really deal with the synthesis and assembly side of the cycle.

- **We have now added the following text,**
'...liberating the viral nucleocapsid to the cytosol of the cell. Upon uncoating, the RNA genome is released into the host cytosol and replication-transcription complexes assemble to drive the replication of the viral genome and the expression of viral proteins. Coronaviruses modify host organelles to generate viral replication factories - so-called DMVs (double-membrane vesicles) that act as hubs for viral

RNA synthesis [10]. SARS-CoV-2 viral budding occurs at ER-to-Golgi intermediate compartments (ERGIC) and newly formed viral particles traffic through secretory vesicles to the plasma membrane where they are released to the extracellular space.'
- **(Lines 99 - 107).**

Line 103, why are the neighbouring cells 'naive'?

- **'naïve' removed.**

Line 112 - 113, delete last phrase; tetherin is described as an IFN stimulated gene in line 111; to be accurate, the beginning of the sentence should be 'Tetherin is expressed from a type 1 Interferon stimulated gene ...'

- **Amended as suggested.**

Line 118 - 119, should say 'For tetherin-restricted enveloped viruses' as not all enveloped viruses are restricted by tetherin.

- **Amended as suggested.**

Line 131, coronaviruses are not the only family of tetherin-restricted viruses that assemble on intracellular membranes, e.g. bunyaviruses.

- **This has been modified and now reads,**
'In order for tetherin to tether coronaviruses, tetherin must be incorporated in the virus envelope during budding which occurs in intracellular organelles.' - **(Lines 138 - 140).**

Line 192, there is no EM data in Supplemental Fig 1C.

- **This has now been removed.**

Line 251, 'a synchronous infection event' should be 'synchronous infection' as there will be multiple infection events

- **This has been changed.**

Page 13 (and elsewhere), unlike Southern, 'Western' should not have a capital letter, except at the start of a sentence.

- **These have been updated throughout the manuscript.**

Lines 330 and 352, can the authors quantitate S protein-induced reduction in cell surface tetherin rather than using the somewhat subjective 'mild'?

- **These are now changed to,**
'Transient transfection of cells with ss-HA-Spike caused a 32% decrease in tetherin as observed by immunofluorescence (**Figures EV 4A, 4B**), with...' – **(Line 328 - 330).**

'To explore whether the [~~mild – removed~~] Spike-induced tetherin downregulation altered virus release, we performed experiments with virus like particles (VLPs) in HEK293T ...' – **(Line 353 - 354).**

Line 379, OFR, should be ORF.

- **Thank you, changed.**

Line 448, 'Tetherin retains the ability' - did it ever lose it?

- **This has been rephrased to,**
'Tetherin has the ability to restrict a number of different enveloped viruses that bud at distinct organelles.' - **(Line 464).**

Line 451, 'luminal' is confusing in this context.

- **This has been modified to,**
'Tetherin forms homodimers between opposing membranes (e.g., plasma membrane and viral envelope) that are linked via disulphide bonds.' - **(Line 465 - 466).**

Line 453, the process of virus envelopment is likely to be more than a 'single step'

- **This has been removed and now reads,**
'...virus during viral budding, which occurs in modified ERGIC organelles.' - **(Line 468).**

Line 457, in my view the notion that Vpu abrogation of tetherin restriction is just due to redistribution of tetherin to the TGN is somewhat simplistic and disregards a lot of other work.

- **We have removed mention of mechanisms of tetherin antagonism by other viruses. The key point we wish to make here is that tetherin is lost from the budding compartment. This now reads,**
'Many enveloped viruses antagonise tetherin by altering its localisation and removing it from the respective site of virus budding.' - **(Line 468 - 470).**

Line 472, what is meant by 'resting states'?

- **This should have been 'in the absence of stimulation' and have now been re-written,**
'Tetherin is an IFN-stimulated gene (ISG) [13], and many cell types express low levels of tetherin in the absence of stimulation.' - **(Line 476).**

Line 1204, how were 'mock infected cells infected'?

- **This has now been re-written,**
'Differentiated nasal primary human airway epithelial (HAE) cells were embedded to OCT....' - **(Line 1277).**

Reviewer #2 (Significance (Required)):

This study builds on published work supporting the notion that SARS-Cov-2 ORF3a is an antagonist for the restriction factor tetherin. Importantly, it provides insights to the mechanism of ORF3a mediated tetherin antagonism, specifically to ORF3a inhibits tetherin cycling, diverting the protein to lysosomes and away from compartment(s) where virions assemble. Overall, the authors provide good supporting evidence for these conclusions, however there are issues that the authors need to address.

- **We wish to thank Reviewer 2 for their thoughtful comments and suggestions. Our revised manuscript benefits from their input and suggestions for improvement.**

Reviewer #3 (Evidence, reproducibility and clarity (Required)):

Restriction factors are major barriers against viral infections. A prime example is Tetherin (aka BST2), which is able to physically tether budding virions to the plasma membrane preventing release of the infectious particles. Of note, tetherin has broad anti-viral activity and has been established as a crucial innate immune defense factor against HIV, IAV, SARS-CoV-2 and other important human pathogens. However, successful viruses like SARS-CoV-2 evolved strategies to counteract restriction factors and promote their replication. Important restriction factors, such as tetherin, may often be targeted by multiple viral strategies to ensure complete suppression of their anti-viral activities by the pathogen. Of note, it was previously published that the accessory protein ORF7a of SARS-CoV-2 binds to (Petrosino et al, Chemistry Europe, 2021) and antagonizes it (Martin-Sancho et al, Molecular Cell, 2021). Previous data on SARS-CoV also revealed that ORF7a promotes cleavage of tetherin (Taylor et al, 2015, J Virol).

In this manuscript, the authors show that tetherin restricts SARS-CoV-2 by tethering virions to the plasma membrane and propose that tetherin is targeted by two proteins of SARS-CoV-2. Whereas the Spike protein promotes degradation of tetherin, the accessory protein ORF3a redirects tetherin away from newly forming SARS-CoV-2 virions.

While the overall findings that both S and ORF3a are additionally targeting tetherin is both novel and intriguing, additional evidence is needed to support this. In addition, the authors show that in their experimental setups ORF7a does not induce cleavage of tetherin. This is in direct contrast to previously published data both on SARS-CoV(-1) and -2 (Taylor et al, 2015, J Virol; Petrosino et al, Chemistry Europe, 2021; Martin-Sancho et al, Molecular Cell, 2021). From my point of view that needs further experimental confirmation. While the authors state that the impact of Spike on tetherin is mild, the experiments should still allow the conclusion whether there is a (mild) effect or not. The mechanism of ORF3a is fortunately more robustly assessed and provides some novel insights.

Unfortunately, the whole manuscript suffers from a striking lack of quantifications. In addition, it is not clear whether and how many times experiments were repeated to the same results. Overall, the data in this manuscript seem very speculative and preliminary and thus do not support the authors conclusions.

Major:

Much of the data seems like it was only done once. As I am sure that this is a writing issue, please clearly state how many times the individual assays were repeated, provide the quantification graphs and appropriate statistics. Some

experiments may need additional quantification and confirmation by other methods to be convincing.

- **Additional quantification is provided throughout the revised manuscript. The figure legends have also been updated to provide information on quantification, statistical analysis and number of independent experiments (biological or technical replicates indicated).**
- **A full list of additional quantification is provided as a table at the end of this document.**

For example, Figure 1A, C and D: Please quantify the levels of tetherin and use an alternative readout, e.g. Western blotting of infected cells.

- **Quantification has been performed and included in our revised manuscript in Extended View Figure 1C, 1E. Tetherin is not shown in Figure 1C. A table is provided on the last page of this document to highlight the additional quantification.**

Figure 2A: Please quantify.

- **We are not sure we understand this point. The western blot shown in Figure 2A demonstrates the ectopic expression of ACE2 in our A549 cell line. A549 cells have been used by many labs to study SARS-CoV-2 infection, but express negligible ACE2. Infection of A549 or HeLa cells by SARS-CoV-2 requires ectopic expression of ACE2 (<https://doi.org/10.1016/j.cell.2020.02.052>, <https://doi.org/10.1016/j.cell.2020.04.026>)**

Fig 3A: Please show and confirm successful tetherin KO in the cell lines that are used not only in microscopy.

- **The tetherin KO HeLa cell line used in this study were generated and characterised by our lab previously (Edgar et al, 2016). A new blot is now shown in Figure 3A to demonstrate the presence / absence of ACE2 and tetherin in the cell lines used.**

Figure 4C: Please quantify.

- **Flow data including gating strategies and MFIs are now provided with Source Data. Two independent experiments were performed and as such data cannot be quantified as per EMBO Reports policy. No reproducible differences were observed on surface tetherin levels upon expression of SARS-CoV-1 or SARS-CoV-2 ORF7a. The flow cytometry shown in Figure 4C reproduces our fluorescence and biochemical data (4A, 4B).**

Figure 4D: Please quantify the effects are not obvious from the images provided.

- **Quantification is now provided in Expanded View Figure 4E.**

Figure 4E,F Please provide a quantification of multiple independent repeats, the claimed differences are neither striking nor obvious.

- **Figure 4E - as above, the gating strategy and MFIs are now provided with Source data. A mild decrease in tetherin is observed upon Spike induction, and is reproduced by Figures 4D (EV4E), 4F (EV4G), EV4F, 4G (EV4I).**
- **Quantification of 4F is now provided in Expanded View Figure 4G. In these experiments, Tetherin levels were quantified to be reduced by 25% (SD: 8%) by addition of Doxycycline and induction of ss-HA-Spike. These data are from three biological replicates.**

Figure 5A: Please quantify.

- **Two independent experiments were performed. No reproducible effect can be seen on surface tetherin levels by expression of any ORF examined. The MFIs are now provided with Source Data. Although the flow cytometry experiments were not revealing, this approach was more fruitful when looking at the fluorescence localisation by confocal microscopy (shown Appendix Figure S1B).**

Figure 3C and D: At timepoint 0 the infection input levels are different. The initial infection levels have to be the same to draw the conclusion that tetherin KO affects virion release and not the initial infection efficiency. Can the authors either normalize or ensure that the initial infection is the same in all conditions and that variations in the initial infection efficiency do not correlated with the impact of tetherin on replication/release ? How often were those experiments repeated? Are the marginal differences in infectious titre significant? Overall the impact of tetherin on SARS-CoV-2 is very underwhelming but that may be due to efficient viral tetherin-counteraction strategies. Why is the phenotype inverted at 72 h?

- **Equal amounts of virus, as measured by plaque-forming units (PFU), were used for both HeLa cell lines and thus at 0 hours post infection the variation seen is within the parameters of the assay. It remains possible that tetherin affects virus entry, but this is unlikely and this assay was not designed to investigate that hypothesis. Given both cell lines express equivalent levels of ACE2 we do not believe there are any additional controls we could include to ensure equal input. Normalisation to the 0**

hour sample would not be an accurate way to display the data: due to viral infection following a Poisson distribution, there will be variation between wells during each experiment, and each well was separately harvested at 0, 24 and 48 hours post infection. However, the main figure has now been edited to display the released virus as a percentage of total virus for each sample (Figure 3C); this method of normalisation removes any confounding effects which may inadvertently arise from different input levels.

- The released and intracellular virus titres are displayed in detail in Figure EV 2. Three independent biological replicates were conducted (using different virus stocks) and the data represents the mean +/- SEM. Multiple *t* tests were conducted on the viral titres using the Holm–Sidak method for multiple comparisons ($\alpha = 0.05$). Each time point was analysed individually, without assuming a consistent SD.
- A ratio paired *t*-test was conducted on the released virus percentages (Figure 3C). All the data sets exhibit a similar obvious trend, indicating that more virus particles are released from Bst2KO HeLa cells than WT, with statistical significance observed at 24 hours post infection (MOI 5).
- The 72 hour time point, included in the previous manuscript version, has been removed. At this time point, the extensive cell death caused by viral replication (especially at higher MOIs) makes it difficult to accurately separate the released from intracellular fractions and conclusions cannot be accurately drawn from the data.

Figure 4B and C: Can the authors provide an explanation why SARS-CoV ORF7a is not inducing cleavage/removes glycosylation of tetherin. To show that the assays work, an independent positive control needs to be included.

- **We are careful in the manuscript not to claim that ORF7a has no effect on tetherin. When discussing our data we state,** ‘ORF7a does not directly influence tetherin localisation, abundance, glycosylation or dimer formation’ - (Line 320 - 321).
- **We were unable to reproduce an effect of ORF7a on tetherin glycosylation previously reported. Our data conflicts with that presented by Taylor et al, 2015, where ORF7a impaired tetherin glycosylation and ORF7a localised to the plasma membrane in tetherin expressing cells. The experiments performed by Taylor et al used HEK293 cells and ectopically expressed tagged tetherin. The differences in results may be attributed to the differences between cell lines or due to differences between endogenous or ectopic / tagged tetherin.**
- **Our manuscript states that although we cannot find any effect of ORF7a on tetherin localisation, abundance, glycosylation, or dimer formation, we cannot exclude that ORF7a impacts tetherin by another mechanism. For example, ORF7a has been found to antagonise interferon responses. Tetherin is**

abundantly expressed in HeLa cells and expression does not require induction through interferon. None of our experiments above would be impacted by interferon antagonism yet this could impact other cell types besides infection *in vivo*. These possibilities may explain the reported differential impact of ORF7a by different labs. An addition comment has been added to the discussion to reflect this,

'We cannot exclude that ORF7a requires other viral proteins to antagonise tetherin, or that ORF7a antagonises tetherin via another mechanism. For example, ORF7a potentially antagonises IFN signalling [38], which would impair tetherin induction in many cell types.' - (Line 543 - 546).

Note - Reference 38 has been added to the manuscript – Xia et al., Cell Reports DOI: [10.1016/j.celrep.2020.108234](https://doi.org/10.1016/j.celrep.2020.108234)

The FACS data in C is unfortunately not quantified.

- **MFIs are now provided with Source Data. Two independent experiments were performed and as a result not quantified as per EMBO Reports policy. A small, but reproducible decrease in surface tetherin is observed upon induction with Doxycycline.**

Fig 4G: The rationale and result of this experiment are not clear.

- **The rationale for Spike VLP experiments is explained at Line 353, 'To explore whether the Spike-induced tetherin downregulation altered virus release, we performed experiments with virus like particles (VLPs) in HEK293T cells which do not express endogenous tetherin.'**

Fig 6: What is the benefit of doing the VLP assays as opposed to genuine virus experiments? To me it rather seems to be making the data unnecessarily complex. Again, no quantifications or repeats are provided.

- **VLPs are used to separate the budding and release process from the replication process of RNA viruses. VLPs have been used in a number of SARS-CoV (DOI: [10.1002/jmv.25518](https://doi.org/10.1002/jmv.25518)) and HIV-1 (DOI: <https://doi.org/10.1186/1742-4690-7-51>) studies to analyse the impact of tetherin (and tetherin mutants) on release.**
- **VLP experiment quantification are now included throughout. For each experiment, three independent biological replicates have been quantified and are now shown, discussed in the text and included in Figure legends.**

Minor

Fig 1D: How do the authors explain the mainly intracellular Spike staining?

- **We do not understand this point. Spike staining is intracellular, whether expressed alone or in the context of infected cells.**
- **Anti-Spike antibodies labelled intracellular membranes in HeLa cells following SARS-CoV-2 infections. Further examples are now included with Source Data.**

Please add statistical analyses on the data e.g. Fig. 3 C and D

- **We have now performed statistical analysis for these experiments.**
- **In order to do so, we performed additional experiments and included experiments at MOI 0.01. The main figure has now been edited to display the released virus as a percentage of total virus for each sample (Figure 3C). The released and intracellular virus titres are displayed in detail in Figure EV 2. Three independent biological replicates were conducted (using different virus stocks) and the data represents the mean +/- SEM. For the viral titres, multiple *t* tests were conducted using the Holm–Sidak method for multiple comparisons ($\alpha = 0.05$). Each time point was analysed individually, without assuming a consistent SD. A ratio paired *t*-test was conducted on the released virus percentages (Figure 3C). All the data sets exhibit a similar obvious trend, indicating that more virus particles are released from Bst2KO HeLa cells than WT, with statistical significance observed at 24 hours post infection (MOI 5). The 72 hour time point, included in the previous manuscript, has been removed.**

Fig. 4B and F: Why do the annotated sizes of tetherin differ between the blots?

- **Figures 4B and 4F are run in non-reduced and reduced conditions respectively. In order to best show the dimer deficient C3A-Tetherin, blots are typically run in non-reduced conditions to exemplify dimer formation and to highlight any defects in dimer formation. The rest of the blots in the manuscript are run in denaturing conditions to aid blotting of other proteins. This is now highlighted in the Figure legend, 'Bst2KO cells rescued with C3A-Tetherin-HA express tetherin that is unable to form homodimers. Blots were performed in non-reducing conditions to highlight the loss of dimer formation by C3A-Tetherin.' - (Line 1178 – 1180).**

Fig. 5A: What is ORF6a? Do the authors mean ORF6?

- **Thanks, this has been changed.**

An MOI of 1 is NOT considered a low or relevant MOI. Can the authors either rephrase or repeat experiments with an actual low or relevant MOI i.e. 0.01 ?

- **Additional experiments have now been included at MOI 0.01, in addition to the previous MOIs (1 and 5). The data for these experiments is now shown in Figure 3C and Figure EV 2. Regardless of the MOI used, we observed a clear reproducible trend indicating that infected Bst2KO HeLa cells released a higher proportion of their virions than WT cells.**

Why were the cell models switched between Figure 1 and 2 and essentially the same experiments repeated?

- **HeLa cells express high levels of tetherin at steady state, whilst A549 cells require IFN stimulation. HeLa cells demonstrate that tetherin downregulation occurs via an IFN-independent manner. A549 and T84 cells are more physiologically relevant cell types for SARS-CoV-2 infection. These points are stated in Lines 230 and 265.**
- **The use of analogous approaches allowed us to demonstrate that tetherin downregulation by SARS-CoV-2 was a common feature of infection. The use of HeLa cells also made us more confident that tetherin antagonism was occurring independently of any IFN antagonism.**

The manuscript may benefit a lot from streamlining and removing unessential deviations from the main message (e.g. discussions why multistep/single step growth curves are used/not relevant; why are they shown if the authors conclude that a single step is not relevant?). The discussion is extremely lengthy and does not provide sufficient discussion of the presented data.

- **As the reviewer suggested, we have removed a significant amount of this text. We feel that the use of three different MOIs for these experiments is important as it allows comparisons across a range of physiologically relevant situations, and indeed it has revealed a robust phenotype confirming tetherin restriction of SARS-CoV-2. The text has been edited to highlight the main conclusions of these experiments (Line 265 – 275). It now reads:**
'For all MOIs tested, intracellular virions accumulated primarily in the first 24 hours post infection. WT HeLa+ACE2 cells showed significantly higher intracellular titres than Bst2KO cells in the lowest MOI (0.01) experiments (Figure EV 2B), whilst the released virus titres were fairly similar (Figure EV 2A). This provides evidence of tetherin-mediated restriction, whereby virions accumulated in WT cells. In contrast, at the higher MOIs of 1 and 5, the intracellular viral titres were similar regardless of the cell genotype whilst the Bst2KO cells released

reproducibly higher virion numbers. Irrespective of the MOI used, when the released virions were viewed as a proportion of total infectious particles, the Bst2KO HeLa+ACE2 cells consistently released more than WT HeLa+ACE2 cells at both 24 and 48 hpi, due to their inability to tether nascent virions (Figure 3C).'

- **We have removed from the Discussion a small section discussing ORF7a mutants, given that the emphasis of our manuscript is not on ORF7a. We have also removed a small section describing the rearrangements of intracellular organelles by SARS-CoV-2 as it does not directly relate to the central message of our manuscript.**

Reviewer #3 (Significance (Required)):

According to my opinion, the current manuscript does not provide significant advancement for the field. While the intention was to update and expand our existing knowledge about tetherin restriction by SARS-CoV-2, the experiments do not support this yet. However, the general premise and approach/concept of the manuscript would be appealing to a broader audience. I especially like the notion that multiple proteins of SARS-CoV-2 could synergistically counteract an important innate immune defense factor, tetherin.

My expertise is on SARS-CoV-2 and the interplay between the virus and host cell restriction factors.

- **We wish to thank Reviewer 3 for their comments and suggestions for this work.**

Summary of new data included in revised manuscript

Quantification of existing data:

Data shown	Quantification now shown in	Method	Analysis
Figure 1A	Figure EV1C	IF	HAE (-/+ SARS-CoV-2) - Tetherin total fluorescence intensity
Figure 1D	Figure EV1E	IF	HeLa+ACE2 (-/+ SARS-CoV-2) - Tetherin total fluorescence intensity
Figure 2C	Figure EV1H	IF	A549+ACE2 (-/+ SARS-CoV-2)- Tetherin total fluorescence intensity
Figure 2G	Figure EV1J	IF	T84 (-/+ SARS-CoV-2) - Tetherin total fluorescence intensity
Figure EV4A	Figure EV4B	IF	HeLa + ss-HA-Spike transients (-/+ HA stained cells) - Tetherin total fluorescence intensity
Figure 4D	Figure EV4E	IF	HeLa + TetOne ss-HA-Spike stables (-/+ Dox) - Tetherin total fluorescence intensity
Figure 4F	Figure EV4G	W blot - lysates	HeLa + TetOne ss-HA-Spike stables (-/+ Dox) – Tetherin abundance
Figure 4G	Figure EV4I	W blot – lysates	Spike VLP experiments – tetherin abundance
Figure 4G	Figure EV4J	W blot - VLPs	Spike VLP experiments - N-FLAG abundance
Figure 6A	Appendix FS2A	W blot – lysates	ORF3a VLP experiments – tetherin abundance
Figure 6A	Appendix FS2B	W blot - VLPs	ORF3a VLP experiments - N-FLAG

New data:

* Figure 3C: Additional biological replicates were performed at MOI 1 and MOI 5. New growth curve assays were performed at MOI 0.01 and three biological replicates were performed. This data is shown in Figure 3C and Expanded View Figure 2A and 2B.

* Expanded View Figure 1I: More TEM images provided of SARS-CoV-2 infected A549+ACE2 cells. Arrowheads and asterisks highlighting virions at the cell surface and DMVs respectively.

Updated / modified data:

* Figure 1E: Micrograph (i) has been uncropped to show DMV. Micrograph in (iii) has been changed to show clear DMVs and budding virus.

* Figure 2D: Micrographs (ii) and (iii) have been added to replace previous images which were blowups from part (i).

* Additional annotation added to Figures 1E, 2D, 2E, 2H, EV F1I, EV F1K.

* Figure 3A: Western blot updated to include tetherin for WT/Bst2KO HeLa +/- ACE2 cell lines.

* All source data is now included. All images used during analysis have been deposited to the BioImage Archive database, <https://www.ebi.ac.uk/biostudies> and assigned the identifier S-BIAD687.

Dear Dr. Edgar

Thank you for the submission of your research manuscript to our editorial offices. Please accept my apologies for the delay in handling your manuscript. We have now received the enclosed reports on it.

As you will see, the referees acknowledge that you have significantly improved your study and data during the revision, but several concerns, some of them major, remain:

- Tetherin and Spike protein largely localize to intracellular compartments rather than at the plasma membrane. How reliable is the HeLa/A549 cell system to study the functional impact of ORF7a on tetherin localization?
- VLP data cannot replace experiments with genuine SARS-CoV2 virus.
- Effect of ORF3a on tetherin expression and virus release is unclear/not fully supported by the data.

I have discussed these concerns further with the referees. Referee 1 indicated that the concern regarding HeLa/A549 system can be addressed by toning down and discussion, as experiments in other cell types were already provided. Referee 1 however also considers it essential to perform at least on genuine virus experiment (i.e., not VLPs) to show how virus expressed ORF3a impacts tetherin and the subsequent release of infectious viruses. Referee 1 agreed with this suggestion to perform confirmatory experiments with authentic virus.

Given these contrasting but also supporting comments from all three referees, we have decided to give you the possibility of another round of revision to address these major and all the other minor concerns.

I have to tell you though that it is EMBO reports policy to reassess novelty if manuscripts are resubmitted more than 6 months after a first decision has been made but this usually does not pose a problem.

In addition to these referee concerns there are also a number of things that I ask you to address from the editorial side:

- Please update the 'Conflict of interest' paragraph to our new 'Disclosure and competing interests statement'. For more information see <https://www.embopress.org/page/journal/14693178/authorguide#conflictsofinterest>
- Please remove the Author Contributions from the manuscript file and make sure that the author contributions in our online submission system are correct and up-to-date. The information you specified in the system will be automatically retrieved and typeset into the article. You can enter additional information in the free text box provided, if you wish.
- Please check your funding information in the online submission system: 204464/Z/16/Z is listed twice with different funders.
- The funding information in the manuscript should be included in the Acknowledgements section.
- Please call out all panels for EV Figures. Callouts are currently missing for: EV1A-K, EV2A-B, EV3A-E, EV4B-J, EV5A-G
- Preprint citations: Please also include the prefix 'preprint' in the text whenever you cite a preprint, e.g. (preprint: Matheson & Lehner, 2020).
- Appendix Figure S1 and S2 should be included in a single Appendix PDF with a table of contents (incl. page numbers). The legends should be removed from manuscript file and placed below the corresponding figure.
- I attach to this email a related manuscript file with comments by our data editors. Please address all comments and upload a revised file with tracked changes with your final manuscript submission. I have also taken the liberty to make some changes to the Abstract. Could you please review it.
- Finally, EMBO reports papers are accompanied online by A) a short (1-2 sentences) summary of the findings and their significance, B) 2-3 bullet points highlighting key results and C) a synopsis image that is 550x300-600 pixels large (width x height) in PNG for JPG format. You can either show a model or key data in the synopsis image. Please note that the size is rather small and that text needs to be readable at the final size. Please send us this information along with the revised manuscript.

We look forward to seeing a further revised version of your manuscript as soon as possible. Please let me know if you have any questions regarding the revision.

With kind regards

Referee #1:

I really appreciate the quality and the quantity of the additional work that the authors performed. It significantly strengthens the study and provides optimized image quality as well as quantifications of the results. Unfortunately, a few of my original concerns were not fully addressed.

I do appreciate the honesty regarding ORF7a. I personally do not think that all published data has to be reproduced, especially if the own, carefully controlled and reproduced experiments paint a different picture. I do think that information would also be important for future readers. Thus, I would encourage the authors to comment on this finding in the paper, and not ignore it.

I had assumed that both virus-expressed Spike and endogenous tetherin represent proteins that should mainly localize to the plasma membrane. Thus, I was concerned why in the current study (usually in HeLa/A549 cells) both endogenous tetherin and Spike have a predominantly intracellular. This may be caused by different cell types or permeabilization methods. For example, in Fig. 4A or D and in Fig 5D or F mock, tetherin is a bit at the membranes but the majority of it seems to cluster at/around the TGN. Compare to that Fig. 1A and 2G, show a much more 'expected' Spike and tetherin localization. Thus, I was wondering whether the functional impact of tetherin on the virus/VLPs as well as shifts in tetherin localization upon ORF7a expression can be accurately assessed in HeLa/A549 cells. Can the authors explain/clarify this?

The VLP experiments still seem a bit of an inferior and convoluted approach compared to experiments with genuine SARS-CoV-2. I understand the argument that this system separates replication and entry and I do realize that VLP or Pseudotype systems have been used in the majority of previous studies. However, I am concerned that this may have led occasionally to misleading results. Especially the last two/three years have highlighted for me that pseudotyping systems have been excessively used to replace genuine virus experiments, sometimes resulting in conflicting data i.e. interpretations that may be true for the pseudotype, but not the genuine virus. Thus, I would have appreciated that the impact on entry would have been studied using real SARS-CoV-2 that has the right shape and protein composition to accurately assess the impact of tetherin. Experimentally the VLP data can be complemented, by synchronizing a SARS-CoV-2 infection, and then quantifying the release of viral genomic RNA into the cytoplasm or quantifying the first subgenomic mRNAs appearing.

Referee #2:

Stewart et al. have addressed many of the points I raised in my initial review and they have improved their manuscript. However, I think there are still a few changes that are necessary before the paper can be published.

My major concern is that, in my view, the authors need to be more cautious with some of their conclusions. For example, ORF3a does not appear to affect tetherin expression (line 389-392), yet the authors report that virus infection of cells does decrease tetherin levels (e.g. line 189). Although the data show an impact of ORF3a expression on tetherin retrograde trafficking, they do not actually know the extent to which tetherin levels are modulated in the early compartments of the exocytic pathway, e.g. ERGIC, where virions are believed to assemble (rebuttal letter). For these reasons, the authors need to be very cautious with their conclusions, statements such as 'By removing tetherin from the Coronavirus budding compartments, ORF3a enhances virus release.' (line 34) are in my view not fully supported by the data.

In addition, there are a number of minor points that should be fixed/considered.

Line 50 - Should be 'lysosome-associated' not 'lysosomal-associated'.

Line 54 - I don't think 'voracious' is an appropriate adjective.

Line 167 - Should 'uninfected' be 'infected'?

Line 168 - In my view 'demonstrating' should be 'suggesting' as they don't show that IFN is produced.

Line 189 - While it does appear that tetherin expression is lower in infected cells, I think it is very hard to see clearly that there is tetherin on the PM and less tetherin on the PM of infected cells. The authors should limit their claim to tetherin reduction in infected cells.

Line 231 and Fig.2D - the message is confusing. Line 231 states that very few surface-associated virions were present, likely due to the significant tetherin downregulation' but the Fig 2D legend states 'Infected cells displayed occasional virion clusters'.

Can the authors clarify please.

Line 265 and Fig.EV2 - it's not clear to me how reliable the data is for intracellular infectious viruses; how stable are these viruses to freeze/thaw cycles and is it known whether these viruses have undergone furin-mediated S1/S2 cleavage. Virions with non-cleaved S presumably do not form plaques.

Line 465 - the sentence 'Tetherin forms homodimers between opposing membranes (e.g., plasma membrane and viral envelope) that are linked via disulphide bonds.' Should be rewritten as 'Tetherin forms disulphide-linked homodimers that can link opposing membranes (e.g., plasma membrane and viral envelope).' To avoid suggesting membranes are linked by disulphide bonds.

Line 497 - defective recycling of tetherin has been demonstrated for HIV-1 Vpu-mediated downregulation of tetherin; thus, it could be argued that altered trafficking is not a 'novel' mechanism.

Line 506 - I think the authors should perhaps give more emphasis to the Miller et al. paper (2023); these authors show very clearly that ORF3a is not an ion channel.

Line 557 - the authors introduce the concept of cell-to-cell transmission - this needs to be explained.

Referee #3:

In my original review I outline three major areas that required revision - image quantification, image annotation/explanation, and proper exploration/discussion of the data.

The authors have made considerably efforts to address these concerns - providing lots of image quantification data, annotation to EM images and two new paragraphs in the discussion. By and large I am satisfied that my concerns have been reasonably considered and addressed.

I would note that the authors have provided statistical significance testing on many, but not all, of the new data. For clarity, testing should be performed throughout, and if no significance is found this should be stated on the figure. This may also require some rewording to ensure the text properly reflects data significance.

Dear Dr. Edgar

Thank you for the submission of your research manuscript to our editorial offices. Please accept my apologies for the delay in handling your manuscript. We have now received the enclosed reports on it.

As you will see, the referees acknowledge that you have significantly improved your study and data during the revision, but several concerns, some of them major, remain:

- Tetherin and Spike protein largely localize to intracellular compartments rather than at the plasma membrane. How reliable is the HeLa/A549 cell system to study the functional impact of ORF7a on tetherin localization?
- VLP data cannot replace experiments with genuine SARS-CoV2 virus.
- Effect of ORF3a on tetherin expression and virus release is unclear/not fully supported by the data.

I have discussed these concerns further with the referees. Referee 1 indicated that the concern regarding HeLa/A549 system can be addressed by toning down and discussion, as experiments in other cell types were already provided. Referee 1 however also considers it essential to perform at least on genuine virus experiment (i.e., not VLPs) to show how virus expressed ORF3a impacts tetherin and the subsequent release of infectious viruses. Referee 1 agreed with this suggestion to perform confirmatory experiments with authentic virus.

Given these contrasting but also supporting comments from all three referees, we have decided to give you the possibility of another round of revision to address these major and all the other minor concerns.

I have amended the manuscript as per our discussion via Teams last week. As discussed in our call, we have re-written parts of the manuscript to address points relating to HeLa/A549, and to clarify the effect of ORF3a on tetherin (details below). Whilst genuine virus experiments would be an excellent addition, there are several reasons why ORF3a-deletion viruses are not appropriate for this study (as discussed in our recent call and outlined below).

I have to tell you though that it is EMBO reports policy to reassess novelty if manuscripts are resubmitted more than 6 months after a first decision has been made but this usually does not pose a problem.

In addition to these referee concerns there are also a number of things that I ask you to address from the editorial side:

- Please update the 'Conflict of interest' paragraph to our new 'Disclosure and competing interests statement'. For more information see <https://www.embopress.org/page/journal/14693178/authorguide#conflictsofinterest>

Done.

- Please remove the Author Contributions from the manuscript file and make sure that the author contributions in our online submission system are correct and up-to-date. The information you specified in the system will be automatically retrieved and typeset into the article. You can enter additional information in the free text box provided, if you wish.

Done.

- Please check your funding information in the online submission system: 204464/Z/16/Z is listed twice with different funders.

Discussed in the call. This applies to two grants (216370/Z/19/Z and 204464/Z/16/Z. I think this is because these funds comes jointly from the Wellcome Trust and the Royal Society. On the online system I can only add one funder at a time so I have done it twice for each. In the manuscript they are listed (correctly) as 'jointly funded by the Wellcome Trust and the Royal Society'

- The funding information in the manuscript should be included in the Acknowledgements section.

Done.

- Please call out all panels for EV Figures. Callouts are currently missing for: EV1A-K, EV2A-B, EV3A-E, EV4B-J, EV5A-G

Done. I have removed spaces between EV and numbers.

- Preprint citations: Please also include the prefix 'preprint' in the text whenever you cite a preprint, e.g. (preprint: Matheson & Lehner, 2020).

Done – I have been through the reference list and updated where appropriate. Have also added 'Preprint' as a prefix where appropriate in the text.

- Appendix Figure S1 and S2 should be included in a single Appendix PDF with a table of contents (incl. page numbers). The legends should be removed from manuscript file and placed below the corresponding figure.

Done.

- I attach to this email a related manuscript file with comments by our data editors. Please address all comments and upload a revised file with tracked changes with your final manuscript submission. I have also taken the liberty to make some changes to the Abstract. Could you please review it.

Done – many thanks for looking through the manuscript and suggested edits. I have made comments on the file and further edits (as discussed) are shown by track changes.

- Finally, EMBO reports papers are accompanied online by A) a short (1-2 sentences) summary of the findings and their significance, B) 2-3 bullet points highlighting key results and C) a synopsis image that is 550x300-600 pixels large (width x height) in PNG for JPG format. You can either show a model or key data in the synopsis image. Please note that the size is rather small and that text needs to be to be readable at the final size. Please send us this information along with the revised manuscript.

A) The anti-viral restriction factor Tetherin blocks the egress of SARS-CoV-2. Expression of ORF3a and Spike protein antagonise tetherin to allow SARS-CoV-2 escape.

B)

- Tetherin loss aids SARS-CoV-2 viral spread
- SARS-CoV-2 ORF3a alters tetherin localisation by impairing retrograde traffic and enhances virus release
- SARS-CoV-2 Spike causes tetherin downregulation.

C)

We look forward to seeing a further revised version of your manuscript as soon as possible. Please let me know if you have any questions regarding the revision.

With kind regards

Referee #1:

I really appreciate the quality and the quantity of the additional work that the authors performed. It significantly strengthens the study and provides optimized image quality as well as quantifications of the results. Unfortunately, a few of my original concerns were not fully addressed.

I do appreciate the honesty regarding ORF7a. I personally do not think that all published data has to be reproduced, especially if the own, carefully controlled and reproduced experiments paint a different picture. I do think that information would also be important for future readers. Thus, I would encourage the authors to comment on this finding in the paper, and not ignore it.

I had assumed that both virus-expressed Spike and endogenous tetherin represent proteins that should mainly localize to the plasma membrane. Thus, I was concerned why in the current study (usually in HeLa/A549 cells) both endogenous tetherin and Spike have a predominantly intracellular. This may be caused by different cell types or permeabilization methods. For example, in Fig. 4A or D and in Fig 5D or F mock, tetherin is a bit at the membranes but the majority of it seems to cluster at/around the TGN. Compare to that Fig. 1A and 2G, show a much more 'expected' Spike and tetherin localization. Thus, I was wondering whether the functional impact of tetherin on the virus/VLPs as well as shifts in tetherin localization upon ORF7a expression can be accurately assessed in HeLa/A549 cells. Can the authors explain/clarify this?

As suggested, I have added a comment at Line 609-613 which states our experiments may differ from others due to differences in cell lines and endogenous vs ectopic tetherin.

The points on localisation of tetherin / Spike are covered in the previous rebuttal (25th July 2023).

The VLP experiments still seem a bit of an inferior and convoluted approach compared to experiments with genuine SARS-CoV-2. I understand the argument that this system separates replication and entry and I do realize that VLP or Pseudotype systems have been used in the majority of previous studies. However, I am concerned that this may have led occasionally to misleading results. Especially the last two/three years have highlighted for me that pseudotyping systems have been excessively used to replace genuine virus experiments, sometimes resulting in conflicting data i.e. interpretations that may be true for the

pseudotype, but not the genuine virus. Thus, I would have appreciated that the impact on entry would have been studied using real SARS-CoV-2 that has the right shape and protein composition to accurately assess the impact of tetherin. Experimentally the VLP data can be complemented, by synchronizing a SARS-CoV-2 infection, and then quantifying the release of viral genomic RNA into the cytoplasm or quantifying the first subgenomic mRNAs appearing.

As discussed in our call on 18th August, we believe that genuine virus experiments are not appropriate here as ORF3a-deletion viruses are severely attenuated and so equal levels of infectivity cannot be achieved. We have added a comment to the discussion to reflect this (Line 534 - 542).

Referee #2:

Stewart et al. have addressed many of the points I raised in my initial review and they have improved their manuscript. However, I think there are still a few changes that are necessary before the paper can be published.

My major concern is that, in my view, the authors need to be more cautious with some of their conclusions. For example, ORF3a does not appear to affect tetherin expression (line 389-392), yet the authors report that virus infection of cells does decrease tetherin levels (e.g. line 189). Although the data show an impact of ORF3a expression on tetherin retrograde trafficking, they do not actually know the extent to which tetherin levels are modulated in the early compartments of the exocytic pathway, e.g. ERGIC, where virions are believed to assemble (rebuttal letter). For these reasons, the authors need to be very cautious with their conclusions, statements such as 'By removing tetherin from the Coronavirus budding compartments, ORF3a enhances virus release.' (line 34) are in my view not fully supported by the data.

We agree with the point here regarding careful wording regarding how ORF3a impairs tetherin function. We have edited the text to make sure that we do not state or infer that tetherin is removed from the ERGIC as we do not formally demonstrate this here. We do not formally demonstrate that the defective retrograde traffic of tetherin causes fewer tetherin molecules to be incorporated to forming virions, reducing virus tethering, so have removed the suggestion in the manuscript (e.g. Line 33).

We have modified the text relating to this point (including at line 33).

We have reworded Line 153 to clarify this point further.

We have edited Line 547 to state that ORF3a causes defective retrograde traffic, rather than inhibits the retrograde traffic of tetherin alone (as is the case with HIV Vpu) (see below).

We have been careful to limit our claims to the following points throughout the discussion:

- ORF3a expression alters the steady-state distribution of tetherin towards endolysosomal organelles without altering cellular levels.
- ORF3a limits the retrograde recycling of tetherin and causes it to accumulate in late endosomes / lysosomes.
- Expression of ORF3a enhances virus egress.

In addition, there are a number of minor points that should be fixed/considered.

Line 50 - Should be 'lysosome-associated' not 'lysosomal-associated'. – done (either is accepted for LAMP1). Now Line 52.

Line 54 - I don't think 'voracious' is an appropriate adjective. – done. Now Line 56.

Line 167 - Should 'uninfected' be 'infected'? – done. Now Line 171.

Line 168 - In my view 'demonstrating' should be 'suggesting' as they don't show that IFN is produced. – done. Now Line 172.

Line 189 - While it does appear that tetherin expression is lower in infected cells, I think it is very hard to see clearly that there is tetherin on the PM and less tetherin on the PM of infected cells. The authors should limit their claim to tetherin reduction in infected cells. – done. Line 198.

Line 231 and Fig.2D - the message is confusing. Line 231 states that very few surface-associated virions were present, likely due to the significant tetherin downregulation' but the Fig 2D legend states 'Infected cells displayed occasional virion clusters'. Can the authors clarify please. – done. Figure legend modified, Line 1233-1235.

Line 265 and Fig.EV2 - it's not clear to me how reliable the data is for intracellular infectious viruses; how stable are these viruses to freeze/thaw cycles and is it known whether these viruses have undergone furin-mediated S1/S2 cleavage. Virions with non-cleaved S presumably do not form plaques.

Methods state our method for quantifying released / released + intracellular virions.

Line 465 - the sentence 'Tetherin forms homodimers between opposing membranes (e.g., plasma membrane and viral envelope) that are linked via disulphide bonds.' Should be rewritten as 'Tetherin forms disulphide-linked homodimers that can link opposing membranes (e.g., plasma membrane and viral envelope).' To avoid suggesting membranes are linked by disulphide bonds. – done, now Line 505.

Line 497 - defective recycling of tetherin has been demonstrated for HIV-1 Vpu-mediated downregulation of tetherin; thus, it could be argued that altered trafficking is not a 'novel' mechanism. Done – we have reworded this to state that ORF3a causes defective retrograde traffic, rather than ORF3a having a direct effect on tetherin (as is the case for HIV-1 Vpu). Now Line 547.

Line 506 - I think the authors should perhaps give more emphasis to the Miller et al. paper (2023); these authors show very clearly that ORF3a is not an ion channel. - done, additional comment added at line 551-554.

Line 557 - the authors introduce the concept of cell-to-cell transmission - this needs to be explained. Done – extra text and edits added from 619- 629.

Referee #3:

In my original review I outline three major areas that required revision - image quantification, image annotation/explanation, and proper exploration/discussion of the data.

The authors have made considerably efforts to address these concerns - providing lots of image quantification data, annotation to EM images and two new paragraphs in the discussion. By and large I am satisfied that my concerns have been reasonably considered and addressed.

I would note that the authors have provided statistical significance testing on many, but not all, of the new data. For clarity, testing should be performed throughout, and if no significance is found this should be stated on the figure. This may also require some rewording to ensure the text properly reflects data significance.

We have now added 'ns' to Figure 3C where appropriate (and mentioned in figure legend) and added statistics to EV4G (and figure legend). Figure legends have also been updated to include all P values per each experiment.

Dr. James Edgar
University of Cambridge
Cambridge Institute for Medical Research
Wellcome Trust/MRC Building
Cambridge CB2 0XY
United Kingdom

Dear James,

I am very pleased to accept your manuscript for publication in the next available issue of EMBO reports. Thank you for your contribution to our journal.

At the end of this email I include important information about how to proceed. Please ensure that you take the time to read the information and complete and return the necessary forms to allow us to publish your manuscript as quickly as possible.

As part of the EMBO publication's Transparent Editorial Process, EMBO reports publishes online a Review Process File to accompany accepted manuscripts. As you are aware, this File will be published in conjunction with your paper and will include the referee reports, your point-by-point response and all pertinent correspondence relating to the manuscript.

If you do NOT want this File to be published, please inform the editorial office within 2 days, if you have not done so already, otherwise the File will be published by default [contact: emboreports@embo.org]. If you do opt out, the Review Process File link will point to the following statement: "No Review Process File is available with this article, as the authors have chosen not to make the review process public in this case."

Thank you again for your contribution to EMBO reports and congratulations on a successful publication. Please consider us again in the future for your most exciting work.

Kind regards,

Martina

THINGS TO DO NOW:

Please note that you will be contacted by Wiley Author Services to complete licensing and payment information. The required 'Page Charges Authorization Form' is available here: https://www.embopress.org/pb-assets/embo-site/er_apc.pdf - please download and complete the form and return to embopressproduction@wiley.com

EMBO Press participates in many Publish and Read agreements that allow authors to publish Open Access with reduced/no publication charges. Check your eligibility: <https://authorservices.wiley.com/author-resources/Journal-Authors/open-access/affiliation-policies-payments/index.html>

You will receive proofs by e-mail approximately 2-3 weeks after all relevant files have been sent to our Production Office; you should return your corrections within 2 days of receiving the proofs.

Please inform us if there is likely to be any difficulty in reaching you at the above address at that time. Failure to meet our deadlines may result in a delay of publication, or publication without your corrections.

All further communications concerning your paper should quote reference number EMBOR-2023-57224V3 and be addressed to emboreports@wiley.com.

Should you be planning a Press Release on your article, please get in contact with emboreports@wiley.com as early as possible, in order to coordinate publication and release dates.